# Celestial $w_{1+\infty}$ charges and the
# subleading structure of asymptotically-flat spacetimes

Marc Geiller

*Univ Lyon, ENS de Lyon, CNRS, Laboratoire de Physique, F-69342 Lyon, France*

**Abstract**

We study the subleading structure of asymptotically-flat spacetimes and its relationship to the $w_{1+\infty}$ loop algebra of higher spin charges. We do so using both the Bondi–Sachs and the Newman–Penrose formalism, via a dictionary built from a preferred choice of tetrad. This enables us to access properties of the so-called higher Bondi aspects, such as their evolution equations, their transformation laws under asymptotic symmetries, and their relationship to the Newman–Penrose and the higher spin charges. By studying the recursive Einstein evolution equations defining these higher spin charges, we derive the general form of their transformation behavior under BMSW symmetries. This leads to an immediate proof that the spin 0,1 and spin $s$ brackets reproduce upon linearization the structure expected from the $w_{1+\infty}$ algebra. We then define renormalized higher spin charges which are conserved in the radiative vacuum at quadratic order, and show that they satisfy for all spins the $w_{1+\infty}$ algebra at linear order in the radiative data.

# 1    Introduction

Asymptotically-flat spacetimes play a central role in the description of gravitational radiation [1–8]. They are also the natural arena where one can unravel and study the interplay between asymptotic symmetries [1–4], memory effects [9–13], and soft theorems for zero rest mass fields [14, 15]. The connection between these seemingly unrelated aspects, made explicit for the first time in [16–18], beautifully ties together non-perturbative observables with infrared properties of the classical and quantum gravitational scattering. Research along these lines has brought a wealth of new results, such as the discovery of new soft graviton theorems [19–27], new memory effects [28–39], and new asymptotic symmetries [40–51]. These developments are particularly interesting in light of the prospects for the detection of gravitational memories and their use in the improvement of waveforms [52–61]. They have also culminated in the introduction of celestial and Carrollian holography [62–69], which are proposals for a holographic description of asymptotically-flat spacetimes.

There are strong indications that soft theorems, memories, and asymptotic symmetries are organized in a tower of sub$^s$-leading tripartite relationships [70]. At leading order, the Ward identities arising from BMS supertranslations are equivalent to the soft graviton theorem, which can be written as the Fourier transform of the displacement memory, which itself can be understood as a radiation-induced transition between two vacua related by a supertranslation. At subleading order, enlargements of the BMS group (first to include superrotations [40–44, 71] and then to allow for arbitrary smooth diffeomorphisms of the asymptotic sphere [46, 47]) have enabled to establish a relationship between asymptotic symmetries, the subleading soft graviton theorem [19, 20, 72] and the so-called spin memory [28]. Incidentally, it is also this subleading structure which has pointed towards the existence of a dual two-dimensional conformal field theory living on the celestial sphere [62, 63, 73–75], eventually leading to the program of celestial holography. Going deeper down the tower, it has been shown that the sub-subleading soft graviton theorem [20–26, 76] and its collinear contribution can also be understood as the conservation law of a spin 2 charge generating a non-local symmetry [27], and in turn related to so-called higher memories [33–36, 61]. In light of these developments, a natural question is therefore how to get a handle on this subleading structure, and what are its organizing principles.

In the context of celestial holography, it was realized that an infinite tower of conformally soft graviton symmetries is generated when the leading and subleading soft graviton symmetries are supplemented by the sub-subleading soft graviton [77, 78]. It was subsequently shown in [79, 80] that this symmetry structure is that of the $w_{1+\infty}$ algebra [81, 82], which can also be unravelled in twistor theory [83, 84] and is known to be a symmetry of self-dual gravity [79, 85–88]. The representation of this $w_{1+\infty}$ symmetry algebra on the gravitational phase space in Bondi gauge was then studied in terms of higher spin charges in [89], where it was also shown that the soft part of the higher spin fluxes corresponds indeed to the tower of sub$^s$-leading soft gravitons. The relationship with twistors and celestial/Carrollian holography was further investigated in [69, 90–92]. In particular, it was shown in [92] that the non-local action of the higher spin symmetries at null infinity becomes local in twistor space. However, although the $w_{1+\infty}$ algebra can be extracted from the gravitational phase space of asymptotically-flat spacetimes, the role of the self-dual and single helicity conditions in this context are not clear, and it is therefore also not clear which sector of the gravitational dynamics is captured by this symmetry. We comment on this point at the beginning of section 3.3.

In the present work, our aim is to clarify and tighten the relationship between the $w_{1+\infty}$ algebra of celestial symmetries and the subleading structure of asymptotically flat spacetimes. When studying such spacetimes in Bondi coordinates, the solution space contains the shear, whose time evolution is unconstrained, and an infinite tower of fields satisfying flux-balance relations. This tower of fields features the mass at spin 0 (whose symmetry counterpart is the supertranslation), the angular momentum at spin 1 (whose symmetry counterpart is the superrotation), and then extends in terms of the so-called higher Bondi aspects, which are the spin 2 fields parametrizing the radial expansion of the transverse metric. In [93–95] it was shown

that these higher Bondi aspects and the higher spin charges are related to the canonical multipole moments of the linearized gravitational field. This suggests an interesting tomographic approach to reconstruct the metric from the knowledge of the higher spin celestial charges.

Here our goal is to explain in detail how the higher Bondi aspects are related to the generators of the $w_{1+\infty}$ algebra. In short, our starting point for this work is the line element (2.1) for asymptotically-flat spacetimes in Bondi–Sachs gauge, and our end point is the bracket of the $w_{1+\infty}$ algebra

$$\left\{Q_{s_1}(Z_1), Q_{s_2}(Z_2)\right\}^{(1)} = -Q^1_{s_1+s_2-1}\big((s_1+1)Z_1\eth Z_2 - (s_2+1)Z_2\eth Z_1\big) \tag{1.1}$$

where $Q_s(Z)$ are the smeared conserved charges of spin $s$ and $^{(1)}$ denotes the truncation to linear order[1]. Along the way, the construction exploits the Newman–Penrose formalism (NP hereafter) [96–100] and requires in particular to define the conserved higher spin charges. This contains many technicalities which we are now in position of working out building up on [89].

The outline of this work and of the new results is as follows. We start in section 2 by studying the asymptotic Einstein equations in Bondi–Sachs gauge (BS hereafter) at sub$^s$-leading order. This enables to access the explicit metric expressions for the evolution equations of the first few higher Bondi aspects, thereby recovering and extending the results of [35].

In section 3 we map these results in BS gauge to the NP formalism, and explain how the higher Bondi aspects appear in the radial expansion of the Weyl scalar $\Psi_0$. The flux-balance laws for the higher Bondi aspects are contained in the radial expansion of the evolution equation for $\Psi_0$. In order to write these evolution equations in compact form, a preferred choice of tetrad is required so that the spin coefficients satisfy $\kappa = \epsilon = \pi = 0$. We show that this is not satisfied by the "canonical" Bondi tetrad used e.g. in [49, 89, 101–103], but that it can be achieved with two Lorentz transformations. At the end of this construction, we obtain a NP representation of the solution space and the evolution equations in BS gauge. These equations differ slightly from those appearing e.g. in [44, 104–109] because these references use the Newman–Unti gauge instead of the Bondi–Sachs one. We then explain how higher spin charges can be identified in the radial expansion of $\Psi_0$, and set out to study their evolution equations.

The proof of the $w_{1+\infty}$ bracket of higher spin charges relies on the assumption that their Einstein evolution equations are given by the recursion relation (3.21). We verify that this recursion relation indeed holds for spins $-1 \leq s \leq 3$, but show that it develops unwanted contributions starting at spin 4. We clarify the nature of these terms by deriving the spin 4 and spin 5 evolution equations, and show that the recursion relation (3.21) can be obtained if we allow for non-local terms in the map (3.20) between the expansion of $\Psi_0$ and the higher spin charges. If we consider the positive and negative helicity shear $\sigma_2$ and $\bar{\sigma}_2$ as independent (as it would be the case in split signature), the constraint $\bar{\sigma}_2 = 0$ can be used as a self-dual condition. By comparing our spin 4 evolution equation with that derived in [104, 108] (they differ because they are obtained from two different null tetrads), we give evidence that there is an improved choice of tetrad for which the above non-local terms vanish in the self-dual theory. This hints towards a potential proof of the integrability of self-dual gravity directly in the NP formalism, but we keep this investigation for future work.

We continue in section 4 with a study of the action of BMS–Weyl (BMSW hereafter) transformations on the asymptotic solution space, and in particular on the higher spin charges. After writing the transformation laws of standard objects (i.e. shear, news, mass, angular momentum, . . . ) in metric and NP language, we study the action of BMSW transformations on the expansion of $\Psi_0$. We show that the transformation laws

---

[1] This truncation means that only a linear (i.e. soft) part is appearing on the right-hand side. This is possible because the defining bracket on the left-hand side contains only soft-soft and soft-hard contributions, but no hard-hard contributions (see (6.22) below).

for spins $-2 \leq s \leq 2$ follow the recursive pattern (4.18), but that the scalar $\Psi_0^1$ fails to transform accordingly. The recursive pattern is recovered when transforming instead the spin 3 charge defined as $\Psi_0^1 = -\bar{\eth} \mathcal{Q}_3$. This leads to one of the main results, which is formula (4.22) for the transformation law of all the higher spin charges under the action of spin 0 and spin 1 BMS transformations. We show that these transformation laws are the ones which are compatible with the recursive evolution equation (4.21). This proof is in the same spirit as the "gravity from symmetry" construction [110], since it ties the evolution equations of the higher spin charges with their transformation behavior under the BMSW group. From the result (4.22), one can already obtain the bracket (1.1) for $s_1 = 0, 1$ and $s_2 = s$.

In section 5 we briefly study for completeness the relationship between the expansion of $\Psi_0$, the subleading BMS charges [101–103], and the Newman–Penrose charges [104, 111]. We also compute the BMS charge algebra in NP form (5.14) [108] and the algebra of the real BMS fluxes (5.20) [112, 113].

Finally, section 6 is devoted to the study of the algebra of higher spin charges. Following [89], we first solve the recursion relation (4.21) to express the charges in terms of the news and the shear, and then decompose them as a sum (6.4) of soft, quadratic hard, and higher order contributions in the radiative data. We then study the action of the first higher spin charges $\mathcal{Q}_{0,1,2,3}$ on the shear in order to guess the form of the renormalized charges which are conserved in the radiative vacuum. This leads to the other main result of this work, which is formula (6.18) for the conserved higher spin charges up to quadratic order in the radiative data. This formula extends the proposal used in [89], which corresponds only to the first sum, and the paragraph below (6.31) explains this mismatch. We then show that the conserved higher spin charges (6.18) reproduce the correct action (6.35a) on the shear, and then use this result to compute the charge bracket at linear order. This computation, which leads to the $w_{1+\infty}$ bracket (1.1), is considerably shorter than the original proof given in [89] because we work directly with the smeared charges. As in [89], this derivation uses the fall-off conditions (6.2), which for arbitrary high spin implies in particular that the shear falls off faster than any power law in $u$ towards the corners of $\mathcal{I}^+$. Whether these (very restrictive) fall-off conditions can be relaxed is an open question.

We give perspectives for future work in section 7. The main text is followed by appendices in which we gather conventions and formulas to set up the NP formalism, various identities, and details on many of the calculations.

## 2 Subleading structure of the Bondi–Sachs gauge

A natural starting point for this work is to recall how to solve the Einstein equations in Bondi gauge near future null infinity using the metric formalism. While this is a standard calculation [1, 2, 5, 114, 115], we want nonetheless to perform it at a quite subleading order so as to access the evolution equations for the first few higher Bondi aspects. These metric expressions for the evolution equations will then be mapped to the corresponding equations in the NP formalism in section 3 as an important consistency check.

### 2.1 Solution space

Let us start by considering Bondi coordinates $(u, r, x^a)$ and the four-dimensional line element

$$ds^2 = \frac{V}{r} e^{2B} du^2 - 2e^{2B} du \, dr + \gamma_{ab}(dx^a - U^a du)(dx^b - U^b du), \tag{2.1}$$

where the functions $B(u, r, x^a)$, $V(u, r, x^a)$, and $U^a(u, r, x^a)$ will be determined below by four of the vacuum Einstein equations $\mathbb{E}_{\mu\nu} := R_{\mu\nu} = 0$ subject to the boundary conditions

$$g_{uu} = \mathcal{O}(1), \qquad g_{ur} = -1 + \mathcal{O}(r^{-2}), \qquad g_{ua} = \mathcal{O}(1), \qquad g_{ab} = \mathcal{O}(r^2). \tag{2.2}$$

The line element (2.1) satisfies the three Bondi gauge conditions $g_{rr} = 0 = g_{ra}$, which we supplement by the Bondi–Sachs (BS) determinant condition[2]

$$\mathbb{C} := \partial_r \left( \frac{\gamma}{r^4} \right) \overset{!}{=} 0, \tag{2.3}$$

where $\gamma := \det(\gamma_{ab})$. This differential condition will enable us to describe non-trivial Weyl transformations, as desired since we want to work in the context of BMSW. Let us now consider an expansion for the angular metric of the form[3]

$$\gamma_{ab} = r^2 q_{ab} + r C_{ab} + \boldsymbol{D}_{ab} + \sum_{n=1}^{\infty} \frac{\boldsymbol{E}_{ab}^n}{r^n}, \qquad \boldsymbol{E}_{ab}^n = \frac{1}{2} q_{ab} \boldsymbol{E}^n + E_{ab}^n, \tag{2.4}$$

where the tensors $\boldsymbol{D}_{ab}$ and $\boldsymbol{E}_{ab}^n$ are determined by their trace $\boldsymbol{E}^n$ and trace-free parts $E_{ab}^n$ with respect to $q_{ab}$. The symmetric and trace-free tensors $E_{ab}^n$ are the so-called higher Bondi aspects whose dynamics we want to study. Now, when plugging (2.4) into (2.3), the determinant condition translates into conditions on the traces[4]. More precisely, at leading order it implies that $q^{ab} C_{ab} = 0$, while for the first few subleading orders it fixes the traces in the following manner:

$$\boldsymbol{D} = \frac{1}{2}[CC], \tag{2.5a}$$

$$\boldsymbol{E}^1 = [CD], \tag{2.5b}$$

$$\boldsymbol{E}^2 = [CE^1] + \frac{1}{2}[DD] - \frac{1}{16}[CC]^2, \tag{2.5c}$$

$$\boldsymbol{E}^3 = [CE^2] + [DE^1] - \frac{1}{4}[CC][CD], \tag{2.5d}$$

$$\boldsymbol{E}^4 = [CE^3] + [DE^2] + \frac{1}{2}[E^1 E^1] - \frac{1}{8}[CC]\big(2[CE^1] + [DD]\big) - \frac{1}{4}[CD]^2 + \frac{1}{64}[CC]^3. \tag{2.5e}$$

Our compact notation for the contraction of tensors is $[CD] = C^{ab} D_{ab}$, and we recall that symmetric trace-free tensors satisfy

$$C_{ad} C_b^d = \frac{1}{2} q_{ab}[CC]. \tag{2.6}$$

The determinant condition (2.3) implies that $\sqrt{\gamma} = r^2 \sqrt{q}$ and therefore $\sqrt{-g} = r^2 e^{2B} \sqrt{q}$. It also gives the identity $\mathcal{D}_a V^a = D_a V^a$ for any vector $V^a$, where $\mathcal{D}_a$ and $D_a$ are the covariant derivatives with respect to $\gamma_{ab}$ and $q_{ab}$ respectively. Note that we can also solve the determinant condition in an elegant manner by writing directly the transverse metric in terms of the trace-free tensors as [35]

$$\gamma_{ab} = r^2 q_{ab} \sqrt{1 + \frac{[\mathcal{CC}]}{2r^2}} + r\mathcal{C}_{ab}, \qquad r\mathcal{C}_{ab} := r C_{ab} + D_{ab} + \sum_{n=1}^{\infty} \frac{E_{ab}^n}{r^n}. \tag{2.7}$$

While it is simpler at this stage to use this expansion instead of (the equivalent form) (2.4), we will see that the latter is more convenient in order to derive the transformation laws of the higher Bondi aspects.

---

[2]Note that we could have chosen to work with the Newman–Unti gauge condition $B = 0$ instead. This has the advantage of simplifying slightly the map between the metric and NP formalism, however at the expense of a departure from the standard known formulas for the solution space, the asymptotic symmetries, and the charges.

[3]We exclude terms in $\ln(r)$ from this expansion although they are allowed by the Einstein equations. For discussions on polyhomogeneous terms and violations of peeling, please see [50, 116–118].

[4]The Newman–Unti gauge also implies (different) conditions on the traces, which are obtained when inserting the gauge condition $B = 0$ in the Einstein equation $\mathbb{E}_{rr} = 0$ given in (2.10) [50]. For example in the Newman–Unti gauge (2.5a) is replaced by $4\boldsymbol{D} = [CC]$.

Let us now set once and for all $D_{ab} = 0$ in order to remove the logarithmic terms which would otherwise appear in the solution for $U^a$ [50, 116–118]. Finally, we will also take the leading boundary metric to be time-independent, i.e. set $\partial_u q_{ab} = 0$.

Now that we have imposed the BS gauge and written the transverse metric satisfying the determinant condition, we can solve the Einstein equations. In particular, we are interested in the evolution equations for the higher Bondi aspects, which are the trace-free terms $E^n_{ab}$ appearing at order $r^{-n}$ in (2.4). These evolution equations are actually the ones appearing at order $r^{-(n+1)}$ in the Einstein equations

$$\mathbb{E}^{\text{TF}}_{ab} := R_{ab} - \frac{1}{2}\gamma_{ab}(\gamma^{cd}R_{cd}) = 0, \tag{2.8}$$

where $R_{ab}$ is the Ricci tensor and TF refers to the trace-free part in the metric $\gamma_{ab}$. The first question we need to answer is therefore at which order to expand the functions $(B, U^a, V)$ in order to access these evolution equations. Then we can also ask at which order to solve the other Einstein equations in order to determine these expansions. The answer is that to obtain $\mathbb{E}^{\text{TF}}_{ab}\big|_{\mathcal{O}(r^{-(n+1)})}$ we need to know $(\gamma^n_{ab}, B_{n+1}, U^a_{n+2}, V_{n-1})$, and these coefficients of the expansion are obtained by solving respectively and in that order

$$\mathbb{C} = \mathcal{O}(r^{-(n+4)}), \qquad \mathbb{E}_{rr} = \mathcal{O}(r^{-(n+4)}), \qquad \mathbb{E}_{ra} = \mathcal{O}(r^{-(n+3)}), \qquad \mathbb{E}_{ru} = \mathcal{O}(r^{-(n+3)}). \tag{2.9}$$

Let us now give explicitly the expansions which solve these equations up to $n = 4$. In (2.7) we have already written the angular metric which solves the determinant condition. This metric then enters the $(rr)$ Einstein equation

$$\mathbb{E}_{rr} := R_{rr} = \frac{4}{r}\partial_r B + \frac{2}{r^2} + \frac{1}{4}\left(\partial_r\gamma^{ab}\right)\left(\partial_r\gamma_{ab}\right) = 0, \tag{2.10}$$

which can be solved for $B$. The solution is $B = r^{-2}B_2 + r^{-3}B_3 + r^{-4}B_4 + r^{-5}B_5 + \mathcal{O}(r^{-6})$ with

$$B_2 = -\frac{1}{32}[CC], \tag{2.11a}$$

$$B_3 = 0, \tag{2.11b}$$

$$B_4 = -\frac{3}{32}[CE^1] + \frac{1}{128}[CC]^2, \tag{2.11c}$$

$$B_5 = -\frac{1}{10}[CE^2], \tag{2.11d}$$

where the boundary conditions (2.2) have enforced $B_0 = 0$. The next Einstein equation to solve is then

$$\mathbb{E}_{ra} := R_{ra} = \frac{1}{2r^2}\partial_r\left(r^2 e^{-2B}\gamma_{ab}\partial_r U^b\right) + \frac{2}{r}\partial_a B - \partial_a\partial_r B - \frac{1}{2}\gamma_{ac}\mathcal{D}_b\partial_r\gamma^{bc} = 0. \tag{2.12}$$

The solution is $U^a = r^{-2}U_2^a + r^{-3}U_3^a + r^{-4}U_4^a + r^{-5}U_5^a + r^{-6}U_6^a + \mathcal{O}(r^{-7})$ with

$$U_2^a = -\frac{1}{2}D_b C^{ab}, \tag{2.13a}$$

$$U_3^a = N^a, \tag{2.13b}$$

$$U_4^a = \frac{3}{4}D_b E_1^{ab} - \frac{3}{4}C^{ab}N_b - \frac{1}{16}C^{ab}\partial_b[CC] + \frac{1}{64}[CC]D_b C^{ab}, \tag{2.13c}$$

$$U_5^a = \frac{2}{5}D_b E_2^{ab} - \frac{2}{5}C^{ab}D^c E_{bc}^1 - \frac{1}{10}C^{bc}D^a E_{bc}^1 + \frac{1}{16}\partial^a[CE^1] + \frac{6}{320}[CC]\big(6N^a + \partial^a[CC]\big), \tag{2.13d}$$

$$U_6^a = \frac{5}{18}D_b E_3^{ab} - \frac{1}{6}C^{ab}D^c E_{bc}^2 - \frac{1}{12}C^{bc}D^a E_{bc}^2 + \frac{1}{20}\partial^a[CE^2] - \frac{1}{2}E_1^{ab}N_b$$
$$+ \frac{17}{576}E_1^{ab}\partial_b[CC] - \frac{1}{16}E_1^{ad}C^{cb}D_b C_{cd} + \frac{5}{96}C^{ab}\big(C^{cd}D_b E_{cd}^1 - 2\partial_b[CE^1]\big)$$
$$+ \frac{1}{32}[CC]\left(\frac{7}{9}D_b E_1^{ab} + C^{ab}N_b + \frac{1}{4}C^{ab}\partial_b[CC] - \frac{3}{32}[CC]D_b C^{ab}\right), \tag{2.13e}$$

$$U_{n\geq4}^a = \frac{(n-1)}{n(n-3)}D_b E_{n-3}^{ab} + (\text{NL}), \tag{2.13f}$$

where $N^a(u, x^b)$ is a radial integration constant, and where we have set $U_0^a = 0$ as required by the boundary conditions. Here (NL) denotes non-linear terms which we omit. The third and last hypersurface equation is $\mathbb{E}_{ur} := R_{ur} = 0$, which is equivalent to $\gamma^{ab}R_{ab} = 0$ with

$$\gamma^{ab}R_{ab} = R[\gamma] + e^{-2B}\left(\frac{2}{r^2}\partial_r V + \frac{4}{r}D_a U^a + \partial_r\big(D_a U^a\big) - \frac{1}{2}e^{-2B}\gamma_{ab}\big(\partial_r U^a\big)\big(\partial_r U^b\big)\right)$$
$$- 2\partial_a\big(\gamma^{ab}\partial_b B\big) - 2\gamma^{ab}\big(\partial_a B\big)\partial_b\big(\ln\sqrt{q} + B\big). \tag{2.14}$$

The solution is $V = rV_{+1} + V_0 + r^{-1}V_1 + r^{-2}V_2 + r^{-3}V_3 + \mathcal{O}(r^{-4})$ with

$$V_{+1} = -\frac{R}{2}, \tag{2.15a}$$

$$V_0 = 2M, \tag{2.15b}$$

$$V_1 = \frac{1}{2}D_a N^a + \frac{1}{8}\big(D_a C_{bc}\big)\big(D^a C^{bc}\big) - \frac{1}{2}\big(D_a C^{ab}\big)\big(D^c C_{cb}\big) + \frac{3}{32}(R - D^2)[CC], \tag{2.15c}$$

$$V_2 = \frac{1}{4}D_a D_b E_1^{ab} + \frac{1}{2}D_a\big(C^{ab}\partial_b B_2\big) - \frac{1}{2}C_{ab}U_2^a U_2^b - \frac{3}{2}N_a U_2^a - B_2 D_a U_2^a, \tag{2.15d}$$

$$V_3 = \frac{1}{10}D_a D_b E_2^{ab} - \frac{3}{4}N^a N_a + (\text{NL with } C_{ab}), \tag{2.15e}$$

$$V_{n\geq2} = \frac{1}{(n-1)(n+2)}D_a D_b E_{n-1}^{ab} + (\text{NL}), \tag{2.15f}$$

where $M(u, x^a)$ is a radial integration constant and where once again we have omitted non-linear terms. Note that the Ricci scalar which appears here is $R = R[q]$.

We have now determined the first few terms in the radial expansion of the line element (2.1) by solving the four hypersurface equations contained in the Einstein equations. In the linearized theory each term has an explicit known form as given above, while in the full non-linear theory the non-linear terms must be computed at each subleading order. In summary, the solution space contains infinitely-many functions of $(u, x^a)$, namely the two integration constants $(M, N^a)$ and the symmetric trace-free tensors $(C_{ab}, E_{ab}^n)$ appearing in the expansion (2.4). As is well-known, the shear $C_{ab}$ represents completely free data on $\mathcal{I}^+$ whose time dependency is undetermined, while $(M, N^a, E_{ab}^n)$ satisfy evolution equations, or flux-balance laws. We will now study these evolution equations.

## 2.2 Evolution equations

In order to write down the evolution equations in a compact form and map them easily to the NP formalism, it is convenient to introduce some notations and field redefinitions. First, we will denote the time derivative of the shear by $N_{ab} = \partial_u C_{ab}$, but keep in mind that this is not strictly speaking the news since it does not contain the contribution from the Geroch tensor [8, 37, 45]. Then, jumping a bit ahead of ourselves, let us consider the objects defined in (3.12), which are often referred to as covariant functionals because of their behavior under BMSW transformations [110]. In terms of these quantities the first evolution equations are

$$\text{definition (3.12e)} \quad \Rightarrow \quad \partial_u \mathcal{J}^a = D_b \mathcal{N}^{ab}, \tag{2.16a}$$

$$\text{definitions (3.12)} \quad \Rightarrow \quad \partial_u \widetilde{\mathcal{M}} = D_a \widetilde{\mathcal{J}}^a + \frac{1}{2} C_{ab} \widetilde{\mathcal{N}}^{ab}, \tag{2.16b}$$

$$\left. \mathbb{E}_{uu} \right|_{\mathcal{O}(r^{-2})} \quad \Rightarrow \quad \partial_u \mathcal{M} = D_a \mathcal{J}^a + \frac{1}{2} C_{ab} \mathcal{N}^{ab}, \tag{2.16c}$$

$$\left. \mathbb{E}_{ua} \right|_{\mathcal{O}(r^{-2})} \quad \Rightarrow \quad \partial_u \mathcal{P}_a = D^b \mathcal{M}_{ab} + 2 C_{ab} \mathcal{J}^b, \tag{2.16d}$$

where we have introduced

$$\mathcal{M}_{ab} \coloneqq \mathcal{M} q_{ab} + \widetilde{\mathcal{M}} \varepsilon_{ab}, \qquad \widetilde{T}^{a \dots a_n} \coloneqq \varepsilon^a{}_b T^{b \dots a_n}, \qquad \varepsilon_{ab} \coloneqq \sqrt{q} \, \epsilon_{ab}, \tag{2.17}$$

and where $\epsilon_{ab}$ is the Levi–Civita symbol. The first of these equations is the evolution of the so-called energy current, and it follows tautologically from the definitions (3.12e) and (3.12f). Similarly, the second equation, which is the evolution of the dual mass, also follows from its definition. The last two equations, which are the flux-balance laws for the mass (i.e. the Bondi mass loss) and the angular momentum, are the only non-trivial informations in the $(uu)$ and $(ua)$ Einstein equations.

The rest of the evolution equations comes entirely from the remaining set (2.8) of Einstein equations[5]. Instead of writing these equations in terms of $E_{ab}^n$, we are going to write them directly in terms of the quantities $\mathcal{E}_{ab}^n$ which we introduce below in (3.16). This will indeed be more convenient when matching these tensorial evolution equations with the scalar NP evolution equations in the next section. Note that all the objects $\mathcal{E}_{ab}^n$ are also rank 2 symmetric trace-free tensors. With a slight abuse of language we will also refer to them as the higher Bondi aspects. The first non-trivial equation is $\left. \mathbb{E}_{ab}^{\mathrm{TF}} \right|_{\mathcal{O}(r^{-2})} = 0$, which takes the form[6]

$$\partial_u \mathcal{E}_{ab}^1 = D_{\langle a} \mathcal{P}_{b \rangle} + \frac{3}{2} \mathcal{M}_{ac} C^c{}_b. \tag{2.18}$$

This is the spin 2 evolution equation already written and studied in [32, 35, 110]. Now, as the subleading equations are going to involve some of the overleading ones, we use the equality $\widehat{=}$ to denote that we iteratively go on-shell. From $\left. \mathbb{E}_{ab}^{\mathrm{TF}} \right|_{\mathcal{O}(r^{-3})} = 0$ we then find

$$\partial_u \mathcal{E}_{ab}^2 \widehat{=} - \frac{1}{2} \big( D^2 + R \big) \mathcal{E}_{ab}^1 - 2 D^c \big( \mathcal{P}_{\langle a} C_{b \rangle c} \big)$$

$$= -D^c \Big( D_{\langle a} \mathcal{E}_{b \rangle c}^1 + 2 \mathcal{P}_{\langle a} C_{b \rangle c} \Big), \tag{2.19}$$

in agreement with [35], and where for the second line we have used (B.1). Then, from $\left. \mathbb{E}_{ab}^{\mathrm{TF}} \right|_{\mathcal{O}(r^{-4})} = 0$ we get

---

[5] Another interesting perspective on the flux-balance laws for the higher Bondi aspects comes from the work [119], in which the evolution equations are derived from the flat limit of the AdS energy-momentum tensor.

[6] We denote $2 D_{\langle a} \mathcal{P}_{b \rangle} = 2 D_{(a} \mathcal{P}_{b)} - q_{ab} D_c \mathcal{P}^c = D_a \mathcal{P}_b + D_b \mathcal{P}_a - q_{ab} D_c \mathcal{P}^c$.

the evolution equation

$$\partial_u \mathcal{E}^3_{ab} \,\widehat{=}\, -\frac{1}{4}\big(D^2 + 4R\big)\mathcal{E}^2_{ab} + \frac{5}{8}[N\mathcal{E}^1]C_{ab} - \frac{5}{32}\partial_u[CC]\mathcal{E}^1_{ab} - \frac{5}{8}[C\mathcal{E}^1]N_{ab}$$
$$- 5\mathcal{P}_{\langle a}\mathcal{P}_{b\rangle} + \frac{1}{4}D_{\langle a}\big([CC]\mathcal{P}_{b\rangle}\big) + \frac{5}{2}\mathcal{P}_{\langle a}C_{b\rangle c}D_d C^{cd}$$
$$+ \frac{15}{2}\mathcal{M}\mathcal{E}^1_{ab} + \frac{15}{8}\big(D_a D_b C^{ab}\big)\mathcal{E}^1_{ab} - \frac{5}{2}\mathcal{E}^1_{\langle ac}D^c D^d C_{b\rangle d} + \frac{5}{4}C_{\langle ac}D^c D^d \mathcal{E}^1_{b\rangle d}$$
$$+ \frac{1}{4}C^{cd}D_c D_{\langle a}\mathcal{E}^1_{b\rangle d} + \frac{5}{2}D_c C^{cd}D_d \mathcal{E}^1_{ab} - D_c C^{cd}D_{\langle a}\mathcal{E}^1_{b\rangle d}. \tag{2.20}$$

Since the equation $\mathbb{E}^{\mathrm{TF}}_{ab}\big|_{\mathcal{O}(r^{-5})} = 0$ is very lengthy, we choose to display it without writing down the shear terms explicitly. This will be sufficient for our purposes. The evolution equation for the next higher Bondi aspect is then

$$\partial_u \mathcal{E}^4_{ab} \,\widehat{=}\, -\frac{1}{6}\big(D^2 + 8R\big)\mathcal{E}^3_{ab} + 8\mathcal{M}\mathcal{E}^2_{ab} + (\text{NL with } C_{ab}) \tag{2.21}$$
$$+ \frac{1}{3}\Big(4D_c\mathcal{P}^c\mathcal{E}^1_{ab} + 7\mathcal{P}^c D_c\mathcal{E}^1_{ab} - 14\mathcal{E}^1_{\langle ac}D^c\mathcal{P}_{b\rangle} + 18\mathcal{P}_{\langle a}D^c\mathcal{E}^1_{b\rangle c} - 4D_{\langle a}\big(\mathcal{E}^1_{b\rangle c}\mathcal{P}^c\big)\Big).$$

Finally, neglecting the non-linear terms, one can show that the general form of the evolution equations contained in $\mathbb{E}^{\mathrm{TF}}_{ab}\big|_{\mathcal{O}(r^{-(n+1)})}$ is

$$\partial_u E^{n\geq 2}_{ab} = -\frac{n}{2(n+2)(n-1)}\left(D^2 + \frac{1}{2}(n^2 + n - 4)R\right)E^{n-1}_{ab} + (\text{NL}), \tag{2.22a}$$
$$\partial_u \mathcal{E}^{n\geq 2}_{ab} = -\frac{1}{2(n-1)}\left(D^2 + \frac{1}{2}(n^2 + n - 4)R\right)\mathcal{E}^{n-1}_{ab} + (\text{NL}') =: \mathscr{D}_n\mathcal{E}^{n-1}_{ab} + (\text{NL}'), \tag{2.22b}$$

where the set of omitted non-linear terms is different. The first line is equation (2.23) of [35] with the shift $n_{\mathrm{here}} = n_{\mathrm{there}} + 2$, while the equivalent form on the second line has been obtained using the relationship (3.16e) between $E^n_{ab}$ and $\mathcal{E}^n_{ab}$. On the second line we have also introduced the differential operator $\mathscr{D}_n$ which controls the evolution of the higher Bondi aspect $\mathcal{E}^n_{ab}$ at the linear level.

## 2.3 Conserved Bondi aspects

We are eventually interested in constructing charges which are conserved in the absence of radiation (or "quasi-conserved" charges). In order to give a precise meaning to this, let us consider the non-radiative vacuum conditions $\mathcal{J}^a \stackrel{\mathrm{vac}}{=} 0 \stackrel{\mathrm{vac}}{=} \mathcal{N}^{ab}$. From (2.16c) one can see that $\partial_u \mathcal{M} \stackrel{\mathrm{vac}}{=} 0$, while (2.16d) shows that $\mathcal{P}_a$ is not conserved in the vacuum. Similarly, the higher Bondi aspects $\mathcal{E}^n_{ab}$ are not conserved in the vacuum either. Conserved quantities in the radiative vacuum can be obtained instead by considering explicit time-dependent combinations of the various data appearing in the solution space. For example, conserved spin 1 and spin 2 charges can be defined as

$$q_{a,1} := \mathcal{P}_a - u D^b \mathcal{M}_{ab}, \tag{2.23a}$$
$$q_{ab,2} := \mathcal{E}^1_{ab} - u\left(D_{\langle a}\mathcal{P}_{b\rangle} + \frac{3}{2}\mathcal{M}_{ac}C^c{}_b\right) + \frac{u^2}{2}\left(D_{\langle a}D^c\mathcal{M}_{b\rangle c} + \frac{3}{2}\mathcal{M}_{ac}N^c{}_b\right). \tag{2.23b}$$

Indeed, these charges have fluxes given by

$$\partial_u q_{a,1} = 2C_{ab}\mathcal{J}^b - u D^b \partial_u \mathcal{M}_{ab}, \tag{2.24a}$$
$$\partial_u q_{ab,2} = -u\left(2D_{\langle a}\big(C_{b\rangle c}\mathcal{J}^c\big) + \frac{3}{2}\partial_u\mathcal{M}_{ac}C^c{}_b\right) + \frac{u^2}{2}\left(D_{\langle a}D^c\partial_u\mathcal{M}_{b\rangle c} + \frac{3}{2}\partial_u\big(\mathcal{M}_{ac}N^c{}_b\big)\right), \tag{2.24b}$$

and these expressions are clearly vanishing in the radiative vacuum defined above since $\partial_u \mathcal{M}_{ab} \overset{\text{vac}}{=} 0$. It is important to note however that the renormalized conserved charges (2.23) are not uniquely defined, since one can add to them any term whose time evolution is proportional to $\mathcal{J}^a$ or $\mathcal{N}^{ab}$ without spoiling the conservation property. One may for example consider the alternative renormalized charges

$$q_{a,1} := \mathcal{P}_a - u\partial_u \mathcal{P}_a, \qquad q_{ab,s \geq 2} := \sum_{n=0}^{s} \frac{(-u)^n}{n!} \partial_u^n \mathcal{E}_{ab}^{s-1}, \qquad (2.25)$$

which evolve as

$$\partial_u q_{a,1} = -u\partial_u^2 \mathcal{P}_a \overset{\text{vac}}{=} 0, \qquad \partial_u q_{ab,s} = \frac{(-u)^s}{s!} \partial_u^{s+1} \mathcal{E}_{ab}^{s-1} \overset{\text{vac}}{=} 0, \qquad (2.26)$$

and are therefore indeed conserved in the non-radiative vacuum. However, one can see that the definitions for $q_{a,1}$ and $q_{ab,2}$ in (2.23) and (2.25) clearly differ.

Alternative proposals for such renormalized Bondi aspects satisfying conservation properties were also studied in [35, 95]. With these various prescriptions, the flux-balance laws for the renormalized higher Bondi aspects are of the schematic form[7]

$$\partial_u q_{ab,2} = \frac{(-u)^2}{2!} D_{\langle a}\big(D_{b\rangle} D_c \mathcal{J}^c + \widetilde{D}_{b\rangle} D_c \widetilde{\mathcal{J}}^c\big) + \mathcal{F}_{ab}^2, \qquad (2.27\text{a})$$

$$\partial_u q_{ab,s \geq 3} = \frac{(-u)^s}{s!} \mathscr{D}_{s-1} \ldots \mathscr{D}_2 D_{\langle a}\big(D_{b\rangle} D_c \mathcal{J}^c + \widetilde{D}_{b\rangle} D_c \widetilde{\mathcal{J}}^c\big) + \mathcal{F}_{ab}^s, \qquad (2.27\text{b})$$

where the first term is the soft piece involving the operators defined in (2.22), and $\mathcal{F}$ is the hard flux (whose general schematic form can be found in [35]). The former can be derived by combining the linear contributions to (2.26) with (2.22) and (2.16). This soft contribution to the flux is related upon integration over $u$ to the higher moments of the news, which are themselves related to the so-called higher memories defined and studied in [33–36, 61].

We are going to introduce and study yet another set of conserved charges (6.18) in section 6 in order to realize the $w_{1+\infty}$ higher spin symmetry algebra. However, since our proposal for the conserved charges is only known in a truncation to quadratic hard order, it cannot fully settle the above-mentioned ambiguities. The spin 1 charge (6.10b) which belongs to the $w_{1+\infty}$ algebra is the dressed angular momentum aspect (2.23) studied e.g. in [37, 120]. However, the spin 2 charge (6.10c) which we will consider differs from both (2.23b) and (2.25). We come back to the precise definition of these conserved charges using the NP formalism in section 6. Up to quadratic order in the radiative data, the flux-balance law for these conserved higher spin charges in NP form is given by (6.20). In particular, the soft flux (6.24) agrees with the soft part of (2.27) upon mapping the higher Bondi aspects $q_{ab,s}$ to the renormalized higher spin charges $q_s$. The hard flux is only known exactly up to spin $s = 3$ in (6.15d).

## 3   Map to the Newman–Penrose formalism

In the previous section we have characterized the solution space in Bondi gauge using the metric formalism, and worked out the first few evolution equations for the data $(\mathcal{M}, \mathcal{P}_a, \mathcal{E}_{ab}^n)$. Our goal is now to obtain an equivalent description of this data and of the evolution equations in the NP formalism [96–100], i.e. in terms of null frames, spin coefficients, and Weyl scalars, as this will be the most convenient framework to study the $w_{1+\infty}$ higher spin charges.

---

[7]The soft parts can be rewritten as in (5) of [95] using the identity $D_a D_b D_c C^{bc} + \widetilde{D}_a D_b D_c \widetilde{C}^{bc} = 2D_c D_{\langle a} D_{b\rangle} C^{bc}$.

The NP formalism has already been applied to the study of BMS asymptotic symmetries and charges in [44, 105–109]. Here however, at the difference with these references, we will not solve the NP evolution equations (i.e. the Einstein equations rewritten in first order scalar form) from scratch. Indeed, this would be redundant since we have already solved the Einstein equations and obtained the solution space in metric form. Instead, our goal is to simply establish a dictionary with the NP formalism. This has not been studied in detail previously because the NP and metric formalisms are usually considered on their own, as standalone approaches to solve the Einstein equations and study the asymptotic symmetries. Moreover, in the NP formalism it is customary to work in the Newman–Unti (NU) gauge (as in the references above), while the metric Einstein equations are usually solved in the Bondi–Sachs (BS) gauge (with the notable exception of [50, 51]). However, this does not prevent us from taking the metric solution space in BS gauge and rewriting it in terms of NP quantities. We simply have to be aware, as we will see, that this leads to NP expressions which differ from those present e.g. in [44, 105–109].

## 3.1 Choice of null tetrad

A first subtlety which we need to address is the choice of null frame. The NP formalism is covariant under Lorentz transformations of the tetrad, meaning that these transformations change the various expressions (for the tetrad itself, the Weyl scalars, the spin coefficients, . . . ) without altering the underlying physics. Since here we are especially interested in rewriting the evolution equations for $\mathcal{E}_{ab}^n$ in NP language, we want to exploit this ambiguity in order to find a choice of tetrad which leads to the simplest possible form of these equations. It turns out that this happens when the spin coefficients $\kappa$, $\epsilon$ and $\pi$ (whose definition is given below in equations (3.3) and (3.2)) are vanishing.

In typical studies of BMS symmetries in the NP formalism (see e.g. [108]), the gauge choice $\kappa = \epsilon = \pi = 0$ is made from the onset, and the tetrad is then reconstructed by solving the so-called metric equations (which form one of the three sets of NP equations). As mentioned above, here we want to proceed the other way around, and instead start from a choice of tetrad which describes the on-shell metric (2.1) before then using it to construct the NP data. A natural choice is to consider the so-called Bondi tetrad $\tilde{e}_i^\mu = (\tilde{\ell}, \tilde{n}, \tilde{m}, \bar{\tilde{m}})$ formed by the null vectors

$$\tilde{\ell} := \partial_r, \qquad \tilde{n} := e^{-2B}\left(\partial_u + \frac{V}{2r}\partial_r + U^a\partial_a\right), \qquad \tilde{m} := \sqrt{\frac{\gamma_{\theta\theta}}{2\gamma}}\left(\frac{\sqrt{\gamma} + i\gamma_{\theta\phi}}{\gamma_{\theta\theta}}\partial_\theta - i\partial_\phi\right). \qquad (3.1)$$

This is for example the choice made in [49, 89, 101–103], and also often in numerical relativity [121, 122]. This tetrad is such that $\tilde{\ell}^\mu \tilde{n}_\mu = -1 = -\tilde{m}^\mu \bar{\tilde{m}}_\mu$ with all the other contractions vanishing, or in other words $g_{\mu\nu}\tilde{e}_i^\mu \tilde{e}_j^\nu = \eta_{ij}$ with the double null internal metric $\eta_{34} = 1 = -\eta_{12}$. The spacetime metric is then given by $g^{\mu\nu} = \tilde{e}_i^\mu \tilde{e}_j^\nu \eta^{ij} = -2\tilde{\ell}^{(\mu}\tilde{n}^{\nu)} + 2\tilde{m}^{(\mu}\bar{\tilde{m}}^{\nu)}$. Starting from this Bondi tetrad, let us now consider the associated spin coefficients

$$\tilde{\gamma}_{ijk} := \tilde{e}_j^\mu \tilde{e}_k^\nu \nabla_\nu \tilde{e}_{i\mu}, \qquad (3.2)$$

and compute in particular

$$\tilde{\kappa} = \tilde{\gamma}_{131} = 0, \qquad (3.3a)$$

$$\tilde{\rho} = \tilde{\gamma}_{134} = \frac{1}{2}\partial_r \ln\sqrt{\gamma} = \bar{\tilde{\rho}}, \qquad (3.3b)$$

$$\tilde{\epsilon} = \frac{1}{2}(\tilde{\gamma}_{121} - \tilde{\gamma}_{341}) = \frac{1}{8}\partial_r \gamma_{ab}(\tilde{m}^a \tilde{m}^b - \bar{\tilde{m}}^a \bar{\tilde{m}}^b) - \partial_r B, \qquad (3.3c)$$

$$\tilde{\pi} = \tilde{\gamma}_{421} = -\left(\frac{1}{2}e^{-2B}\gamma_{ab}\partial_r U^b + \partial_a B\right)\bar{\tilde{m}}^a. \qquad (3.3d)$$

The fact that $\tilde{\kappa} = 0$ means that $\tilde{\ell}$ describes a congruence of null geodesics, and $\text{Im}(\tilde{\rho}) = 0$ that it is furthermore hypersurface-orthogonal (or rotation-free). We see however that with the Bondi tetrad (3.1) we have $\tilde{\epsilon} \neq 0 \neq \tilde{\pi}$ (even if we had chosen the NU gauge condition $B = 0$ instead of the BS condition (2.3)). This means that the four basis vectors $\tilde{e}_i^\mu$ are not parallelly-transported along $\tilde{\ell}$. Working with this tetrad will turn out to be rather inconvenient when writing down the evolution equations in NP form, as we will illustrate below.

There is however a way out of this inconvenience, as the complex spin coefficients $\tilde{\epsilon}$ and $\tilde{\pi}$ can be set to zero by fixing four conditions via two Lorentz transformations [98]. The first one has two real parameters, and the second one a complex parameter. First, we perform a rotation of class III, defined with two real functions $\theta_1$ and $\theta_2$ by

$$\tilde{\ell} \to e^{\theta_1}\tilde{\ell}, \qquad \tilde{n} \to e^{-\theta_1}\tilde{n}, \qquad \tilde{m} \to e^{i\theta_2}\tilde{m}. \tag{3.4}$$

The effect of this rotation is to transform the spin coefficients of interest as

$$\tilde{\kappa} \to e^{2\theta_1+i\theta_2}\tilde{\kappa}, \qquad \tilde{\rho} \to e^{\theta_1}\tilde{\rho}, \qquad \tilde{\epsilon} \to e^{\theta_1}\tilde{\epsilon} - \frac{1}{2}e^{\theta_1}\tilde{\ell}^\mu\partial_\mu(\theta_1 + i\theta_2), \qquad \tilde{\pi} \to e^{-i\theta_2}\tilde{\pi}. \tag{3.5}$$

Then, we perform a rotation of class I, defined with a complex function $a$ by

$$\tilde{\ell} \to \tilde{\ell}, \qquad \tilde{n} \to \tilde{n} + a\bar{a}\tilde{\ell} + \bar{a}\tilde{m} + a\bar{\tilde{m}}, \qquad \tilde{m} \to \tilde{m} + a\tilde{\ell}, \tag{3.6}$$

and transforming the spin coefficients as

$$\tilde{\kappa} \to \tilde{\kappa}, \qquad \tilde{\rho} \to \tilde{\rho} + \bar{a}\tilde{\kappa}, \qquad \tilde{\epsilon} \to \tilde{\epsilon} + \bar{a}\tilde{\kappa}, \qquad \tilde{\pi} \to \tilde{\pi} + 2\bar{a}\tilde{\epsilon} + \bar{a}^2\tilde{\kappa} - \tilde{\ell}^\mu\partial_\mu\bar{a}. \tag{3.7}$$

By combining these class I and class III transformations we obtain the new null vectors

$$\ell := e^{\theta_1}\tilde{\ell}, \qquad n := e^{-\theta_1}\tilde{n} + a\bar{a}e^{\theta_1}\tilde{\ell} + \bar{a}e^{i\theta_2}\tilde{m} + ae^{-i\theta_2}\bar{\tilde{m}}, \qquad m := e^{i\theta_2}\tilde{m} + ae^{\theta_1}\tilde{\ell}. \tag{3.8}$$

Finally, by choosing the parameters of the Lorentz transformations to be given in terms of the old spin coefficients by

$$\theta_1 = 2\int^r \text{Re}(\tilde{\epsilon}) = -2B, \qquad \theta_2 = 2\int^r \text{Im}(\tilde{\epsilon}), \qquad \bar{a} = \int^r e^{-(\theta_1+i\theta_2)}\tilde{\pi}, \tag{3.9}$$

we get that the new spin coefficients satisfy $\kappa = \epsilon = \pi = 0$ and $\rho = \bar{\rho}$, as desired. The geometrical meaning of these conditions is that all the new null vectors are now constant along $\ell$, i.e. $\ell^\mu\nabla_\mu e_i = 0$ for $i \in \{0, 1, 2, 3\}$. The $1/r$ expansion of the new tetrad (3.8) and of the associated spin coefficients are given by appendix A, to an order relevant for the rest of our calculations. One should note in particular that the new frames $m$ are not purely tangential to the celestial sphere anymore (at the difference with $\tilde{m}$), and have a non-vanishing radial component $m^r$.

In the above construction we have obtained the new tetrad (3.8) by first defining the Bondi tetrad (3.1) and then using its non-vanishing coefficients $\tilde{\epsilon}$ and $\tilde{\pi}$ to perform two Lorentz transformations. This is very different from how one would normally proceed in the NP formalism (see e.g. [108]), where the gauge conditions $\epsilon = 0 = \pi$ can be chosen from the onset, and then a tetrad compatible with these conditions and defining the line element can be found by solving part of the NP equations. This difference of approach is due once again to the fact that we want to map the solution space of section 2 to the NP formalism, and not just solve the NP equations in isolation. In particular, one should recall that here we are defining a NP version of the BS gauge, so that $B \neq 0$ even if we have achieved $\epsilon = 0$ with our new tetrad. Because other references usually consider the NP formalism in NU gauge, this means that many subtle differences (for example in numerical factors) between our formulas and that of e.g. [108] will be scattered around. This is also the reason for which one cannot blindly use these previous results and why the dictionary between BS and NP must be reconstructed carefully. Let us then go ahead with the construction of this dictionary and now consider the Weyl scalars.

## 3.2 Expansion of the Weyl scalars

Having defined a null tetrad, we can now compute the Weyl scalars. Note that our convention for the Riemann tensor is $R^\lambda{}_{\sigma\mu\nu} = \partial_\mu \Gamma^\lambda_{\sigma\nu} - \partial_\nu \Gamma^\lambda_{\mu\sigma} + \Gamma^\lambda_{\mu\rho}\Gamma^\rho_{\sigma\nu} - \Gamma^\lambda_{\rho\nu}\Gamma^\rho_{\mu\sigma}$, and that the rest of our conventions for the NP formalism is given in appendix A. Projecting the components of the Weyl tensor we find

$$\Psi_0 := -W_{\ell m \ell m} = \frac{1}{r^5}\mathcal{E}^1_{ab}m_1^a m_1^b + \mathcal{O}(r^{-6}), \tag{3.10a}$$

$$\Psi_1 := -W_{\ell n \ell m} = \frac{1}{r^4}\mathcal{P}_a m_1^a + \mathcal{O}(r^{-5}), \tag{3.10b}$$

$$\Psi_2 := -W_{\ell m \bar{m} n} = -\frac{1}{2}\big(W_{\ell n \ell n} - W_{\ell n m \bar{m}}\big) = \frac{\mathcal{M}_\mathbb{C}}{r^3} + \mathcal{O}(r^{-4}), \tag{3.10c}$$

$$\Psi_3 := -W_{n \bar{m} n \ell} = \frac{2}{r^2}\mathcal{J}_a \bar{m}_1^a + \mathcal{O}(r^{-3}), \tag{3.10d}$$

$$\Psi_4 := -W_{n \bar{m} n \bar{m}} = \frac{2}{r}\mathcal{N}_{ab}\bar{m}_1^a \bar{m}_1^b + \mathcal{O}(r^{-2}), \tag{3.10e}$$

which exhibits the usual peeling behavior. Here

$$m_1^a = \sqrt{\frac{q_{\theta\theta}}{2q}}\left(\frac{\sqrt{q} + iq_{\theta\phi}}{q_{\theta\theta}}\delta_\theta^a - i\delta_\phi^a\right) \tag{3.11}$$

is the first term in the expansion (A.7b) of the angular frame $m^a$, and the factors of 2 in $\Psi_3$ and $\Psi_4$ have been introduced for later convenience when studying the transformation laws. All the terms in this expansion are contracted with the frames $m_1^a$ and/or $\bar{m}_1^a$, thereby inheriting a NP spin as listed in (A.4). Note also that at leading order these Weyl scalars do not depend on whether we use the tetrad (3.1) or (3.8). The new quantities which appear in these expansions of the Weyl scalars are the so-called covariant functionals [110]

$$\mathcal{E}^1_{ab} := 3E^1_{ab} - \frac{3}{16}[CC]C_{ab}, \tag{3.12a}$$

$$\mathcal{P}_a := -\frac{3}{2}N_a + \frac{3}{32}\partial_a[CC] + \frac{3}{4}C_{ab}D_cC^{bc}, \tag{3.12b}$$

$$\mathcal{M} := M + \frac{1}{16}\partial_u[CC], \tag{3.12c}$$

$$\widetilde{\mathcal{M}} := \frac{1}{8}\big(2D_aD_b - N_{ab}\big)\widetilde{C}^{ab}, \tag{3.12d}$$

$$\mathcal{J}_a := \frac{1}{8}\big(2D^bN_{ab} + \partial_aR\big), \tag{3.12e}$$

$$\mathcal{N}_{ab} := \frac{1}{4}\partial_uN_{ab}, \tag{3.12f}$$

where the complex mass is $\mathcal{M}_\mathbb{C} := \mathcal{M} + i\widetilde{\mathcal{M}}$ and the dual shear is $\widetilde{C}^{ab} := q^{ac}\varepsilon_{cd}C^{db}$. Note that this dual shear is still symmetric and trace-free.

We now want to study the subleading expansion of the Weyl scalars, and in particular of $\Psi_{0,1,2}$ since these are the quantities appearing in the time evolution of $\Psi_0$. For this, following the standard notation in the literature let us denote the expansion of the scalar $\Psi_i$ as

$$\Psi_i = \frac{\Psi_i^0}{r^{5-i}} + \frac{\Psi_i^1}{r^{6-i}} + \frac{\Psi_i^2}{r^{7-i}} + \mathcal{O}(r^{(i-8)}). \tag{3.13}$$

In (3.10) we have therefore obtained the leading terms $\Psi_i^0$. Let us next study the expansion of $\Psi_0$. This can be found either by expanding directly the definition (3.10a) or by using the NP equation

$$\Psi_0 = \big(\ell^\mu\partial_\mu + \rho + \bar{\rho} + 3\epsilon - \bar{\epsilon}\big)\sigma - \big(m^\mu\partial_\mu + \tau + 3\beta - \bar{\pi} + \bar{\alpha}\big)\kappa, \tag{3.14}$$

which here with our choice of tetrad reduces simply to

$$\Psi_0 = \left(e^{-2B}\partial_r + 2\rho\right)\sigma = \frac{1}{2}e^{-2B}\left(\partial_r + \partial_r \ln\sqrt{\gamma}\right)\left(e^{-2B}\partial_r\gamma_{ab}m^am^b\right). \tag{3.15}$$

By expanding this expression we find

$$\Psi_0^0 := \mathcal{E}_{ab}^1 m_1^a m_1^b = \left(3E_{ab}^1 - \frac{3}{16}[CC]C_{ab}\right)m_1^a m_1^b, \tag{3.16a}$$

$$\Psi_0^1 := \mathcal{E}_{ab}^2 m_1^a m_1^b = 6E_{ab}^2 m_1^a m_1^b, \tag{3.16b}$$

$$\Psi_0^2 := \mathcal{E}_{ab}^3 m_1^a m_1^b = \left(10E_{ab}^3 + \frac{5}{16}[CC]E_{ab}^1 - \frac{15}{8}[CE^1]C_{ab} + \frac{15}{128}[CC]^2C_{ab}\right)m_1^a m_1^b, \tag{3.16c}$$

$$\Psi_0^3 := \mathcal{E}_{ab}^4 m_1^a m_1^b = \left(15E_{ab}^4 + \frac{3}{4}[CC]E_{ab}^3 - 3[CE^3]C_{ab}\right)m_1^a m_1^b, \tag{3.16d}$$

$$\Psi_0^n := \mathcal{E}_{ab}^{n+1} m_1^a m_1^b = \frac{1}{2}(n+2)(n+3)E_{ab}^{n+1}m_1^a m_1^b + (\text{NL}), \tag{3.16e}$$

where we are now introducing the higher Bondi aspects $\mathcal{E}_{ab}^n$ in terms of which we have written the evolution equations in section 2.2. All the terms $\Psi_0^n$ in the expansion of $\Psi_0$ have NP spin 2.

Similarly, in order to expand $\Psi_{1,2,3,4}$ we can directly use the definitions (3.10), or instead expand the spin coefficients and the NP Bianchi identities

$$\left(\ell^\mu\partial_\mu + 4\rho + 2\epsilon\right)\Psi_1 = \left(\bar{m}^\mu\partial_\mu + 4\alpha - 1\pi\right)\Psi_0 + 3\kappa\Psi_2, \tag{3.17a}$$

$$\left(\ell^\mu\partial_\mu + 3\rho + 0\epsilon\right)\Psi_2 = \left(\bar{m}^\mu\partial_\mu + 2\alpha - 2\pi\right)\Psi_1 + 2\kappa\Psi_1 + 1\lambda\Psi_0, \tag{3.17b}$$

$$\left(\ell^\mu\partial_\mu + 2\rho - 2\epsilon\right)\Psi_3 = \left(\bar{m}^\mu\partial_\mu + 0\alpha - 3\pi\right)\Psi_2 + 1\kappa\Psi_0 + 2\lambda\Psi_1, \tag{3.17c}$$

$$\left(\ell^\mu\partial_\mu + 1\rho - 4\epsilon\right)\Psi_4 = \left(\bar{m}^\mu\partial_\mu - 2\alpha - 4\pi\right)\Psi_3 \qquad + 3\lambda\Psi_2, \tag{3.17d}$$

which can be simplified thanks to the fact that $\kappa = \epsilon = \pi = 0$ because of our choice of tetrad. Using this,

the terms which are relevant for the rest of our calculations are found to be[8]

$$\Psi_1^0 = \mathcal{P}_a m_1^a, \tag{3.18a}$$

$$\Psi_1^1 = -\overline{\eth}\Psi_0^0, \tag{3.18b}$$

$$\Psi_1^2 = -\frac{1}{2}\Big(\overline{\eth}\Psi_0^1 + \bar{\sigma}_2\eth\Psi_0^0 + 5\eth\bar{\sigma}_2\Psi_0^0\Big), \tag{3.18c}$$

$$\Psi_1^3 = \frac{5}{6}\Big(\bar{\Psi}_1^0 - 2\bar{\sigma}_2\overline{\eth}\sigma_2 - \overline{\eth}(\sigma_2\bar{\sigma}_2)\Big)\Psi_0^0 - \frac{1}{3}\bar{\sigma}_2\eth\Psi_0^1 - 2\eth\bar{\sigma}_2\Psi_0^1 - \frac{1}{3}\overline{\eth}\Psi_0^2, \tag{3.18d}$$

$$\Psi_2^0 = \mathcal{M}_{\mathbb{C}}, \tag{3.18e}$$

$$\Psi_2^1 = -\overline{\eth}\Psi_1^0, \tag{3.18f}$$

$$\Psi_2^2 = \frac{1}{2}\Big(\overline{\eth}^2\Psi_0^0 - \partial_u\bar{\sigma}_2\Psi_0^0 - \bar{\sigma}_2\eth\Psi_1^0 - 4\eth\bar{\sigma}_2\Psi_1^0\Big), \tag{3.18g}$$

$$\Psi_2^3 = \frac{2}{3}\Big(\bar{\Psi}_1^0 - 2\bar{\sigma}_2\overline{\eth}\sigma_2 - \overline{\eth}(\sigma_2\bar{\sigma}_2)\Big)\Psi_1^0 + \frac{1}{3}\bar{\sigma}_2\eth\overline{\eth}\Psi_0^0 + \frac{5}{3}\eth\bar{\sigma}_2\overline{\eth}\Psi_0^0 - \frac{1}{12}\bar{\sigma}_2 R\Psi_0^0 - \frac{1}{3}\partial_u\bar{\sigma}_2\Psi_0^1 - \frac{1}{3}\overline{\eth}\Psi_1^2, \tag{3.18h}$$

$$\Psi_3^0 = 2\mathcal{J}_a\bar{m}_1^a = -\eth\lambda_1 + \frac{1}{4}\overline{\eth}R, \tag{3.18i}$$

$$\Psi_3^1 = -\overline{\eth}\Psi_2^0, \tag{3.18j}$$

$$\Psi_4^0 = 2\mathcal{N}_{ab}\bar{m}_1^a\bar{m}_1^b = -\partial_u\lambda_1, \tag{3.18k}$$

$$\Psi_4^1 = -\overline{\eth}\Psi_3^0, \tag{3.18l}$$

where $\lambda_1 = \partial_u\bar{\sigma}_2$. Recall that in the metric formulation we have found in the previous section that the solution space is entirely determined by the free data $C_{ab}$ and the data $(M, N_a, E_{ab}^n)$ subject to evolution equations, or equivalently $(\mathcal{M}, \mathcal{P}_a, \mathcal{E}_{ab}^n)$. The same structure is of course recovered in the NP formalism, where $C_{ab}$ is encoded in $(\Psi_3^0, \Psi_4^0)$ via $\sigma_2$, and $(\mathcal{M}, \mathcal{P}_a, \mathcal{E}_{ab}^n)$ are encoded in $(\Psi_2^0, \Psi_1^0, \Psi_0^n)$. This is the reason for which all the subleading terms in $\Psi_{1,2,3,4}$ are completely determined in terms of $(\sigma_2, \Psi_4^0, \Psi_3^0, \Psi_2^0, \Psi_1^0, \Psi_0^n)$.

Recall that our goal is to access the evolution equations for the higher Bondi aspects. These are the equations for $\mathcal{E}_{ab}^n$ which we have started to list in section 2.2, and which in the NP formalism become evolution equations for $\Psi_0^n$. We are now in position to explain more precisely how $\mathcal{E}_{ab}^n$ and $\Psi_0^n$ are related to the higher spin charges. For this, one can first notice from (3.18) that for $i \geq 1$ the subleading Weyl scalars are all given by

$$\Psi_{i\geq 1}^1 = -\overline{\eth}\Psi_{i-1}^0. \tag{3.19}$$

Such a relation is possible because $\overline{\eth}$ is an operator with NP spin $-1$ and $\Psi_i$ has spin $(2-i)$, and for all $i \geq 1$ there exists an object of spin $(3-i)$ in the solution space. For $i = 0$ however this is a priori not the case since there is no object of spin 3. The idea of [89], inspired by [104], is to extend this relationship to $\Psi_0$ nevertheless by introducing higher spin charges by hand. This can be done by rewriting the expansion of $\Psi_0$ in the form

$$\Psi_0 = \frac{\Psi_0^0}{r^5} + \sum_{n=1}^{\infty}\frac{\Psi_0^n}{r^{n+5}} = \frac{\Psi_0^0}{r^5} + \sum_{s=3}^{\infty}\frac{\Psi_0^{s-2}}{r^{s+3}} = \frac{\Psi_0^0}{r^5} + \sum_{s=3}^{\infty}\frac{1}{r^{s+3}}\frac{(-1)^s}{(s-2)!}\Big(\overline{\eth}^{s-2}\mathcal{Q}_s + \Phi_0^{s-2}\Big). \tag{3.20}$$

In the expansion of $\Psi_0$ all the terms have NP spin 2, but the idea of this rewriting is to trade the spin 2 data $\Psi_0^{s-2}$ for $\Phi_0^{s-2}$ and $\overline{\eth}^{s-2}\mathcal{Q}_s$, which now features implicitly-defined spin $s$ quantities $\mathcal{Q}_s$. These are

---

[8]Note that $\sigma_2$ here is traditionally called $\sigma_0$ in other references [104, 108]. We have chosen however to label the expansion of the spin coefficients as in (A.5) because this allows to keep track of the order at which various terms appear. This notation also has the advantage of remaining consistent in case we change the tetrad and overleading terms appear.

precisely the higher spin charges whose properties we want to investigate. Importantly, we have not directly identified $\Psi_0^{s-2}$ with $\bar{\eth}^{s-2}\mathcal{Q}_s$, but allowed for possible terms $\Phi_0^{s-2}$ which may be required for the rewriting to be consistent with the evolution equations, and which we will have to identify below. In particular, this turns out to depend on the choice of tetrad, and we will see below that for the tetrad (3.8) the spin 3 is defined by $-\bar{\eth}\mathcal{Q}_3 = \Psi_0^1$, while when using the Bondi tetrad (3.1) one has instead $-\bar{\eth}\mathcal{Q}_3 = \Psi_0^1 - 4\tilde{\epsilon}_2\Psi_0^0$. Now that we have introduced the higher spin charges, we can study their evolution equations.

## 3.3 Evolution equations

The proof that the higher spin charges $\mathcal{Q}_s$ form a $w_{1+\infty}$ algebra at linear level relies on the assumption that the evolution equations for these charges can be written as the recursion relation

$$\partial_u \mathcal{Q}_s = \eth\mathcal{Q}_{s-1} - (s+1)\sigma_2\mathcal{Q}_{s-2} \tag{3.21}$$

for all $s \geq -1$ [89]. Our goal is now to investigate these evolution equations and the hypothesis under which they hold. We are going to see that they indeed genuinely take this form up to $s = 3$, as already shown by Newman and Penrose [104], but that for $s \geq 4$ a fine-tuned choice of $\Phi_0^{s-2}$ in (3.20) is required. More precisely, our analysis is going to show that for $s \geq 4$ there are a priori unwanted terms in the evolution equations, but that they can be reabsorbed by a convenient choice of $\Phi_0^{s-2}$ in (3.20) which involves anti-derivatives (i.e. non-local terms) in time. By doing so, one is actually introducing the higher spin charges $\mathcal{Q}_s$ so that they satisfy the evolution equation (3.21) *by definition*. This is possible because the field redefinition (3.20) trades $\Psi_0^{s-2}$ for $\mathcal{Q}_s$ and $\Phi_0^{s-2}$, and the latter can always be chosen such that $\partial_u\mathcal{Q}_s$ has the desired form. The physical meaning of this redefinition is however unclear and will require a separate investigation postponed to future work. Here we push the study of the evolution equations up to $s = 5$. The result for $s = 4$ is already present in [104, 108], but differs slightly from our equation (3.28) because these references use the NU gauge while here we are in BS gauge. The equation (3.32) for $s = 5$ is new.

This analysis is of course intimately related to the question of whether the $w_{1+\infty}$ symmetry algebra of higher spin charges exists in the full theory or only its self-dual truncation. Indeed, the arguments for the existence of this symmetry come so far from self-dual gravity [69, 78–80, 83, 84, 86, 92]. However, here we will obtain the $w_{1+\infty}$ algebra as an immediate consequence of the recursion relation (3.21) without the need to invoke a self-duality condition. There is nonetheless a subtle point, as mentioned above, which is that the obtention of (3.21) requires to introduce the non-local quantities $\Phi_0^{s-2}$. It is possible that these terms will simplify when imposing a self-dual condition, but a systematic understanding of such a mechanism is still missing from our analysis. Moreover, we should recall that the statements made about the evolution equations and the identification of the higher spin charges depend subtly on the choice of tetrad and the choice of gauge (e.g. BS, NU, or any other choice [50, 51]). This is precisely the origin of the mismatch of numerical factors between our equation (3.28) and the corresponding equations in [104, 108]. For $s = 4$, we will see that even when setting $\bar{\sigma}_2 = 0$ as a self-dual condition[9] in order to simplify the evolution equation, there are terms quadratic in the Weyl scalars which need to be reabsorbed with the choice of $\Phi_0^2$. However, these quadratic terms come with different numerical factors than in [104, 108] because of the different choice of gauge and tetrad. This therefore leads us to the conjecture that there exists a preferred choice of gauge and tetrad basis for which the terms $\Phi_0^{s-2}$ in (3.20) become local in time or even possibly vanish, at least in

---

[9]A further subtlety which needs to be clarified in future work is whether $\bar{\sigma}_2 = 0$, which is a single-helicity condition on the radiation, can be used as a suitable or partial self-dual condition. This is a priori not clear, since for example twistorial proofs of the $w_{1+\infty}$ algebra of self-dual gravity use a condition on the Weyl tensor written in split signature, which a priori is unrelated the condition $\bar{\sigma}_2 = 0$. To impose a self-dual condition in the NP formalism in Lorentzian signature (which is our setup), one must probably consider $\sigma_2$ and $\bar{\sigma}_2$ as independent variables. For preliminary work in this direction, see [123].

the self-dual theory. This should be checked already for $s = 4$ and will be the subject of forthcoming work. Let us now make this discussion more explicit.

**Spin $-1 \leq s \leq 2$ evolution.** The evolution equations in NP form come from the Bianchi identities

$$n^\mu \partial_\mu \Psi_3 = m^\mu \partial_\mu \Psi_4 + (1\tau - 4\beta)\Psi_4 + (4\mu + 2\gamma)\Psi_3 - 3\nu\Psi_2, \tag{3.22a}$$

$$n^\mu \partial_\mu \Psi_2 = m^\mu \partial_\mu \Psi_3 + (2\tau - 2\beta)\Psi_3 + (3\mu + 0\gamma)\Psi_2 - 2\nu\Psi_1 - 1\sigma\Psi_4, \tag{3.22b}$$

$$n^\mu \partial_\mu \Psi_1 = m^\mu \partial_\mu \Psi_2 + (3\tau - 0\beta)\Psi_2 + (2\mu - 2\gamma)\Psi_1 - 1\nu\Psi_0 - 2\sigma\Psi_3, \tag{3.22c}$$

$$n^\mu \partial_\mu \Psi_0 = m^\mu \partial_\mu \Psi_1 + (4\tau + 2\beta)\Psi_1 + (1\mu - 4\gamma)\Psi_0 \qquad - 3\sigma\Psi_2, \tag{3.22d}$$

which can be expanded using the formulas for the spin coefficients and the frames given in appendix A. At leading order we find

$$\partial_u \Psi_3^0 = \eth\Psi_4^0, \tag{3.23a}$$

$$\partial_u \Psi_2^0 = \eth\Psi_3^0 - 1\sigma_2\Psi_4^0, \tag{3.23b}$$

$$\partial_u \Psi_1^0 = \eth\Psi_2^0 - 2\sigma_2\Psi_3^0, \tag{3.23c}$$

$$\partial_u \Psi_0^0 = \eth\Psi_1^0 - 3\sigma_2\Psi_2^0, \tag{3.23d}$$

which are the evolution equations for $(\Psi_3^0, \Psi_2^0, \Psi_1^0, \Psi_0^0) = (\mathcal{Q}_{-1}, \mathcal{Q}_0, \mathcal{Q}_1, \mathcal{Q}_2)$. Using the formulas given in appendix B, one can check for consistency that these equations are equivalent to the contraction of (2.16) and (2.18) with the frames $m_1^a$ (the first equation in (2.16) must be contracted with $\bar{m}_1^a$). We can now expand (3.22d) further using (3.18) for the Weyl scalars. This will give the evolution equations for $\Psi_0^{n \geq 1}$, which we will translate into evolution equations for the $s \geq 3$ higher spin charges $\mathcal{Q}_s$.

**Spin 3 evolution.** At order $r^{-6}$ equation (3.22d) leads to

$$\partial_u \Psi_0^1 = -\bar{\eth}\left(\eth\Psi_0^0 - 4\sigma_2\Psi_1^0\right). \tag{3.24}$$

Using the relations given in appendix B, one can check that this evolution equation is equivalent to the contraction of (2.19) with $m_1^a m_1^b$. If, following (3.20), we now denote $\Psi_0^1 \equiv -\bar{\eth}\mathcal{Q}_3$, the evolution equation becomes

$$\partial_u \mathcal{Q}_3 = \eth\Psi_0^0 - 4\sigma_2\Psi_1^0, \tag{3.25}$$

which is the natural continuation of (3.23), or equivalently (3.21) for $s = 3$. If we identify a genuine rank 3 tensor through the relations

$$\mathcal{E}_{ab}^2 \equiv -D^c \mathcal{E}_{abc}^2, \qquad \mathcal{Q}_3 \equiv \mathcal{E}_{abc}^2 m_1^a m_1^a m_1^a, \qquad \Psi_0^1 = \mathcal{E}_{ab}^2 m_1^a m_1^b = -\left(D^c \mathcal{E}_{abc}^2\right) m_1^a m_1^b = -\bar{\eth}\mathcal{Q}_3, \tag{3.26}$$

the tensorial rewriting of the evolution equation for the spin 3 becomes

$$\partial_u \mathcal{E}_{abc}^2 = D_{\langle a} \mathcal{E}_{b\rangle c}^1 + 2\mathcal{P}_{\langle a} C_{b\rangle c}. \tag{3.27}$$

The contraction of this equation with $m_1^a m_1^b m_1^c$ returns as expected (3.25). One can see that the spin 3 charge $\mathcal{Q}_3$ is in fact the potential for the Weyl scalar $\Psi_0^1$.

It is important to note that the form of (3.24) and its rewriting as (3.25) depends on the choice of tetrad. In order to illustrate this point we give in appendix C a derivation of the spin 3 evolution equation using the Bondi tetrad (3.1). In this case the evolution equation can also be written as (3.25), but the relationship between $\Psi_0^1$ and $\mathcal{Q}_3$ is more complicated than above and requires to use a non-vanishing $\Phi_0^1$ in (3.20). It is precisely the fact that these evolution equations depend on the choice of tetrad which gives hope for finding another tetrad which will simplify the terms $\Phi_0^{s-2}$ for $s \geq 4$ (at least in the self-dual theory).

**Spin 4 evolution.** At order $r^{-7}$ equation (3.22d) leads to

$$
\begin{aligned}
\partial_u \Psi_0^2 = {}& -\frac{1}{2}\eth\bar{\eth}\Psi_0^1 - \frac{5}{4}R\Psi_0^1 - \frac{5}{2}\bar{\eth}^2\big(\sigma_2\Psi_0^0\big) \\
& + \frac{15}{2}\Psi_0^0\Psi_2^0 - 5\big(\Psi_1^0\big)^2 \\
& + 2\Big(\eth\big(\sigma_2\bar{\sigma}_2\big) + \sigma_2\bar{\sigma}_2\eth + 5\sigma_2\eth\bar{\sigma}_2\Big)\Psi_1^0 - \frac{1}{2}\Big(6\eth\bar{\sigma}_2\eth + \bar{\sigma}_2\eth^2 + 5\partial_u\bar{\sigma}_2\sigma_2\Big)\Psi_0^0.
\end{aligned}
\tag{3.28}
$$

Using the identities given in appendix B one can show that this equation is equivalent to the contraction of (2.20) with $m_1^a m_1^b$. This is an important consistency check as the evolution equations start to get more involved now. This evolution equation is also given in equation (4.13) of [104] and at the end of appendix D of [108][10]. One should note however the important differences in the numerical factors due to the fact that these references use the NU gauge and therefore a different tetrad than ours. Using the redefinition $\Psi_0^1 \equiv -\bar{\eth}\mathcal{Q}_3$ introduced above and the commutation relations (A.3), the first line on the right-hand side of (3.28) can be rewritten as

$$
-\frac{1}{2}\eth\bar{\eth}\Psi_0^1 - \frac{5}{4}R\Psi_0^1 - \frac{5}{2}\bar{\eth}^2\big(\sigma_2\Psi_0^0\big) = \frac{1}{2}\bar{\eth}^2\big(\eth\mathcal{Q}_3 - 5\sigma_2\Psi_0^0\big) - \frac{3}{4}\mathcal{Q}_3\bar{\eth}R.
\tag{3.29}
$$

Following the ansatz (3.20) we can then introduce the spin 4 charge via the redefinition

$$
\Psi_0^2 \equiv \frac{1}{2}\big(\bar{\eth}^2\mathcal{Q}_4 + \Phi_0^2\big).
\tag{3.30}
$$

Using the operator (6.1) to introduce the non-local quantity

$$
\begin{aligned}
\Phi_0^2 := 2\partial_u^{-1}\Big[ & \frac{15}{2}\Psi_0^0\Psi_2^0 - 5\big(\Psi_1^0\big)^2 - \frac{3}{4}\mathcal{Q}_3\bar{\eth}R \\
& + 2\Big(\eth\big(\sigma_2\bar{\sigma}_2\big) + \sigma_2\bar{\sigma}_2\eth + 5\sigma_2\eth\bar{\sigma}_2\Big)\Psi_1^0 - \frac{1}{2}\Big(6\eth\bar{\sigma}_2\eth + \bar{\sigma}_2\eth^2 + 5\partial_u\bar{\sigma}_2\sigma_2\Big)\Psi_0^0 \Big],
\end{aligned}
\tag{3.31}
$$

we finally obtain the evolution equation (3.21) for $s = 4$.

This construction relies on the definition (3.31). While this is mathematically sound, the physical meaning is less clear since the history of the shear and of certain Weyl scalars from $u' = u$ to $u' = +\infty$ is involved. More precisely, (3.31) contains three types of terms. The terms on the second line, which are both linear and quadratic in the shear, all involve $\bar{\sigma}_2$, and would therefore vanish if we were to define the self-dual condition by $\bar{\sigma}_2 = 0$. The last term on the first line contains the Ricci scalar of the celestial sphere, and would therefore vanish in the case of a sphere metric $q_{ab} = q_{ab}^\circ$ since $R[q^\circ] = 2$. Finally, the first two terms quadratic in the Weyl scalars seem to survive no matter what, and force us to consider a non-local $\Phi_0^2$ even in the self-dual theory with a round sphere metric. However, we should recall at this stage that it is precisely these terms which come with different numerical factors in [104, 108] because of the different choice of gauge and tetrad. It is therefore reasonable to expect that there is a preferred choice of gauge and tetrad for which these quadratic terms are absent from (3.28). This is an important question which will be the subject of future work.

---

[10]One should however take $\gamma^0 = 0$ since this is implied by the condition $\partial_u q_{ab} = 0$ which we are using here, and also recall that here we have labeled the terms in the expansion of the spin coefficients according to their order in $1/r$. For example $\sigma_2\big|_{\text{here}} = \sigma^0\big|_{[108]}$ and $\lambda_1\big|_{\text{here}} = \lambda^0\big|_{[108]}$.

**Spin 5 evolution.** At order $r^{-8}$ equation (3.22d) leads to

$$\partial_u \Psi_0^3 = -\frac{1}{3}\eth\overline{\eth}\Psi_0^2 - \frac{3}{2}R\Psi_0^2$$
$$+ \frac{25}{3}\Psi_1^0\overline{\eth}\Psi_0^0 + \bar{\Psi}_1^0\eth\Psi_0^0 - \frac{10}{3}\Psi_0^0\overline{\eth}\Psi_1^0 + 8\Psi_0^1\Psi_2^0$$
$$+ 4\sigma_2\Big(2\bar{\sigma}_2\overline{\eth}\sigma_2 + \overline{\eth}(\sigma_2\bar{\sigma}_2) - \bar{\Psi}_1^0\Big)\Psi_1^0 - \Big(4\sigma_2\partial_u\bar{\sigma}_2 + 3\overline{\eth}^2\sigma_2 - \eth^2\bar{\sigma}_2 + 3\overline{\eth}\sigma_2\overline{\eth} + \frac{7}{3}\eth\bar{\sigma}_2\eth + \frac{1}{3}\bar{\sigma}_2\eth^2\Big)\Psi_0^1$$
$$- \frac{5}{6}\Big(\eth\big(3\bar{\sigma}_2\overline{\eth}\sigma_2 + \sigma_2\overline{\eth}\bar{\sigma}_2\big) + 18\overline{\eth}\sigma_2\eth\bar{\sigma}_2 + \frac{28}{5}\bar{\sigma}_2\overline{\eth}\sigma_2\eth + \overline{\eth}(\sigma_2\bar{\sigma}_2)\eth$$
$$+ 12\sigma_2\eth\bar{\sigma}_2\overline{\eth} + 3\eth(\sigma_2\bar{\sigma}_2)\overline{\eth} + \frac{9}{5}\sigma_2\bar{\sigma}_2\overline{\eth}\eth - \frac{3}{2}\sigma_2\bar{\sigma}_2 R\Big)\Psi_0^0. \tag{3.32}$$

This is the NP version of the evolution equation (2.21), with all the non-linear terms involving the shear appearing explicitly. As a consistency check, one can verify that the first two lines are equivalent to the contraction of (2.21) (up to its implicit shear terms) with $m_1^a m_1^b$. Using the redefinition (3.30) and the commutation relations (A.3), the first line can be rewritten as

$$-\frac{1}{3}\eth\overline{\eth}\Psi_0^2 - \frac{3}{2}R\Psi_0^2 = -\frac{1}{6}\overline{\eth}^3\big(\eth\mathcal{Q}_4 - 6\sigma_2\mathcal{Q}_3\big) - \overline{\eth}^3\big(\sigma_2\mathcal{Q}_3\big) + \frac{1}{6}\Big(\frac{3}{2}\overline{\eth}R\overline{\eth}\mathcal{Q}_4 + 2\overline{\eth}^2 R\mathcal{Q}_4\Big) - \frac{1}{6}\eth\overline{\eth}\Phi_0^2. \tag{3.33}$$

Denoting

$$\Psi_0^3 \equiv -\frac{1}{6}\big(\overline{\eth}^3\mathcal{Q}_5 + \Phi_0^3\big), \tag{3.34}$$

one can then write $\Phi_0^3$ as a complicated integral over $u$ which reabsorbs all the unwanted terms and leads to the evolution equation $\partial_u\mathcal{Q}_5 = \eth\mathcal{Q}_4 - 6\sigma_2\mathcal{Q}_3$.

While this construction enables to obtain the desired evolution equation for the spin 5 charge, it requires to introduce a complicated non-local function $\Psi_0^3$. This function contains three types of terms, namely terms which vanish when $\bar{\sigma}_2 = 0$, terms which vanish when $R = R[q^\circ] = 2$, and terms which do not vanish in either case (coming e.g. from the second line in (3.32)). As in the case of the spin 4 charge, one should investigate whether there exists a preferred choice of gauge and tetrad for which some of these terms can simplify or possibly vanish. This could allow to clarify the potential role played by the self-dual condition in the study of these higher spin charges.

**Spin $s$ evolution at linear level.** For the sake of completeness we can write the general form of the evolution equations in the linearized theory. Using (2.22), (3.16e), and (A.3) we obtain

$$\partial_u\Psi_0^{n\geq 1} = -\frac{1}{n}\eth\overline{\eth}\Psi_0^{n-1} - \frac{1}{4}(n+3)R\Psi_0^{n-1} + (\text{NL})$$
$$= \frac{(-1)^n}{n!}\eth\overline{\eth}^n\mathcal{Q}_{n+1} + \frac{(-1)^n}{4}\frac{n+3}{(n-1)!}R\overline{\eth}^{n-1}\mathcal{Q}_{n+1} + (\text{NL}')$$
$$= \frac{(-1)^n}{n!}\overline{\eth}^n\eth\mathcal{Q}_{n+1} + (\overline{\eth}R\text{ terms}) + (\text{NL}'), \tag{3.35}$$

where the contributions from the terms $\Phi_0^{n-1}$ in (3.20) have been put in the non-linear terms, as denoted by the change from (NL) to (NL'). Using (3.20) one last time we finally arrive at

$$\overline{\eth}^{s-2}\partial_u\mathcal{Q}_s = \overline{\eth}^{s-2}\eth\mathcal{Q}_{s-1} + (\overline{\eth}R\text{ terms}) + (\text{NL}''). \tag{3.36}$$

This closes the analysis of the evolution equations. We are going to use them in the next section to infer the action of BMS symmetries on the higher spin charges.

# 4 BMSW symmetries and transformation laws

So far we have studied the subleading structure of the solution space and the evolution equations, with a particular emphasis on the map between the metric and the NP formalism. We are now going to extend this dictionary to the study of the asymptotic symmetries and their action on the solution space. For this, it will not be necessary to study the asymptotic symmetries of the first order tetrad formulation as done in [107, 108]. Instead, we will simply start from the results in the metric formulation and then map them to the NP one. This will lead to one of our main results, which is formula (4.22) for the transformation law of the higher spin charges $\mathcal{Q}_s$ under the action of BMSW transformations.

Let us recall that the BMSW vector field $\xi = \xi^u \partial_u + \xi^r \partial_r + \xi^a \partial_a$ preserving the gauge choice (which is the Bondi gauge supplemented by the differential determinant condition (2.3)) and the boundary conditions (2.2) has components given by

$$\xi^u = f, \tag{4.1a}$$

$$\xi^a = Y^a + I^a = Y^a - \int_r^\infty dr' \, e^{2B} \gamma^{ab} \partial_b f = Y^a + \frac{I_1^a}{r} + \frac{I_2^a}{r^2} + \mathcal{O}(r^{-3}), \tag{4.1b}$$

$$\xi^r = -rW + \frac{1}{2}r\big(U^a \partial_a f - D_a I^a\big) = -rW + \xi_0^r + \frac{\xi_1^r}{r} + \frac{\xi_2^r}{r^2} + \mathcal{O}(r^{-3}), \tag{4.1c}$$

with

$$\partial_r f = 0 = \partial_r Y^a, \qquad \partial_u f = W, \qquad \partial_u W = 0, \qquad \partial_u Y^a = 0. \tag{4.2}$$

The sphere vector field $Y^a(x^b)$ parametrizes the superrotations, and the temporal part can be written as $f = T + uW$ where $T(x^a)$ is a pure supertranslation and $W(x^a)$ a Weyl transformation. The terms appearing in the angular and radial parts have expansions given by

$$I_1^a = -\partial^a f, \tag{4.3}$$

$$I_2^a = \frac{1}{2}C^{ab}\partial_b f, \tag{4.4}$$

$$I_3^a = -\frac{1}{16}[CC]\partial^a f, \tag{4.5}$$

$$I_4^a = \frac{1}{4}\left(E_1^{ab} - \frac{1}{16}[CC]C^{ab}\right)\partial_b f, \tag{4.6}$$

$$I_5^a = \frac{1}{5}E_2^{ab}\partial_b f + \frac{1}{16}\left(\frac{3}{32}[CC]^2 - [CE^1]\right)\partial^a f, \tag{4.7}$$

$$\xi_0^r = \frac{1}{2}D^2 f, \tag{4.8}$$

$$\xi_{n>0}^r = \frac{1}{2}\big(U_{n+1}^a \partial_a f - D_a I_{n+1}^a\big). \tag{4.9}$$

With this information we can now compute the action of BMSW transformations on the various fields of the solution space.

## 4.1 Transformation laws

We now study the transformation laws of the various quantities parametrizing the solution space. In order to obtain these transformation laws, we will follow a simple approach which consists in computing the Lie derivative $\delta_\xi g_{\mu\nu} = \mathcal{L}_\xi g_{\mu\nu}$ of the on-shell spacetime metric under the BMSW vector field $\xi$, and then reading off the various transformation laws. In particular, this bypasses any reference to the so-called anomaly

operator $\Delta_\xi$ defined in [124] and studied in [49, 125–128]. It should also be noted that here we are not using a fixed background structure nor a boundary Lagrangian, so there are no anomalies coming from these structures.

First, for the transformation of the full angular metric and its determinant we find by computing $\mathcal{L}_\xi g_{ab}$ the results

$$\delta_\xi \gamma_{ab} = \big(f\partial_u + \mathcal{L}_Y + \mathcal{L}_I\big)\gamma_{ab} + \xi^r\partial_r\gamma_{ab} - 2\gamma_{(ac}U^c\partial_{b)}f, \tag{4.10a}$$

$$\delta_\xi \ln\gamma = f\partial_u\ln\gamma + \frac{4}{r}\xi^r + 2D_a\xi^a - 2U^a\partial_a f. \tag{4.10b}$$

By expanding (4.10a) we can then extract the transformations of the various terms in the radial expansion (2.4). For the first terms we find

$$\delta_\xi q_{ab} = \big(f\partial_u + \mathcal{L}_Y - 2W\big)q_{ab}, \tag{4.11a}$$

$$\delta_\xi C_{ab} = \big(f\partial_u + \mathcal{L}_Y - W\big)C_{ab}$$
$$+ \big(\mathcal{L}_{I_1} + 2\xi_0^r\big)q_{ab}, \tag{4.11b}$$

$$\delta_\xi \boldsymbol{D}_{ab} = \big(f\partial_u + \mathcal{L}_Y\big)\boldsymbol{D}_{ab} - U_{(a}^2\partial_{b)}f$$
$$+ \big(\mathcal{L}_{I_2} + 2\xi_1^r\big)q_{ab} + \big(\mathcal{L}_{I_1} + \xi_0^r\big)C_{ab}, \tag{4.11c}$$

$$\delta_\xi \boldsymbol{E}_{ab}^1 = \big(f\partial_u + \mathcal{L}_Y + W\big)\boldsymbol{E}_{ab}^1 - \Big(U_{(a}^3 + C_{(ac}U_2^c\Big)\partial_{b)}f$$
$$+ \big(\mathcal{L}_{I_3} + 2\xi_2^r\big)q_{ab} + \big(\mathcal{L}_{I_2} + \xi_1^r\big)C_{ab} + \mathcal{L}_{I_1}\boldsymbol{D}_{ab}, \tag{4.11d}$$

$$\delta_\xi \boldsymbol{E}_{ab}^2 = \big(f\partial_u + \mathcal{L}_Y + 2W\big)\boldsymbol{E}_{ab}^2 - \Big(U_{(a}^4 + C_{(ac}U_3^c + \boldsymbol{D}_{(ac}U_2^c\Big)\partial_{b)}f$$
$$+ \big(\mathcal{L}_{I_4} + 2\xi_3^r\big)q_{ab} + \big(\mathcal{L}_{I_3} + \xi_2^r\big)C_{ab} + \mathcal{L}_{I_2}\boldsymbol{D}_{ab} + \big(\mathcal{L}_{I_1} - \xi_0^r\big)\boldsymbol{E}_{ab}^1, \tag{4.11e}$$

$$\delta_\xi \boldsymbol{E}_{ab}^3 = \big(f\partial_u + \mathcal{L}_Y + 3W\big)\boldsymbol{E}_{ab}^3 - \Big(U_{(a}^5 + C_{(ac}U_4^c + \boldsymbol{D}_{(ac}U_3^c + \boldsymbol{E}_{(ac}^1 U_2^c\Big)\partial_{b)}f$$
$$+ \big(\mathcal{L}_{I_5} + 2\xi_4^r\big)q_{ab} + \big(\mathcal{L}_{I_4} + \xi_3^r\big)C_{ab} + \mathcal{L}_{I_3}\boldsymbol{D}_{ab} + \big(\mathcal{L}_{I_2} - \xi_1^r\big)\boldsymbol{E}_{ab}^1 + \big(\mathcal{L}_{I_1} - 2\xi_0^r\big)\boldsymbol{E}_{ab}^2, \tag{4.11f}$$

where $\mathcal{L}_Y C_{ab} = Y^c\partial_c C_{ab} + 2C_{(ac}\partial_{b)}Y^c$ and $\mathcal{L}_Y C^{ab} = Y^c\partial_c C^{ab} - 2C^{(ac}\partial_c Y^{b)}$. The emerging pattern can then easily be continued to write down the transformation law for an arbitrary $\boldsymbol{E}_{ab}^n$. We are actually interested in the transformation laws for the trace-free parts $E_{ab}^n$ instead of $\boldsymbol{E}_{ab}^n$. These are given by

$$\delta_\xi E_{ab}^n = \delta_\xi \boldsymbol{E}_{ab}^n - \frac{1}{2}\big(\delta_\xi q_{ab}\boldsymbol{E}^n + q_{ab}\delta_\xi \boldsymbol{E}^n\big). \tag{4.12}$$

In particular, since $D_{ab} = 0$ we have $\boldsymbol{E}_{ab}^1 = E_{ab}^1$ and therefore $\delta_\xi \boldsymbol{E}_{ab}^1 = \delta_\xi E_{ab}^1$. Next, we also have the transformation laws

$$\delta_\xi \ln\sqrt{q} = D_a Y^a - 2W, \tag{4.13a}$$

$$\delta_\xi C_{ab} = \big(f\partial_u + \mathcal{L}_Y - W\big)C_{ab} - 2D_{\langle a}\partial_{b\rangle}f, \tag{4.13b}$$

$$\delta_\xi N_{ab} = \big(f\partial_u + \mathcal{L}_Y\big)N_{ab} - 2D_{\langle a}\partial_{b\rangle}W, \tag{4.13c}$$

$$\delta_\xi M = \big(f\partial_u + \mathcal{L}_Y + 3W\big)M + 2\mathcal{J}^a\partial_a f + \frac{1}{4}\partial_u\big(C^{ab}D_a\partial_b f\big), \tag{4.13d}$$

$$\delta_\xi N_a = \big(f\partial_u + \mathcal{L}_Y + 2W\big)N_a + \frac{1}{8}\big([CN] - [CC]\partial_u - 16M\big)\partial_a f + \frac{1}{2}\big(D_b D^c C_{ac} - D_a D^c C_{bc}\big)\partial^b f$$
$$- \frac{1}{4}\partial_a\big(C^{ab}D_a\partial_b f\big) - D_{\langle a}\partial_{b\rangle}f D_c C^{bc} - \frac{1}{2}C_{ab}\big(R\partial^b f + \partial^b D^2 f\big), \tag{4.13e}$$

where $\mathcal{L}_Y N_a = Y^b\partial_b N_a + N_b\partial_a Y^b$. It is then convenient to rewrite these transformation laws in terms of

the covariant functionals introduced in (3.12). Recalling the definition $\mathcal{M}_{ab} = \mathcal{M}q_{ab} + \widetilde{\mathcal{M}}\varepsilon_{ab}$, we get[11]

$$\delta_\xi \mathcal{N}^{ab} = \left(f\partial_u + \pounds_Y + 5W\right)\mathcal{N}^{ab}, \tag{4.14a}$$

$$\delta_\xi \mathcal{J}^a = \left(f\partial_u + \pounds_Y + 4W\right)\mathcal{J}^a + 1\mathcal{N}^{ab}\partial_b f, \tag{4.14b}$$

$$\delta_\xi \mathcal{M} = \left(f\partial_u + \pounds_Y + 3W\right)\mathcal{M} + 2\mathcal{J}^a\partial_a f, \tag{4.14c}$$

$$\delta_\xi \mathcal{P}_a = \left(f\partial_u + \pounds_Y + 2W\right)\mathcal{P}_a + 3\mathcal{M}_{ab}\partial^b f, \tag{4.14d}$$

$$\delta_\xi \mathcal{E}^1_{ab} = \left(f\partial_u + \pounds_Y + 1W\right)\mathcal{E}^1_{ab} + 4\mathcal{P}_{\langle a}\partial_{b\rangle} f, \tag{4.14e}$$

$$\delta_\xi \mathcal{E}^2_{ab} = \left(f\partial_u + \pounds_Y + 2W\right)\mathcal{E}^2_{ab} + \partial^c f\left(4D_{\langle a}\mathcal{E}^1_{b\rangle c} - 5D_c\mathcal{E}^1_{ab}\right) - \frac{5}{2}D^2 f\mathcal{E}^1_{ab} + 2\mathcal{P}^c\left(C_{\langle ac}\partial_{b\rangle}f - C_{ab}\partial_c f\right), \tag{4.14f}$$

$$\delta_\xi \mathcal{E}^n_{ab} = \left(f\partial_u + \pounds_Y + nW\right)\mathcal{E}^n_{ab} + \text{(inhomogeneous terms)}. \tag{4.14g}$$

The general form of these transformation laws is

$$\delta_\xi F_{ab\ldots} = \left(f\partial_u + \pounds_Y + (\Delta - j)W\right)F_{ab\ldots} + \text{(inhomogeneous terms)}, \tag{4.15}$$

where $j$ is the 2-dimensional spin (which is different from the NP spin $s$) and $\Delta$ the conformal (or scale) weight [50, 110]. These labels are given by

|   | $\Psi_0^0$ | $\Psi_1^0$ | $\Psi_2^0$ | $\Psi_3^0$ | $\Psi_4^0$ | $\partial_u$ | $D_a$ | $q_{ab}$ | $C_{ab}$ | $N_{ab}$ | $E^n_{ab}$ | $F^{b_1\ldots b_q}_{a_1\ldots a_p}$ | $\mathcal{M}$ | $\mathcal{P}_a$ | $m_1^a$ |
|---|---|---|---|---|---|---|---|---|---|---|---|---|---|---|---|
| $j$ | 0 | 0 | 0 | 0 | 0 | 0 | 1 | 2 | 2 | 2 | 2 | $p-q$ | 0 | 1 | $-1$ |
| $\Delta$ | 3 | 3 | 3 | 3 | 3 | 1 | 1 | 0 | 1 | 2 | $2+n$ | ? | 3 | 3 | 0 |

(4.16)

We can see in (4.14) that the simple pattern for the transformation laws stops at $\mathcal{E}^2_{ab}$, whose transformation contains many inhomogeneous terms. In the NP formalism it will be easy to see that these terms are reabsorbed when trading $\mathcal{E}^2_{ab}$ for the spin 3 charge $\mathcal{Q}_3$.

In order to obtain the transformation laws for the various NP scalars, we now introduce the spin 1 quantity[12] $\mathcal{Y} := Y_a m_1^a$, its complex conjugate, and the notation $\omega := -W - \psi$ with $-2\psi := D_a Y^a = \eth\bar{\mathcal{Y}} + \bar{\eth}\mathcal{Y}$. With this we can compute that the leading angular frames transform as

$$\delta_\xi m_1^a = \left(\frac{1}{2}(\bar{\eth}\bar{\mathcal{Y}} - \eth\mathcal{Y}) - \omega\right)m_1^a - \eth\mathcal{Y}\bar{m}_1^a, \qquad \delta_\xi m_a^1 = \left(\frac{1}{2}(\bar{\eth}\bar{\mathcal{Y}} - \eth\mathcal{Y}) + \omega\right)m_a^1 + \eth\mathcal{Y}\bar{m}_a^1. \tag{4.17}$$

Using this together with (B.35), we then obtain

$$\delta_\xi \sigma_2 = \left(f\partial_u + \mathcal{L}_\mathcal{Y} + 1W\right)\sigma_2 + \eth^2 f, \tag{4.18a}$$

$$\delta_\xi \bar{\lambda}_1 = \left(f\partial_u + \mathcal{L}_\mathcal{Y} + 2W\right)\bar{\lambda}_1 + \eth^2 W, \tag{4.18b}$$

$$\delta_\xi \Psi_4^0 = \left(f\partial_u + \mathcal{L}_\mathcal{Y} + 3W\right)\Psi_4^0, \tag{4.18c}$$

$$\delta_\xi \Psi_3^0 = \left(f\partial_u + \mathcal{L}_\mathcal{Y} + 3W\right)\Psi_3^0 + 1\eth f\Psi_4^0, \tag{4.18d}$$

$$\delta_\xi \Psi_2^0 = \left(f\partial_u + \mathcal{L}_\mathcal{Y} + 3W\right)\Psi_2^0 + 2\eth f\Psi_3^0, \tag{4.18e}$$

$$\delta_\xi \Psi_1^0 = \left(f\partial_u + \mathcal{L}_\mathcal{Y} + 3W\right)\Psi_1^0 + 3\eth f\Psi_2^0, \tag{4.18f}$$

$$\delta_\xi \Psi_0^0 = \left(f\partial_u + \mathcal{L}_\mathcal{Y} + 3W\right)\Psi_0^0 + 4\eth f\Psi_1^0, \tag{4.18g}$$

$$\delta_\xi \Psi_0^1 = \left(f\partial_u + \mathcal{L}_\mathcal{Y} + 4W\right)\Psi_0^1 - 5\eth f\bar{\eth}\Psi_0^0 - 5\Psi_0^0\bar{\eth}\eth f - \eth\Psi_0^0\bar{\eth}f + 4\sigma_2\Psi_1^0\bar{\eth}f, \tag{4.18h}$$

$$\delta_\xi \Psi_0^n = \left(f\partial_u + \mathcal{L}_\mathcal{Y} + (n+3)W\right)\Psi_0^n + \text{(inhomogeneous terms)}, \tag{4.18i}$$

---

[11]Note the mismatch of numerical factor between (4.14c) and (43) of [110], which can be traced back to the two factors of 2 which we have introduced in (3.10). We have chosen these factors in order to get the elegant numerical pattern in (4.14).

[12]Note that in [108] the spins of $\mathcal{Y}$ and $\bar{\mathcal{Y}}$ are opposite to ours, so $\mathcal{Y}|_{\text{here}} = \bar{\mathcal{Y}}|_{[108]}$.

where we have introduced the spin-weighted Lie derivative acting on a generic functional $\mathcal{F}_s$ of spin $s$ as

$$\mathcal{L}_\mathcal{Y} \mathcal{F}_s \coloneqq \left( \mathcal{Y}\bar{\eth} + \bar{\mathcal{Y}}\eth - \frac{s}{2}(\eth + \bar{\eth})(\mathcal{Y} - \bar{\mathcal{Y}}) \right) \mathcal{F}_s. \tag{4.19}$$

Note that this is not the same Lie derivative as the one appearing in the transformation laws (4.14). As mentioned above, one can see that the apparent pattern in these NP transformation laws stops at (4.18h). However, by using the commutation relations (E.7) and (E.9) along with the definition $\Psi_0^1 \equiv -\bar{\eth}\mathcal{Q}_3$ of the spin 3 charge, one can show from the transformation law (4.18h) that $\mathcal{Q}_3$ transforms as

$$\delta_\xi \mathcal{Q}_3 = \left( f\partial_u + \mathcal{L}_\mathcal{Y} + 3W \right)\mathcal{Q}_3 + 5\eth f \Psi_0^0. \tag{4.20}$$

This is now a continuation of the pattern of transformation laws obtained in (4.18) for the charges of spin $-2 \leq s \leq 2$, which strongly suggests that the higher spin charges $\mathcal{Q}_s$ could all transform in a similar fashion. We are now going to show that this is indeed the case, and that this result follows from the form of the evolution equations.

## 4.2 Action on the higher spin charges

In the previous section we have introduced the higher spin charges $\mathcal{Q}_s$ so that they satisfy the evolution equation (3.21) by definition. This requires to introduce a specific $\Phi_0^{s-2}$ at each order in the expansion (3.20). The transformation law (4.18i), if known, can then in principle be used in (3.20) to obtain the transformation law for the corresponding $\mathcal{Q}_s$. This is precisely how we have obtained (4.20). However, since these calculations become extremely tedious (see for example (4.25) of [108] for the transformation of $\Psi_0^2$), a much more systematic and efficient way to proceed is to use the fact that the transformation of a generic $\mathcal{Q}_s$ must be compatible with the evolution equation (3.21). This translates into the requirement that $[\partial_u, \delta_\xi]\mathcal{Q}_s = 0$. By studying the pattern in (4.18) and (4.20) one can easily find the ansatz for the transformation law. This leads us to consider

$$\partial_u \mathcal{Q}_s = \eth\mathcal{Q}_{s-1} - (s+1)\sigma_2\mathcal{Q}_{s-2} \tag{4.21}$$

$$\delta_\xi \mathcal{Q}_s = \left( f\partial_u + \mathcal{L}_\mathcal{Y} + 3W \right)\mathcal{Q}_s + (s+2)\eth f \mathcal{Q}_{s-1} \tag{4.22}$$

We are now going to show that these two operations intertwine each other. Such a proof is in the spirit of the "gravity from symmetry" approach studied in [110], and it ties the dynamics of the theory with its symmetry properties. Note that $\delta_\xi$ only gives the action of supertranslations (i.e. spin 0), superrotations (i.e. spin 1), and Weyl transformations on the higher spin charges $\mathcal{Q}_s$. The action of the higher spin charges on themselves will be the subject of section 6. Importantly, one should recall that (4.21) is, strictly speaking, only proven for $-1 \leq s \leq 3$. As explained in section 3.3 above, for $s \geq 4$ one can argue that quantities $\mathcal{Q}_{s\geq 4}$ satisfying (4.21) should exist, but the definition of these quantities relies on redefinitions using non-local objects $\Phi_0^{s-2}$ as in (3.30) and (3.31). In the present section, we *assume* that there is an infinite set of higher spin charges satisfying (4.21), and show that if that it the case then these quantities transform under supertranslations ans superrotations as in (4.22).

The proof of the identity $[\partial_u, \delta_\xi]\mathcal{Q}_s = 0$ requires two steps. First, we show by a direct computation detailed in appendix D that the commutator is given by

$$\left[ \partial_u, \delta_\xi \right]\mathcal{Q}_s = \left[ \mathcal{L}_\mathcal{Y}, \eth \right]\mathcal{Q}_{s-1} - \psi\eth\mathcal{Q}_{s-1} - (s-1)\eth\psi\mathcal{Q}_{s-1}, \tag{4.23}$$

where $-2\psi = D_a Y^a = \eth\bar{\mathcal{Y}} + \bar{\eth}\mathcal{Y}$. Then, we show in appendix E the identity (E.9), which in turn implies the vanishing of the commutator and the announced result. Note that we prove (E.9) in the special case of a

boundary metric of the form $q_{ab} = e^\phi q^\circ_{ab}$ for the sake of simplicity, although the result can be shown to hold also for an arbitrary $q_{ab}$ [129].

Anticipating the results of section 6, we can now already use the transformation law (4.22) to obtain the linearized $w_{1+\infty}$ bracket (1.1) of the spin 0 and spin 1 charges acting on an arbitrary $\mathcal{Q}_s$. For this let us freeze the Weyl transformations and set $W = (\eth\bar{\mathcal{Y}} + \bar{\eth}\mathcal{Y})/2$. Following section 2.3 we first need to construct charges which are conserved in the non-radiative vacuum. In NP language the vacuum conditions $\mathcal{J}^a \stackrel{\text{vac}}{=} 0 \stackrel{\text{vac}}{=} \mathcal{N}^{ab}$ translate to $\mathcal{Q}_{-1} \stackrel{\text{vac}}{=} 0 \stackrel{\text{vac}}{=} \mathcal{Q}_{-2}$. The conserved spin 0 and spin 1 charges are then given by $q_0 := \mathcal{Q}_0$ itself and $q_1 := \mathcal{Q}_1 - u\eth\mathcal{Q}_0$, which is the contraction of (2.23a) with $m_1^a$. Let us then use the equation of motion (4.21) to write (4.22) explicitly as

$$
\begin{aligned}
\delta_\xi \mathcal{Q}_s &= \big(f\partial_u + \mathcal{L}_{\mathcal{Y}} + 3W\big)\mathcal{Q}_s + (s+2)\eth f \mathcal{Q}_{s-1}\\
&= (T + uW)\big(\eth\mathcal{Q}_{s-1} - (s+1)\sigma_2\mathcal{Q}_{s-2}\big) + \mathcal{L}_{\mathcal{Y}}\mathcal{Q}_s + 3W\mathcal{Q}_s + (s+2)(\eth T + u\eth W)\mathcal{Q}_{s-1}. \quad (4.24)
\end{aligned}
$$

Keeping only the linear terms, the action of a supertranslation is

$$
\big(\delta_T \mathcal{Q}_s\big)^{(1)} = T\eth\mathcal{Q}_{s-1} + (s+2)\eth T\mathcal{Q}_{s-1}, \quad (4.25)
$$

where the superscript is meant to denote the linearized action, as in (1.1). Using $W = (\eth\bar{\mathcal{Y}} + \bar{\eth}\mathcal{Y})/2$, the action of a superrotation with e.g. $\bar{\mathcal{Y}} \neq 0$ and $\mathcal{Y} = 0$ is

$$
\begin{aligned}
\big(\delta_{\bar{\mathcal{Y}}} \mathcal{Q}_s\big)^{(1)} &= \mathcal{L}_{\bar{\mathcal{Y}}}\mathcal{Q}_s + 3W\mathcal{Q}_s + u\big(\delta_{(T=\eth\bar{\mathcal{Y}}/2)}\mathcal{Q}_s\big)^{(1)}\\
&= \Big(\bar{\mathcal{Y}}\eth + \frac{s+3}{2}\eth\bar{\mathcal{Y}}\Big)\mathcal{Q}_s + u\big(\delta_{(T=\eth\bar{\mathcal{Y}}/2)}\mathcal{Q}_s\big)^{(1)}. \quad (4.26)
\end{aligned}
$$

Next, let us consider the smeared charges

$$
\begin{aligned}
Q_0(T) &:= \oint T(z)q_0(z) = \mathcal{Q}_0(T),\\
Q_1(\bar{\mathcal{Y}}) &:= \oint \bar{\mathcal{Y}}(z)q_1(z) = \oint \big(\bar{\mathcal{Y}}(z)\mathcal{Q}_1(z) + u\eth\bar{\mathcal{Y}}(z)\mathcal{Q}_0(z)\big) = \mathcal{Q}_1(\bar{\mathcal{Y}}) + u\mathcal{Q}_0(\eth\bar{\mathcal{Y}}), \quad (4.27a)
\end{aligned}
$$

where the integral on the celestial sphere is defined as in (5.6) and has allowed us to perform an integration by parts. Note the difference between the smeared charges $Q_1$ and $\mathcal{Q}_1$: the former is conserved in the vacuum while the latter is not. In section 6 we are going to show that $Q_1(\bar{\mathcal{Y}})$ generates the superrotations $2\delta_{\bar{\mathcal{Y}}}$ (with an important factor of 2). Combining these ingredients, when acting on a smeared charge

$$
\mathcal{Q}_s(Z) := \oint Z(z)\mathcal{Q}_s(z) \quad (4.28)
$$

we finally obtain after integrating by parts

$$
\begin{aligned}
\big\{\mathcal{Q}_0(T), \mathcal{Q}_s(Z)\big\}^{(1)} &= \big(\delta_T \mathcal{Q}_s(Z)\big)^{(1)}\\
&= \oint \Big(TZ\eth\mathcal{Q}_{s-1} + (s+2)Z\eth T\mathcal{Q}_{s-1}\Big)\\
&= -\mathcal{Q}_{s-1}\big(T\eth Z - (s+1)Z\eth T\big), \quad (4.29a)\\
\big\{\mathcal{Q}_1(\bar{\mathcal{Y}}), \mathcal{Q}_s(Z)\big\}^{(1)} &= \big\{Q_1(\bar{\mathcal{Y}}), \mathcal{Q}_s(Z)\big\}^{(1)} - u\big\{\mathcal{Q}_0(\eth\bar{\mathcal{Y}}), \mathcal{Q}_s(Z)\big\}^{(1)}\\
&= 2\big(\delta_{\bar{\mathcal{Y}}}\mathcal{Q}_s(Z)\big)^{(1)} - u\big(\delta_{(T=\eth\bar{\mathcal{Y}})}\mathcal{Q}_s(Z)\big)^{(1)}\\
&= \oint \Big(2Z\bar{\mathcal{Y}}\eth\mathcal{Q}_s + (s+3)Z\eth\bar{\mathcal{Y}}\mathcal{Q}_s\Big)\\
&= -\mathcal{Q}_s\big(2\bar{\mathcal{Y}}\eth Z - (s+1)Z\eth\bar{\mathcal{Y}}\big). \quad (4.29b)
\end{aligned}
$$

As announced, this reproduces the bracket (1.1) for $s_1 = 0, 1$ and $s_2 = s$, which is the action of the supertranslations and superrotations on the higher spin charges. Note however that here we have obtained this bracket by considering the "bare" higher spin charges $\mathcal{Q}_s$ which are not conserved in the vacuum. In section 6 we will introduce charge aspects $q_s$ and smeared charges $Q_s(Z)$ which are conserved in the vacuum to quadratic order, and give a proof of the algebra (1.1) involving these renormalized charges.

# 5 Asymptotic charges

So far we have extensively used the terminology "charge" in a loose sense to refer to various data in the solution space. In this section our aim is now to explain how the higher Bondi aspects $\mathcal{E}_{ab}^n$ and the higher spin charges $\mathcal{Q}_s$ are related to subleading asymptotic charges [101–103] and to the so-called Newman–Penrose charges [104, 111]. For the sake of simplicity and clarity we are going to consider from now on that $\delta q_{ab} = 0$. The transformation laws (4.11a) and (4.13a) then imply that $Y^a$ satisfies the conformal Killing equation[13] $D_a Y_b + D_b Y_a = (D_c Y^c) q_{ab}$ and that $W = D_a Y^a / 2$. In NP form these conditions are $\eth \mathcal{Y} = 0 = \bar{\eth} \bar{\mathcal{Y}}$ and $W = (\eth \bar{\mathcal{Y}} + \bar{\eth} \mathcal{Y}) / 2$.

## 5.1 Subleading BMS charges

We start by studying the leading and subleading asymptotic BMS charges. These can be found using covariant phase space methods [130–134], and we are going to focus on the Iyer–Wald prescription. In terms of the Komar aspect and the pre-symplectic potential given respectively by

$$K_\xi^{\mu\nu} = -\sqrt{-g}\, \nabla^{[\mu} \xi^{\nu]}, \qquad \theta^\mu = \sqrt{-g}\left( g^{\alpha\beta} \delta \Gamma_{\alpha\beta}^\mu - g^{\mu\alpha} \delta \Gamma_{\alpha\beta}^\beta \right), \tag{5.1}$$

the Iyer–Wald charge aspect at a cut of $\mathcal{I}^+$ is given by the $(ur)$ component of

$$
\begin{aligned}
\slashed{\delta} H_{\mathrm{IW}}^{\mu\nu}(\xi) &= \delta K_\xi^{\mu\nu} - K_{\delta\xi}^{\mu\nu} + \xi^{[\mu} \theta^{\nu]} \\
&= \sqrt{-g}\left[ \xi^{[\mu} \left( \nabla^{\nu]} \delta g - \nabla_\alpha \delta g^{\nu]\alpha} \right) + \xi_\alpha \nabla^{[\mu} \delta g^{\nu]\alpha} + \left( \frac{1}{2} \delta g g^{[\mu\alpha} - \delta g^{[\mu\alpha} \right) \nabla_\alpha \xi^{\nu]} \right],
\end{aligned}
\tag{5.2}
$$

where $\delta g \coloneqq g_{\mu\nu} \delta g^{\mu\nu}$. Our goal is to write the first terms in the expansion of the asymptotic charges[14]

$$\slashed{\delta} H_{\mathrm{IW}}^{ur}(\xi) = \sum_{n=0}^{\infty} \frac{\slashed{\delta} H_n}{r^n}, \qquad \slashed{\delta} H_n = \sqrt{q}\left( \delta H_n^{\mathrm{i}} + H_n^{\mathrm{f}} + H_n^{\mathrm{b}} \right), \tag{5.3}$$

where the contribution $\slashed{\delta} H_n$ at every order is decomposed as the sum of an integrable part $\delta H_n^{\mathrm{i}}$, a non-integrable flux term $H_n^{\mathrm{f}}$ (which contains variations), and a boundary term $H_n^{\mathrm{b}}$ which we will omit (since it plays no role when integrating the charge aspect against the celestial sphere). Note that the decomposition between integrable and flux parts is partly arbitrary. This expansion of the charges has already been studied in [101–103], but we give here a more straightforward presentation along with the explicit dictionary between

---

[13]More precisely, one should recall that this conformal Killing equation holds globally and admits six independent solutions, which correspond to the six Lorenz generators in $\mathrm{SL}(2, \mathbb{C})$. As in the so-called extended BMS setup [41], one may also consider local solutions with poles, so as to still have $\delta q_{ab} \neq 0$. Here we are *not* considering such local solutions, and therefore the codimension-two sphere integrals do not pick up poles.

[14]We denote the asymptotic charges by $H$ in order to avoid conflicting notation with the higher spin charges $\mathcal{Q}$ and $Q$.

the metric expressions and their NP counterpart. Up to sub-subleading order we find

$$\oint H_0 \begin{cases} H_0^{\mathrm{i}} = 2f\mathcal{M} + Y^a\left(\mathcal{P}_a - \frac{1}{16}\partial_a[CC] + \frac{1}{2}C_{ab}U_2^b\right) = f\Psi_2^0 + \bar{\mathcal{Y}}\left(\Psi_1^0 - \frac{1}{2}\eth(\sigma_2\bar{\sigma}_2) - \sigma_2\eth\bar{\sigma}_2\right) + (\mathrm{cc}), \\ H_0^{\mathrm{f}} = -\frac{1}{4}fC^{ab}\delta N_{ab} = -f\sigma_2\delta\lambda_1 + (\mathrm{cc}), \end{cases}$$

$$\oint H_1 \begin{cases} H_1^{\mathrm{i}} = -\frac{1}{2}Y^a\left(D^b\mathcal{E}_{ab}^1 - \frac{1}{16}C_{ab}\partial^b[CC] + \frac{1}{8}[CC]U_a^2\right) = -\frac{1}{2}\bar{\mathcal{Y}}\overline{\eth}\left(\Psi_0^0 + \bar{\sigma}_2\sigma_2^2\right) + (\mathrm{cc}), \\ H_1^{\mathrm{f}} = 0, \end{cases}$$

$$\oint H_2 \begin{cases} \begin{aligned} H_2^{\mathrm{i}} = & -\frac{1}{6}f\left(D^aD^b\mathcal{E}_{ab}^1 + 4\mathcal{P}_aU_2^a - \frac{3}{32}C^{ab}D_a\partial_b[CC] + 9C_{ab}U_2^aU_2^b\right) \\ & -\frac{1}{6}Y^a\left(D^b\mathcal{E}_{ab}^2 + 2\mathcal{E}_{ab}^1U_2^b + C_{ab}D_c\mathcal{E}^{bc} - \frac{1}{2}C^{bc}D_a\mathcal{E}_{bc}^1 - \frac{1}{4}\partial_a[C\mathcal{E}_1]\right), \end{aligned} \\ \begin{aligned} H_2^{\mathrm{f}} = & \frac{1}{2}f\left[U_2^a\delta\left(\frac{4}{3}\mathcal{P}_a + 4C_{ab}U_2^b - \frac{1}{8}\partial_a[CC]\right) - D_a\left(\delta C^{ab}C_{bc}U_2^c\right) + \frac{5}{16}\delta[CC]D_aU_2^a - \frac{1}{32}D^aD^b\left(C_{ab}\delta[CC]\right)\right. \\ & + \frac{1}{2}\delta C^{ab}\left(\frac{2}{3}D_a\mathcal{P}_b + \frac{1}{3}\partial_u\mathcal{E}_{ab}^1 + D_a\left(C_{bc}U_2^c\right) + U_2^cD_cC_{ab} - 2C_{ac}D^cU_b^2 - \frac{1}{16}D_a\partial_b[CC] + \frac{1}{8}[CC]N_{ab}\right) \\ & \left. -\frac{1}{4}\mathcal{M}\delta[CC] + \frac{1}{6}\delta\mathcal{E}_1^{ab}N_{ab} - \frac{1}{64}\delta[CC]\partial_u[CC]\right] - \frac{1}{12}Y^a\partial_a\left(C^{bc}\delta\mathcal{E}_{bc}^1\right), \end{aligned} \end{cases}$$

$$\oint H_n \begin{cases} H_n^{\mathrm{i}} = -\frac{1}{n(n+1)}\left(fD^aD^b\mathcal{E}_{ab}^{n-1} + Y^aD^b\mathcal{E}_{ab}^n\right) + (\mathrm{NL}) = -\frac{1}{n(n+1)}\left(f\overline{\eth}^2\Psi_0^{n-2} + \bar{\mathcal{Y}}\overline{\eth}\Psi_0^{n-1}\right) + (\mathrm{NL}) + (\mathrm{cc}), \\ H_n^{\mathrm{f}} = (\mathrm{NL}), \end{cases}$$

where for $\oint H_n$ we have only given the linearized expressions.

The first contribution $\oint H_0$ is the usual BMS asymptotic charge. The second contribution $\oint H_1$ has the particularity of vanishing identically once we integrate by parts and use the conformal Killing equation $\eth\mathcal{Y} = 0 = \overline{\eth}\bar{\mathcal{Y}}$. We have still chosen to write it before the integration by parts to illustrate (part of) the contributions which would arise in the case of boundary conditions with $\delta q_{ab} \neq 0$. Similarly, the first term with $Y^a$ in $H_2^{\mathrm{i}}$ is also vanishing after integrating by parts, but $\oint H_2$ still contains many non-vanishing terms. Finally, in the linearized theory the (sub)$^n$-leading contribution $\oint H_n$ has a non-vanishing contribution from $f$ while the term $Y^aD^b\mathcal{E}_{ab}^n$ is vanishing after integrating by parts. We explain in the next subsection how the non-vanishing supertranslation contribution is related to the Newman–Penrose charges.

One should note that although the radial expansion of the charges (5.3) at order $r^{-n}$ involves the charge aspects $fD^aD^b\mathcal{E}_{ab}^{n-1}$, the higher spin charges $\mathcal{Q}_s$ do not appear in a "canonical" manner, i.e. unless we force them to appear by using the rewriting (3.20). Since we have computed the charges arising from the asymptotic Killing vector (4.1), the only symmetry parameters are the supertranslations $T$ and the superrotations $Y^a$. Loosely speaking, in the leading asymptotic charge $T$ is paired with the mass $\mathcal{M}$ (i.e. the real part of the spin 0 charge $\mathcal{Q}_0$) while $Y^a$ is paired with the momentum $\mathcal{P}_a$. However, there is of course no independent symmetric rank 2 tensor $Z^{ab}$ which could be paired with the spin 2 charge $\mathcal{E}_{ab}^1$. By integrating by parts, one could view $D^aD^bf$ as the electric part of $Z^{ab}$, but this then lacks a magnetic part and also does not come with an independent symmetry parameter. Based on the existence of the tower of charges $\mathcal{Q}_s$ and their properties, it is however tempting to conjecture that there could be an associated tower of higher rank tensors (or scalars $Z^{a_1\ldots a_s}\bar{m}_{a_1}^1\ldots\bar{m}_{a_s}^1$ of NP spin $-s$) serving as independent symmetry parameters for the spin $s \geq 2$ charges. This could potentially arise from so-called hidden symmetries and the notion of asymptotic Killing–Stäckel or Killing–Yano tensors [135, 136] (see also [87, 88] for Killing spinors and the hidden symmetries of the anti-self-dual Einstein equations). This is an interesting direction for future work,

which is related to the question of the physical role of the higher spin symmetries. The interpretation of these symmetries given in [27, 89, 90] in terms of pseudo-vector fields could also potentially be reinterpreted in terms of Killing tensors.

## 5.2 Map to the Newman–Penrose charges

Newman and Penrose have identified in [104, 111] an infinite set of conserved quantities in the linearized theory, and 10 real exactly conserved quantities in the full non-linear theory (even in the presence of radiation). As suggested by the expansion of the Iyer–Wald charges (5.3) obtained above, these NP charges are related to subleading asymptotic BMS charges. This was already investigated in [101–103]. Here we simplify and rephrase these results since they fit naturally in our discussion of the subleading structure of the Bondi gauge. We also point out the relation to the higher spin charges $\mathcal{Q}_s$.

In order to proceed, let us consider the spin-weighted spherical harmonics $Y_{\ell,m}^s$ with $|m| \le \ell$ and $|s| \le \ell$. These have the important property of being eigenfunctions of the operator $\bar{\eth}\eth$ since[15]

$$\eth Y_{\ell,m}^s = \frac{1}{\sqrt{2}}\sqrt{(\ell-s)(\ell+s+1)}\, Y_{\ell,m}^{s+1}, \tag{5.5a}$$

$$\bar{\eth} Y_{\ell,m}^s = -\frac{1}{\sqrt{2}}\sqrt{(\ell+s)(\ell-s+1)}\, Y_{\ell,m}^{s-1}, \tag{5.5b}$$

$$\bar{\eth}\eth Y_{\ell,m}^s = -\frac{1}{2}(\ell-s)(\ell+s+1)Y_{\ell,m}^s. \tag{5.5c}$$

For simplicity, we will now denote the integrals on the celestial sphere by

$$\oint := \oint_S \sqrt{q}\, \mathrm{d}\theta\, \mathrm{d}\phi, \tag{5.6}$$

and therefore omit the measure factor. The NP charges are defined for $n \ge 0$ as

$$G_m^{n,k} := \oint \bar{Y}_{n+k+2,m}^2 \Psi_0^{n+1} \propto \oint \bar{Y}_{n+k+2,m}\bar{\eth}^2 \Psi_0^{n+1} = \frac{1}{2}\oint \bar{Y}_{n+k+2,m} D^a D^b\left(\mathcal{E}_{ab}^{n+2} - i\widetilde{\mathcal{E}}_{ab}^{n+2}\right), \tag{5.7}$$

where for the second equality we have integrated by parts and dropped the numerical factors entering the definition $\bar{Y}_{\ell,m}^s \propto \bar{\eth}^s \bar{Y}_{\ell,m}$ of the spin-weighted harmonics [104]. In the last equality we have used (B.25) and (B.26) to rewrite the complex Weyl scalar $\Psi_0^n$ in terms of its real and imaginary parts given respectively by $\mathcal{E}_{ab}^{n+1}$ and its dual $\widetilde{\mathcal{E}}_{ab}^{n+1}$. We can now show that in the linearized theory the NP charges coincide with the higher spin charges. Indeed, using (3.20) and neglecting the non-linear terms we find

$$G_m^{n,k} \approx \frac{(-1)^{n+1}}{(n+1)!}\oint \bar{Y}_{n+k+2,m}^2 \bar{\eth}^{n+1} \mathcal{Q}_{n+3} \propto \oint \bar{Y}_{n+k+2,m}^{n+3} \mathcal{Q}_{n+3}, \tag{5.8}$$

where the weak equality $\approx$ means that we have neglected non-linear terms, and $\propto$ that we have also dropped numerical factors.

We are now going to show that the charges $G_m^{n,0}$ are conserved in the linearized theory, while the charges $G_m^{0,0}$ are conserved even in the full non-linear theory [104, 111]. Using the linearized field equations (3.35), we find that the time evolution is

$$\partial_u G_m^{n,k} \approx -\frac{1}{n+1}\oint \bar{Y}_{n+k+2,m}^2 \left(\bar{\eth}\eth + \frac{1}{2}n(n+5)\right)\Psi_0^n = \frac{k(k+2n+5)}{2(n+1)}G_m^{n-1,k+1}, \tag{5.9}$$

---

[15]Note that there are factors of $\sqrt{2}$ and 2 with respect to [104] due to a different normalization of the "edth" operator $\eth$.

where we have neglected non-linear terms for the first equality. Taking $k = 0$, this shows that the charges $G_m^{n,0}$ are conserved in the linearized theory. If we now focus on

$$G_m^{0,k} := \oint \bar{Y}_{k+2,m}^2 \Psi_0^1, \tag{5.10}$$

the evolution equation (3.24) can be used to find

$$\partial_u G_m^{0,k} = - \oint \bar{Y}_{k+2,m}^2 \overline{\eth}\big(\eth\Psi_0^0 - 4\sigma_2\Psi_1^0\big) = \sqrt{\frac{k(k+5)}{2}} \oint \bar{Y}_{k+2,m}^3 \big(\eth\Psi_0^0 - 4\sigma_2\Psi_1^0\big), \tag{5.11}$$

which shows that the 5 complex charges $G_m^{0,0}$ are conserved even in the full non-linear theory. One can furthermore show that $G_m^{0,0}$ is invariant under arbitrary supertranslations. Indeed, using (3.24) and (4.18h) with $f = T$ we find

$$\delta_T G_m^{0,0} = \oint \bar{Y}_{2,m}^2 \delta_T \Psi_0^1 = - \oint \bar{Y}_{2,m}^2 \overline{\eth}\Big(T\big(\eth\Psi_0^0 - 4\sigma_2\Psi_1^0\big) + 5\Psi_0^0\eth T\Big), \tag{5.12}$$

which vanishes after integrating by parts.

## 5.3 Algebra of charges and fluxes

Before moving on to the study of the $w_{1+\infty}$ algebra of higher spin charges, we conclude this section with a review of the BMS algebra of charges and fluxes in NP form. Recall that because we set $\delta q_{ab} = 0$ we have $\eth\mathcal{Y} = 0 = \overline{\eth}\bar{\mathcal{Y}}$ and $W = \big(\eth\bar{\mathcal{Y}} + \overline{\eth}\mathcal{Y}\big)/2$, and also the commutation property $[\delta_\xi, \eth] = 0$. Let us consider the leading asymptotic charge $\not{\!}Q_0$ and rename it

$$\not{\!}Q_\xi = \delta Q_\xi^{\mathrm{i}} + \Xi_\xi[\delta] = \delta\left[f\Psi_2^0 + \bar{\mathcal{Y}}\left(\Psi_1^0 - \frac{1}{2}\eth(\sigma_2\bar{\sigma}_2) - \sigma_2\eth\bar{\sigma}_2\right)\right] - f\sigma_2\delta\lambda_1 + (\mathrm{cc}), \tag{5.13}$$

where we have chosen a split between integrable part $\delta Q_\xi^{\mathrm{i}}$ and flux $\Xi_\xi[\delta]$. Using the evolution equations (3.23) one can check that the integrable part satisfies the Wald–Zoupas conservation criterion $\partial_u Q_\xi^{\mathrm{i}} \overset{\mathrm{vac}}{=} 0$ when $\partial_u\sigma_2 \overset{\mathrm{vac}}{=} 0$. The algebra of these integrable charges can then be obtained using the Barnich–Troessaert bracket [42]. Using the transformation laws (4.18), a lengthy calculation[16] leads to

$$\begin{aligned}
\big\{Q_{\xi_1}^{\mathrm{i}}, Q_{\xi_2}^{\mathrm{i}}\big\}_{\mathrm{BT}} &= \delta_{\xi_2}Q_{\xi_1}^{\mathrm{i}} + \Xi_{\xi_2}[\delta_{\xi_1}] \\
&= Q_{\xi_{12}}^{\mathrm{i}} + \Big((f_1\eth f_2 - f_2\eth f_1)\Psi_3^0 + (\mathrm{cc})\Big) + \eth(\dots),
\end{aligned} \tag{5.14}$$

where the right-hand side features the modified Lie bracket $[\xi_1, \xi_2]_* = [\xi_1, \xi_2] - \delta_{\xi_1}\xi_2 + \delta_{\xi_2}\xi_1 = \xi_{12}$ with

$$f_{12} := \mathcal{Y}_1\overline{\eth}f_2 + \bar{\mathcal{Y}}_1\eth f_2 + \frac{1}{2}f_1\big(\eth\bar{\mathcal{Y}}_2 + \overline{\eth}\mathcal{Y}_2\big) - (1 \leftrightarrow 2), \qquad \bar{\mathcal{Y}}_{12} := \bar{\mathcal{Y}}_1\eth\bar{\mathcal{Y}}_2 - (1 \leftrightarrow 2). \tag{5.15}$$

As expected, the integrable charges represent the asymptotic symmetry algebra up to a field-dependent 2-cocycle. The latter is however different from the cocycle appearing in [42, 108] because we have chosen a different split (5.13) than in these reference, and because the Barnich–Troessaert bracket is sensitive to this split. Here the cocycle involves $\Psi_3^0$ and is therefore vanishing in the radiative vacuum. Alternatively, we could have considered another prescription for the bracket, namely the Koszul bracket [137, 138], which

---

[16]The calculation uses in particular the fact that $(A - \bar{A})\Psi_2^0 + (\mathrm{cc}) = (A - \bar{A})(\Psi_2^0 - \bar{\Psi}_2^0) = A(\Psi_2^0 - \bar{\Psi}_2^0) + (\mathrm{cc})$ for an arbitrary complex function $A$, and the rewriting of $(\Psi_2^0 - \bar{\Psi}_2^0)$ as in (B.34c).

has the advantage of being independent from the split and of cancelling the field-dependency of the cocycle [51, 139].

In the next section we are going to compute the higher spin $w_{1+\infty}$ algebra using the fluxes instead of the charges. As a preliminary result, it is therefore useful to recall the computation of the BMS flux algebra along the lines of [112, 113]. For this, we define the spin 0 and spin 1 complex flux aspects

$$\mathcal{F}_0 := \int_{-\infty}^{+\infty} \mathrm{d}u\, \partial_u \mathcal{Q}_0 = \mathcal{Q}_0\Big|_{\mathcal{I}_-^+}^{\mathcal{I}_+^+}, \qquad \mathcal{F}_1 := \int_{-\infty}^{+\infty} \mathrm{d}u\, \partial_u\big(\mathcal{Q}_1 - u\eth\mathcal{Q}_0\big) = \big(\mathcal{Q}_1 - u\eth\mathcal{Q}_0\big)\Big|_{\mathcal{I}_-^+}^{\mathcal{I}_+^+}. \tag{5.16}$$

Imposing the boundary conditions $\mathcal{Q}_{-1}\big|_{\mathcal{I}_\pm^+} = 0 = \mathcal{Q}_{-2}\big|_{\mathcal{I}_\pm^+}$, meaning that radiative vacua are recovered at the corners $\mathcal{I}_\pm^+$, one can show that these flux aspects transform as

$$\delta_\xi \mathcal{F}_0 = \big(\mathcal{L}_{\mathcal{Y}} + 3W\big)\mathcal{F}_0 = \left(\mathcal{Y}\bar{\eth} + \bar{\mathcal{Y}}\eth + \frac{3}{2}\big(\eth\bar{\mathcal{Y}} + \bar{\eth}\mathcal{Y}\big)\right)\mathcal{F}_0, \tag{5.17a}$$

$$\delta_\xi \mathcal{F}_1 = \big(\mathcal{L}_{\mathcal{Y}} + 3W\big)\mathcal{F}_1 + \big(T\eth + 3\eth T\big)\mathcal{F}_0 = \big(\mathcal{Y}\bar{\eth} + \bar{\mathcal{Y}}\eth + 2\eth\bar{\mathcal{Y}} + \bar{\eth}\mathcal{Y}\big)\mathcal{F}_1 + \big(T\eth + 3\eth T\big)\mathcal{F}_0. \tag{5.17b}$$

Let us then integrate the flux aspects over the celestial 2-sphere to obtain the real-valued BMS fluxes

$$F(\xi) := \oint \Big(T\mathcal{F}_0 + T\bar{\mathcal{F}}_0 + \bar{\mathcal{Y}}\mathcal{F}_1 + \mathcal{Y}\bar{\mathcal{F}}_1\Big). \tag{5.18}$$

Defining the bracket

$$\big\{F(\xi_1), F(\xi_2)\big\} := \delta_{\xi_1} F(\xi_2), \tag{5.19}$$

we find after several integrations by parts on the sphere that

$$\big\{F(\xi_1), F(\xi_2)\big\} = -F(\xi_{12}) + \oint \big(\mathcal{F}_0 - \bar{\mathcal{F}}_0\big)\left(\bar{\mathcal{Y}}_2\eth - \mathcal{Y}_2\bar{\eth} + \frac{1}{2}\big(\bar{\eth}\mathcal{Y}_2 - \eth\bar{\mathcal{Y}}_2\big)\right)T_1, \tag{5.20}$$

with

$$T_{12} := \mathcal{Y}_1\bar{\eth}T_2 + \bar{\mathcal{Y}}_1\eth T_2 + \frac{1}{2}T_1\big(\eth\bar{\mathcal{Y}}_2 + \bar{\eth}\mathcal{Y}_2\big) - (1 \leftrightarrow 2), \qquad \bar{\mathcal{Y}}_{12} := \bar{\mathcal{Y}}_1\eth\bar{\mathcal{Y}}_2 - (1 \leftrightarrow 2). \tag{5.21}$$

This means that the fluxes represent the BMS algebra when the contribution of the dual mass

$$\mathcal{F}_0 - \bar{\mathcal{F}}_0 = 2i\widetilde{\mathcal{M}}\Big|_{\mathcal{I}_-^+}^{\mathcal{I}_+^+} \tag{5.22}$$

is vanishing at the corners of $\mathcal{I}$. This is the so-called electricity condition on the shear [112, 113].

# 6 Higher spin charges and $w_{1+\infty}$ algebra

Now that we have introduced all the background material, notations and conventions, we turn to the second main scope of this paper, which is the study of the higher spin charges and their $w_{1+\infty}$ algebra. This will follow closely the construction and the proof presented in [89], although with two important differences. First, and most importantly, our formula (6.18) for the conserved charges at quadratic level corrects formula (48) of [89] by providing the non-local terms required for conservation when $s \geq 2$. Second, we will considerably shorten the proof by working with the smeared charges instead of the local charge aspects.

## 6.1 Setup and idea of the proof

For the sake of compactness, let us denote the shear and the news by $C := \sigma_2$ and $\bar{N} := \lambda_1 = \partial_u \bar{\sigma}_2$, so that $\mathcal{Q}_{-2} = -\partial_u \bar{N}$ and $\mathcal{Q}_{-1} = -\eth \bar{N}$. With these notations, the master evolution equation for the higher spin charges becomes $\partial_u \mathcal{Q}_s = \eth \mathcal{Q}_{s-1} - (s+1)C\mathcal{Q}_{s-2}$. Following [27], let us then introduce for any function $\mathcal{F}(u,z)$ the iterated antiderivative

$$\left(\partial_u^{-n}\mathcal{F}\right)(u,z) = \int_{+\infty}^{u} \mathrm{d}u_1 \int_{+\infty}^{u_1} \mathrm{d}u_2 \cdots \int_{+\infty}^{u_{n-1}} \mathrm{d}u_n \, \mathcal{F}(u_n,z), \tag{6.1}$$

where from now on we will also use $z$ as a complex coordinate on the celestial 2-sphere. This can be used to integrate and recursively combine the evolution equations, provided we use the boundary conditions

$$\bar{N} = \mathcal{O}\left(u^{-(1+s+\epsilon)}\right), \qquad \lim_{u \to +\infty} \mathcal{Q}_s = 0, \tag{6.2}$$

where $\epsilon > 0$. Integrating the first few equations of motion then leads to

$$\mathcal{Q}_s = \partial_u^{-1}\eth\mathcal{Q}_{s-1} - (s+1)\partial_u^{-1}\left(C\mathcal{Q}_{s-2}\right), \tag{6.3a}$$

$$\mathcal{Q}_{-2} = -\partial_u \bar{N}, \tag{6.3b}$$

$$\mathcal{Q}_{-1} = -\eth \bar{N}, \tag{6.3c}$$

$$\mathcal{Q}_0 = -\eth^2 \partial_u^{-1}\bar{N} + \partial_u^{-1}\left(C\partial_u\bar{N}\right), \tag{6.3d}$$

$$\mathcal{Q}_1 = -\eth^3 \partial_u^{-2}\bar{N} + \partial_u^{-2}\eth\left(C\partial_u\bar{N}\right) + 2\partial_u^{-1}\left(C\eth\bar{N}\right), \tag{6.3e}$$

$$\mathcal{Q}_2 = -\eth^4 \partial_u^{-3}\bar{N} + \partial_u^{-3}\eth^2\left(C\partial_u\bar{N}\right) + 2\partial_u^{-2}\eth\left(C\eth\bar{N}\right) + 3\partial_u^{-1}\left(C\partial_u^{-1}\eth^2\bar{N}\right) - 3\partial_u^{-1}\left(C\partial_u^{-1}(C\partial_u\bar{N})\right). \tag{6.3f}$$

For arbitrary spin $s$, this rewriting of the charges using the iterated equations of motion will be of the form

$$\mathcal{Q}_s = \sum_{k=1}^{k_{\max}} \mathcal{Q}_s^k, \qquad k_{\max} = \left\lfloor \frac{s}{2} \right\rfloor + 2, \tag{6.4}$$

where each $\mathcal{Q}_s^k$ features exactly $(k-1)$ contributions from $C$ and 1 contribution from $\bar{N}$. We can then solve (6.3a) recursively to obtain the soft, quadratic hard, and higher order contributions[17] [90]. They are given by[18]

$$\mathcal{Q}_{s\geq-2}^1 = -\left(\partial_u^{-1}\eth\right)^{s+2}\partial_u\bar{N}, \tag{6.5a}$$

$$\mathcal{Q}_{s\geq0}^2 = \sum_{\ell=0}^{s}(\ell+1)\partial_u^{-1}(\partial_u^{-1}\eth)^{s-\ell}\left(C(\partial_u^{-1}\eth)^\ell\partial_u\bar{N}\right), \tag{6.5b}$$

$$\mathcal{Q}_{s\geq2(k-2)}^{k\geq2} = -\sum_{\ell=0}^{s}(\ell+1)\partial_u^{-1}(\partial_u^{-1}\eth)^{s-\ell}\left(C\mathcal{Q}_{\ell-2}^{k-1}\right), \tag{6.5c}$$

where we have indicated the spins at which the various orders start to appear.

---

[17]It is interesting to note that some of the higher order contributions can be interpreted as collinear corrections to the soft theorem upon computing the Fourier transform of the action on the shear. This was studied for example in section 5 of [27] in the case of the spin 2 charge and the sub-subleading soft graviton. We thank the anonymous referee for pointing this out. It would be interesting to obtain a systematic understanding of these collinear contributions.

[18]Translating to the notations of [89] we have

$$C = \sigma_2 = -\frac{1}{2}C_{ab}m_1^a m_1^b = -\frac{1}{2}C_{[89]}, \qquad \partial_u\bar{N} = \partial_u^2\bar{\sigma}_2 = -\frac{1}{2}\mathcal{N}_{[89]}, \qquad \bar{N} = N_{[89]}, \qquad \kappa_{[89]}^2 = 8.$$

With these ingredients in sight, the idea of the proof is the following. As shown by Ashtekar and Streubel [140], future null infinity comes equipped with a symplectic structure in which the shear $C$ is conjugated to the news $\bar{N}$. The $w_{1+\infty}$ algebra at linear level will essentially follow from the resulting Poisson bracket between the soft (6.5a) and quadratic hard charges (6.5b). However, as noted above the charges (6.3) are not conserved for $s \geq 1$. The brackets will therefore not be computed directly with the charges $\mathcal{Q}_s$, but instead with renormalized charges $q_s$ which are conserved (at quadratic level). In order to guess the general spin $s$ formula for these renormalized charges, it is informative to study the action of $\mathcal{Q}_{0,1,2,3}$ on the shear.

## 6.2 Action of $\mathcal{Q}_{0,1,2,3}$ on the shear

The symplectic structure on $\mathcal{I}^+$ is

$$\Omega = \frac{1}{4} \oint \int_{-\infty}^{+\infty} \mathrm{d}u \, \delta N_{ab} \delta C^{ab} = \oint \int_{-\infty}^{+\infty} \mathrm{d}u \, \delta\lambda_1 \delta\sigma_2 + (\text{cc}) = \oint \int_{-\infty}^{+\infty} \mathrm{d}u \, \delta\bar{N}\delta C + (\text{cc}). \tag{6.6}$$

This leads to the fundamental Poisson brackets

$$\left\{\bar{N}(u,z), C(u',z')\right\} = \delta(u-u')\delta(z-z'), \tag{6.7a}$$

$$\left\{\mathcal{Q}_{-2}(u,z), C(u',z')\right\} = -\partial_u\delta(u-u')\delta(z-z') = \partial_{u'}\delta(u-u')\delta(z-z'), \tag{6.7b}$$

$$\left\{\mathcal{Q}_{-1}(u,z), C(u',z')\right\} = -\delta(u-u')\eth_z\delta(z-z'). \tag{6.7c}$$

Using these brackets, the integrated equations of motion (6.3), and the identities given in appendix F, we can compute the action of the charges $\mathcal{Q}_{0,1,2,3}$ on the shear $C$. The full results including the theta functions are given in appendix G. For $u' > u$ these brackets simplify to

$$\left\{\mathcal{Q}_0(u,z), C(u',z')\right\} = \eth_z^2\delta(z-z') + N(u',z)\delta(z-z'), \tag{6.8a}$$

$$\begin{aligned}\left\{\mathcal{Q}_1(u,z), C(u',z')\right\} = &(u-u')\eth_z\left(\eth_z^2\delta(z-z') + N(u',z)\delta(z-z')\right) \\ &- \eth_z\left(C(u',z)\delta(z-z')\right) - 2C(u',z)\eth_z\delta(z-z'),\end{aligned} \tag{6.8b}$$

$$\begin{aligned}\left\{\mathcal{Q}_2(u,z), C(u',z')\right\} = &\frac{1}{2}(u-u')^2\eth_z^2\left(\eth_z^2\delta(z-z') + N(u',z)\delta(z-z')\right) \\ &- (u-u')\eth_z\left(\eth_z\left(C(u',z)\delta(z-z')\right) + 2C(u',z)\eth_z\delta(z-z')\right) \\ &+ 3C(u',z)^2\delta(z-z') + 3H(u,u',z)\left(\eth_z^2\delta(z-z') + N(u',z)\delta(z-z')\right),\end{aligned} \tag{6.8c}$$

$$\begin{aligned}\left\{\mathcal{Q}_3(u,z), C(u',z')\right\} = &\frac{1}{6}(u-u')^3\eth_z^3\left(\eth_z^2\delta(z-z') + N(u',z)\delta(z-z')\right) \\ &- \frac{1}{2}(u-u')^2\eth_z^2\left(\eth_z\left(C(u',z)\delta(z-z')\right) + 2C(u',z)\eth_z\delta(z-z')\right) \\ &+ 3(u-u')\eth_z\left(C(u',z)^2\delta(z-z')\right) \\ &+ 3\eth_z\left(\left(\partial_{u'}^{-2}C(u',z) - \partial_u^{-2}C(u,z)\right)\left(\eth_z^2\delta(z-z') + N(u',z)\delta(z-z')\right)\right) \\ &+ 3(u-u')\eth_z\left(\partial_{u'}^{-1}C(u',z)\left(\eth_z^2\delta(z-z') + N(u',z)\delta(z-z')\right)\right) \\ &- 4H(u,u',z)\left(\eth_z\left(C(u',z)\delta(z-z')\right) + 2C(u',z)\eth_z\delta(z-z')\right) \\ &- 4u'H(u,u',z)\eth_z\left(\eth_z^2\delta(z-z') + N(u',z)\delta(z-z')\right) \\ &+ 4\int_u^{u'}\mathrm{d}u'' \, u''C(u'',z)\eth_z\left(\eth_z^2\delta(z-z') + N(u',z)\delta(z-z')\right),\end{aligned} \tag{6.8d}$$

where we have introduced the non-local history of the shear

$$H(u, u', z) := \int_u^{u'} \mathrm{d}u''\, C(u'', z).$$ (6.9)

Such non-local terms start to appear in the action of the charges on the shear for $s \geq 2$.

In order to continue, we should now note that there are two (related) issues with the charges $\mathcal{Q}_{s \geq 1}$. The first one is that their action on $C$ depends on $u$ (and is therefore divergent in the limit $u \to -\infty$), and the second is that they are not conserved when $\mathcal{Q}_{-1} \overset{\text{vac}}{=} 0 \overset{\text{vac}}{=} \mathcal{Q}_{-2}$. In order to solve these issues, one can consider the renormalized charge aspects

$$\tilde{q}_0 := \mathcal{Q}_0,$$ (6.10a)

$$\tilde{q}_1 := \mathcal{Q}_1 - u\eth\mathcal{Q}_0,$$ (6.10b)

$$\tilde{q}_2 := \mathcal{Q}_2 - u\eth\mathcal{Q}_1 + \frac{u^2}{2}\eth^2\mathcal{Q}_0 + 3\big(\partial_u^{-1}C\big)\mathcal{Q}_0,$$ (6.10c)

$$\tilde{q}_3 := \mathcal{Q}_3 - u\eth\mathcal{Q}_2 + \frac{u^2}{2}\eth^2\mathcal{Q}_1 - \frac{u^3}{6}\eth^3\mathcal{Q}_0 + 4\big(\partial_u^{-1}C\big)\mathcal{Q}_1 - 4\big(\partial_u^{-2}C\big)\eth\mathcal{Q}_0 - 3\eth\big(\partial_u^{-1}(uC)\mathcal{Q}_0\big),$$ (6.10d)

which are evaluated at $(u, z)$, and where $\partial_u^{-1}C(u,z) = H(+\infty, u, z)$. Using (6.8) we can compute that for $u' > u$ these quantities act on the shear as

$$\big\{\tilde{q}_0(u,z), C(u',z')\big\} = \eth_z^2 \delta(z - z') + N(u',z)\delta(z - z'),$$ (6.11a)

$$\big\{\tilde{q}_1(u,z), C(u',z')\big\} = -u'\eth_z\big\{\tilde{q}_0(u,z), C(u',z')\big\} - \eth_z\big(C(u',z)\delta(z - z')\big) - 2C(u',z)\eth_z\delta(z - z'),$$ (6.11b)

$$\big\{\tilde{q}_2(u,z), C(u',z')\big\} = -\frac{u'^2}{2}\eth_z^2\big\{\tilde{q}_0(u,z), C(u',z')\big\} - u'\eth_z\big\{\tilde{q}_1(u,z), C(u',z')\big\}$$
$$+ 3\partial_{u'}^{-1}C(u',z)\big\{\tilde{q}_0(u,z), C(u',z')\big\} + 3C(u',z)^2\delta(z - z'),$$ (6.11c)

$$\big\{\tilde{q}_3(u,z), C(u',z')\big\} = \frac{u'^3}{3}\eth_z^3\big\{\tilde{q}_0(u,z), C(u',z')\big\} + \frac{u'^2}{2}\eth_z^2\big\{\tilde{q}_1(u,z), C(u',z')\big\}$$
$$+ \Big(\partial_{u'}^{-1}\big(u'C(u',z)\big)\eth_z - 3\partial_{u'}^{-1}\eth_z\big(u'C(u',z)\big)\Big)\big\{\tilde{q}_0(u,z), C(u',z')\big\}$$
$$+ 4\partial_{u'}^{-1}C(u',z)\big\{\tilde{q}_1(u,z), C(u',z')\big\} - 3u'\eth_z\Big(C(u',z)^2\delta(z - z')\Big),$$ (6.11d)

where in the last bracket we have used $H(u, u', z) = H(u, +\infty, z) + H(+\infty, u', z) = -\partial_u^{-1}C(u,z) + \partial_{u'}C(u',z)$ and the integral Leibniz rule

$$\partial_u^{-1}\big(uC(u,z)\big) = u\partial_u^{-1}C(u,z) - \partial_u^{-2}C(u,z).$$ (6.12)

The charge actions described by the brackets (6.11) are indeed $u$-independent and have moreover been shown in [27] to lead for $s = 0, 1, 2$ to the leading, subleading, and sub-subleading soft graviton theorems. This therefore tells us that a prescription for the conserved charges *must* reproduce in particular (6.10c) for $s = 2$.

In order to recast the action of the charges on the shear in a more familiar form, let us introduce a function $Z$ which has NP spin weight $-2$ and consider the smeared charges

$$Q_0(T; u) := \oint T(z)\tilde{q}_0(u,z), \qquad Q_1(\bar{\mathcal{Y}}; u) := \oint \bar{\mathcal{Y}}(z)\tilde{q}_1(u,z), \qquad Q_2(Z; u) := \oint Z(z)\tilde{q}_2(u,z).$$ (6.13)

We then find that these charges act on the shear as

$$\delta_T^0 C(u', z') = \{Q_0(T; u), C(u', z')\}$$
$$= TN + \eth^2 T, \tag{6.14a}$$

$$\delta_{\bar{\mathcal{Y}}}^1 C(u', z') = \{Q_1(\bar{\mathcal{Y}}; u), C(u', z')\}$$
$$= (3\eth\bar{\mathcal{Y}} + 2\bar{\mathcal{Y}}\eth)C + u'(\eth\bar{\mathcal{Y}}N + \eth^3\bar{\mathcal{Y}})$$
$$= (3\eth\bar{\mathcal{Y}} + 2\bar{\mathcal{Y}}\eth)C + u'\delta_{\eth\bar{\mathcal{Y}}}^0 C(u', z'), \tag{6.14b}$$

$$\delta_Z^2 C(u', z') = \{Q_2(Z; u), C(u', z')\}$$
$$= 3ZC^2 + 3(N + \eth^2)(Z\partial_{u'}^{-1}C) + u'(3\eth^2 Z + 2\eth Z\eth)C + \frac{u'^2}{2}(\eth^2 ZN + \eth^4 Z)$$
$$= 3ZC^2 + 3(N + \eth^2)(Z\partial_{u'}^{-1}C) + u'\delta_{\eth Z}^1 C - \frac{u'^2}{2}\delta_{\eth^2 Z}^0 C, \tag{6.14c}$$

where the results on the right-hand side are understood as evaluated at $(u', z')$. This consistently reproduces the transformation (4.18a) of the shear under supertranslations and superrotations, with $f = T + uW$ and $W = (\eth\bar{\mathcal{Y}} + \bar{\eth}\mathcal{Y})/2$, up to an overall factor of $1/2$ for the superrotations. The action of $Q_2$ can be understood in terms of spin 2 pseudo-vector fields [27, 89].

We can now also check that the charges (6.10) are conserved in the vacuum $\mathcal{Q}_{-1} \overset{\text{vac}}{=} 0 \overset{\text{vac}}{=} \mathcal{Q}_{-2}$, provided however that the shear satisfies $\lim_{u \to +\infty} C = 0$. Indeed, one can compute their evolution to find

$$\partial_u \tilde{q}_0 = \eth\mathcal{Q}_{-1} - C\mathcal{Q}_{-2}, \tag{6.15a}$$

$$\partial_u \tilde{q}_1 = -u\eth\partial_u\mathcal{Q}_0 - 2C\mathcal{Q}_{-1}, \tag{6.15b}$$

$$\partial_u \tilde{q}_2 = \frac{u^2}{2}\eth^2\partial_u\mathcal{Q}_0 + 2u\eth(C\mathcal{Q}_{-1}) + 3(\partial_u^{-1}C)\partial_u\mathcal{Q}_0, \tag{6.15c}$$

$$\partial_u \tilde{q}_3 = -\frac{u^3}{6}\eth^3\partial_u\mathcal{Q}_0 - u^2\eth^2(C\mathcal{Q}_{-1}) - 8(\partial_u^{-1}C)C\mathcal{Q}_{-1} - 4(\partial_u^{-2}C)\eth\partial_u\mathcal{Q}_0 - 3\eth(\partial_u^{-1}(uC)\partial_u\mathcal{Q}_0), \tag{6.15d}$$

where the boundary condition on the shear was used to obtain $\partial_u\partial_u^{-1}C(u, z) = \partial_u H(+\infty, u, z) = C(u, z)$. With the charges (6.10) we therefore obtain both finiteness as $u \to -\infty$ when acting on $C$ and conservation in the vacuum. The form of these charges will now help us find a general expression for arbitrary spin $s$ charges which are conserved at quadratic level.

## 6.3 Conserved charges at quadratic level

Let us start with the renormalized charges of [89], which are given by

$$\hat{q}_s := \sum_{n=0}^s \frac{(-u)^n}{n!}\eth^n\mathcal{Q}_{s-n}. \tag{6.16}$$

These charges evolve as

$$\partial_u\hat{q}_s = \frac{(-u)^s}{s!}\eth^{s+1}\mathcal{Q}_{-1} - \sum_{\ell=0}^s \frac{(-u)^{s-\ell}(\ell+1)}{(s-\ell)!}\eth^{s-\ell}(C\mathcal{Q}_{\ell-2}) \tag{6.17a}$$

$$= \frac{(-u)^s}{s!}\eth^s\partial_u\mathcal{Q}_0 - 2\frac{(-u)^{s-1}}{(s-1)!}\eth^{s-1}(C\mathcal{Q}_{-1}) - \sum_{\ell=2}^s \frac{(-u)^{s-\ell}(\ell+1)}{(s-\ell)!}\eth^{s-\ell}(C\mathcal{Q}_{\ell-2}), \tag{6.17b}$$

and therefore fail to be conserved in the vacuum $\mathcal{Q}_{-2} \overset{\text{vac}}{=} 0 \overset{\text{vac}}{=} \mathcal{Q}_{-1}$ for $s \geq 2$ because of the last term on the second line. We therefore need an alternative definition for the conserved charges. A first guess would be to

consider renormalized charges defined as in (2.25) by a sum over $\partial_u^n \mathcal{Q}_s$. While this can indeed be used to define conserved charges, one can check for $s = 2$ that this prescription differs from (6.10c), and is therefore not viable. Alternatively, one could subtract from $\hat{q}_s$ the anti-derivative $\partial_u^{-1}$ of the last term in (6.17b). This would also lead to conserved charges, but once again to the incorrect action on the shear for $s = 2$.

After studying the first few conserved charges (6.10), and also anticipating the consistency results which we will obtain below (i.e. the general formula for the action on the shear), it turns out that the correct expression for conserved charges of arbitrary spin $s$ is given to quadratic order by

$$
\tilde{q}_s := \sum_{n=0}^{s} \frac{(-u)^n}{n!} \eth^n \mathcal{Q}_{s-n} + \sum_{\ell=2}^{s} \sum_{n=0}^{\ell-2} \frac{(-1)^n (\ell+1)}{(s-\ell)!} \eth^{s-\ell} \left( \partial_u^{-(n+1)} \left( (-u)^{s-\ell} C \right) \eth^n \mathcal{Q}_{\ell-2-n} \right) + \mathcal{O}(\mathbb{F}^3) \Big|_{s \geq 4}
$$

(6.18)

The first sum corresponds to $\hat{q}_s$, and $\mathcal{O}(\mathbb{F}^3)$ denotes cubic and higher order contributions in the fields $(C, \mathcal{Q})$ (and therefore also in the fields $(C, \bar{N})$) which start to appear for spins $s \geq 4$. For example for $s = 4$ this contribution is

$$
\mathcal{O}(\mathbb{F}^3) \big|_{s=4} = \frac{15}{2} (\partial_u^{-1} C)(\partial_u^{-1} C) \mathcal{Q}_0.
$$

(6.19)

The evolution of the charges (6.18) is given by

$$
\partial_u \tilde{q}_s = \frac{(-u)^s}{s!} \eth^s \left( \eth \mathcal{Q}_{-1} - C \mathcal{Q}_{-2} \right) + \sum_{\ell=1}^{s} \frac{(-1)^\ell (\ell+1)}{(s-\ell)!} \eth^{s-\ell} \left( \partial_u^{1-\ell} \left( (-u)^{s-\ell} C \right) \eth^{\ell-1} \mathcal{Q}_{-1} \right)
$$

$$
- \sum_{\ell=2}^{s} \sum_{n=0}^{\ell-2} \frac{(-1)^n (\ell+1)}{(s-\ell)!} (\ell - 1 - n) \eth^{s-\ell} \left( \partial_u^{-(n+1)} \left( (-u)^{s-\ell} C \right) \eth^n \left( C \mathcal{Q}_{\ell-4-n} \right) \right) + \partial_u \mathcal{O}(\mathbb{F}^3) \big|_{s \geq 4}, \quad (6.20)
$$

where the whole second line is of order $\mathcal{O}(\mathbb{F}^3)$. This therefore shows that the proposal (6.18) indeed defines charges which are conserved in the radiative vacuum at quadratic order, i.e. up to corrections of order $\mathbb{F}^3$. We are now going to study the linear and quadratic components of these conserved higher spin charges, and in particular rewrite their action on the shear. This will then enable us to compute their bracket at linear level.

## 6.4 Higher spin brackets at linear level

Let us consider the renormalized higher spin charge aspects (6.18). Using (6.4), these charges can also be decomposed into soft, quadratic hard and higher order contributions as

$$
\tilde{q}_s = \sum_{k=1}^{k_{\max}} \tilde{q}_s^k.
$$

(6.21)

Using the soft and quadratic hard contributions obtained respectively from $k = 1$ and $k = 2$, our goal is to prove the bracket (1.1). This is a smeared version of the bilocal Poisson bracket of higher spin charges defined by the linearization

$$
\left\{ q_{s_1}(z), q_{s_2}(z') \right\}^{(1)} := \left\{ q_{s_1}^2(z), q_{s_2}^1(z') \right\} + \left\{ q_{s_1}^1(z), q_{s_2}^2(z') \right\},
$$

(6.22)

where $q_s(z) := \lim_{u \to -\infty} \tilde{q}_s(u, z)$. It is indeed clear from (6.5a) and (6.7) that the soft contributions commute, so that the linearized bracket reduces to the two brackets between soft and quadratic hard contributions.

In order to compute this bracket, we will now study separately the two main building blocks, which are the action of the soft and quadratic hard contributions on the shear.

Interestingly, before delving into the details of the computation, let us mention that the brackets (1.1) for $Q_{0,1,2}$ can be checked in a very direct (yet heuristic) manner by computing the linearized bracket of the smeared fluxes. This is explained in appendix H.

### 6.4.1 Soft higher spin charges

Using (6.5a) and (6.18), we find that the soft part of the renormalized higher spin charges is given by

$$\tilde{q}_s^1 = \hat{q}_s^1 = \sum_{n=0}^{s} \frac{(-u)^n}{n!} \eth^n \mathcal{Q}_{s-n}^1 = -\sum_{n=0}^{s} \frac{(-u)^n}{n!} \partial_u^{n-s-1} \eth^{s+2} \bar{N}. \tag{6.23}$$

This is therefore the same result as in [89] since there is no difference between the charges $\tilde{q}_s$ and $\hat{q}_s$ at the linear (i.e. soft) level. Taking the time derivative leads to

$$\partial_u \tilde{q}_s^1 = -\frac{(-u)^s}{s!} \eth^{s+2} \bar{N}, \tag{6.24}$$

which can be integrated again to obtain

$$\tilde{q}_s^1(u, z) = -\int_{+\infty}^{u} \mathrm{d}u' \frac{(-u')^s}{s!} \eth_z^{s+2} \bar{N}(u', z). \tag{6.25}$$

The steps leading to this result are equivalent to using the integral Leibniz rule (F.3). Finally, we can take the limit in $u$ to obtain

$$q_s^1(z) := \lim_{u \to -\infty} \tilde{q}_s^1(u, z) = \eth_z^{s+2} \bar{N}_s(z), \qquad \bar{N}_s(z) := \frac{(-1)^s}{s!} \int_{-\infty}^{+\infty} \mathrm{d}u\, u^s \bar{N}(u, z), \tag{6.26}$$

where $\bar{N}_s$ is the negative helicity (sub)$^s$-leading soft graviton operator. In Fourier modes, this coincides with the leading, subleading, and sub-subleading soft charges for $s = 0, 1, 2$ respectively [24–27, 74].

Using the bracket (6.7a), we then find that the action of the soft higher spin charges on the shear is given by

$$\left\{ q_s^1(z), C(u', z') \right\} = \frac{(-u')^s}{s!} \eth_z^{s+2} \delta(z - z'). \tag{6.27}$$

As a consistency check, one can verify that this action coincides for $s = 0, 1, 2, 3$ with the field-independent order $C^0$ terms appearing in the brackets (6.11). It should also be noted that (6.26) is ill-defined for $s = -1$, and in particular does not correspond to the (purely soft) charge $\mathcal{Q}_{-1}$. This is the reason why the bracket (1.1) excludes the case where $s_1 = 0 = s_2$, which explains why the spin interval in $w_{1+\infty}$ is $s \in [\![1, +\infty[\![$ and not just $\mathbb{N}$.

### 6.4.2 Hard higher spin charges

Using (6.5a) and (6.5b), we find that the quadratic contribution in the conserved higher spin charges (6.18) is given by

$$
\begin{aligned}
\tilde{q}_s^2 &= \sum_{n=0}^{s} \frac{(-u)^n}{n!} \eth^n \mathcal{Q}_{s-n}^2 + \sum_{\ell=2}^{s} \sum_{n=0}^{\ell-2} \frac{(-1)^n (\ell+1)}{(s-\ell)!} \eth^{s-\ell} \Big( \partial_u^{-(n+1)} \big( (-u)^{s-\ell} C \big) \eth^n \mathcal{Q}_{\ell-2-n}^1 \Big) \\
&= \sum_{n=0}^{s} \sum_{\ell=0}^{n} \frac{(-u)^{s-n}(\ell+1)}{(s-n)!} \partial_u^{\ell-n-1} \eth^{s-\ell} \big( C(\partial_u^{-1}\eth)^\ell \partial_u \bar{N} \big) \\
&\quad - \sum_{\ell=2}^{s} \sum_{n=0}^{\ell-2} \frac{(-1)^n (\ell+1)}{(s-\ell)!} \eth^{s-\ell} \Big( \partial_u^{-(n+1)} \big( (-u)^{s-\ell} C \big) \partial_u^{n-\ell+1} \eth^\ell \bar{N} \Big).
\end{aligned}
\tag{6.28}
$$

In order to compute the action of this hard term on the shear, it turns out to be much more convenient to act with the quadratic part of the evolution equation (6.20), which is

$$
\partial_u \tilde{q}_s^2 = \frac{(-u)^s}{s!} \eth^s \big( C \partial_u \bar{N} \big) - \sum_{\ell=1}^{s} \frac{(-1)^\ell (\ell+1)}{(s-\ell)!} \eth^{s-\ell} \Big( \partial_u^{1-\ell} \big( (-u)^{s-\ell} C \big) \eth^\ell \bar{N} \Big).
\tag{6.29}
$$

This enables us to obtain the key identity

$$
\begin{aligned}
\big\{ \tilde{q}_s^2(u,z), C(u',z') \big\} &\overset{(1)}{=} \partial_u^{-1} \left[ \frac{(-u)^s}{s!} \eth_z^s \Big( C(u,z) \partial_u \delta(u-u') \delta(z-z') \Big) \right] \\
&\quad - \partial_u^{-1} \left[ \sum_{\ell=1}^{s} \frac{(-1)^\ell (\ell+1)}{(s-\ell)!} \eth_z^{s-\ell} \Big( \partial_u^{1-\ell} \big( (-u)^{s-\ell} C(u,z) \big) \delta(u-u') \eth_z^\ell \delta(z-z') \Big) \right] \\
&\overset{(2)}{=} -\partial_u^{-1} \left[ \frac{(-u)^s}{s!} \eth_z^s \Big( C(u,z) \partial_{u'} \delta(u-u') \delta(z-z') \Big) \right] \\
&\quad - \partial_u^{-1} \left[ \sum_{\ell=1}^{s} \frac{(-1)^\ell (\ell+1)}{(s-\ell)!} \eth_z^{s-\ell} \Big( \partial_{u'}^{1-\ell} \big( (-u')^{s-\ell} C(u',z) \big) \delta(u-u') \eth_z^\ell \delta(z-z') \Big) \right] \\
&\overset{(3)}{=} -\partial_u^{-1} \partial_{u'} \left[ \frac{(-u')^s}{s!} \eth_z^s \Big( C(u',z) \delta(u-u') \delta(z-z') \Big) \right] \\
&\quad - \partial_u^{-1} \left[ \sum_{\ell=1}^{s} \frac{(-1)^\ell (\ell+1)}{(s-\ell)!} \eth_z^{s-\ell} \Big( \partial_{u'}^{1-\ell} \big( (-u')^{s-\ell} C(u',z) \big) \delta(u-u') \eth_z^\ell \delta(z-z') \Big) \right] \\
&\overset{(4)}{=} \partial_{u'} \left[ \frac{(-u')^s}{s!} \eth_z^s \Big( C(u',z) \theta(u'-u) \delta(z-z') \Big) \right] \\
&\quad + \sum_{\ell=1}^{s} \frac{(-1)^\ell (\ell+1)}{(s-\ell)!} \eth_z^{s-\ell} \Big( \partial_{u'}^{1-\ell} \big( (-u')^{s-\ell} C(u',z) \big) \theta(u'-u) \eth_z^\ell \delta(z-z') \Big).
\end{aligned}
\tag{6.30}
$$

In step (1) we have used the fundamental Poisson brackets (6.7) and applied the operator $\partial_u^{-1}$ to the whole bracket. In step (2) we have used the delta function identities (F.1b) and (F.1c). In step (3) we have pulled out the derivative $\partial_{u'}$ on the first line and then used (F.1c) once again. Finally, in step (4) we have used (F.1d). In the final result the $u$-dependency is only in the theta functions, which allows to easily take the limit $u \to -\infty$ in order to obtain the action of $q_s^2(z) := \lim_{u \to -\infty} \tilde{q}_s^2(u,z)$. The two lines then recombine and

we finally arrive at

$$\{q_s^2(z), C(u', z')\} = \sum_{\ell=0}^{s} \frac{(-1)^\ell (\ell+1)}{(s-\ell)!} \eth_z^{s-\ell} \partial_{u'}^{1-\ell} \Big( (-u')^{s-\ell} C(u', z) \eth_z^\ell \delta(z-z') \Big)$$

$$= \sum_{\ell=0}^{s} \sum_{n=0}^{\ell} \frac{(-1)^{s-n}(\ell+1)}{(s-\ell)!} \binom{\ell}{n} \partial_{u'}^{1-\ell} \Big( (u')^{s-\ell} \eth_{z'}^n C(u', z') \eth_z^{s-n} \delta(z-z') \Big), \qquad (6.31)$$

where for the second equality we have used (F.1b) and (F.1c) repeatedly.

The result (6.31) is the same as equation (60) of [89], although in this reference the starting point (56) corresponds only to the first term of our starting point (6.28). We therefore arrive at the same (correct) result but from a different starting point. The explanation for this apparent paradox is that after starting from the charges (56) which are *not* conserved (since they correspond to $\hat{q}_s^2$ and not $\tilde{q}_s^2$), reference [89] uses an identity on the first line of (58) which does not hold in general[19] (and this is indeed the only identity which we have *not* needed to use in the above derivation). Astonishingly, the combination of this formula with the starting point (56) conspire precisely in a way leading to the correct result (60), or here (6.31)! Note that this does not invalidate any of the other results of [89].

Let us now go back to the bracket (6.31). In order to continue, we need to use identities from pseudo-differential calculus which are derived in [89], and in particular the higher order integral Leibniz rule

$$\partial_{u'}^{1-\ell} \big( (u')^{s-\ell} C(u', z') \big) = (\Delta_{u'} - \ell)_{s-\ell} \partial_{u'}^{1-s} C(u', z'), \qquad (6.32)$$

where $\Delta_u := u\partial_u + 1$ and $(x)_\ell = x(x-1)\dots(x-\ell+1)$ is the falling factorial with $(x)_0 = 1$. Using this we obtain

$$\{q_s^2(z), C(u', z')\} = \sum_{\ell=0}^{s} \sum_{n=0}^{\ell} \frac{(-1)^{s-n}(\ell+1)}{(s-\ell)!} \binom{\ell}{n} (\Delta_{u'} - \ell)_{s-\ell} \partial_{u'}^{1-s} \eth_{z'}^n C(u', z') \eth_z^{s-n} \delta(z-z'). \qquad (6.33)$$

We can then switch the sums using $\sum_{\ell=0}^{s} \sum_{n=0}^{\ell} = \sum_{n=0}^{s} \sum_{\ell=n}^{s}$, and the sum over $\ell$ can be computed with the identity

$$\sum_{\ell=n}^{s} \frac{(\ell+1)!}{(\ell-n)!(s-\ell)!} (\Delta_{u'} - \ell)_{s-\ell} = \frac{(n+1)!}{(s-n)!} (\Delta_{u'} + 2)_{s-n}. \qquad (6.34)$$

This finally leads to the action of the quadratic charges on the shear and its complex conjugate in the form

$$\{q_s^2(z), C(u', z')\} = \sum_{n=0}^{s} \frac{(-1)^{s-n}(n+1)}{(s-n)!} (\Delta_{u'} + 2)_{s-n} \partial_{u'}^{1-s} \eth_{z'}^n C(u', z') \eth_z^{s-n} \delta(z-z'), \qquad (6.35a)$$

$$\{q_s^2(z), \bar{C}(u', z')\} = \sum_{n=0}^{s} \frac{(-1)^{s-n}(n+1)}{(s-n)!} (\Delta_{u'} - 2)_{s-n} \partial_{u'}^{1-s} \eth_{z'}^n \bar{C}(u', z') \eth_z^{s-n} \delta(z-z'). \qquad (6.35b)$$

We do not reproduce the detailed calculations leading to the second bracket, but they can be found in [89]. As a consistency check for the general formula (6.35a) we can expand explicitly the brackets for $s = 0, 1, 2, 3$,

---

[19]This identity would imply in particular that $f(u)\partial_u^{-a}\delta(u-u') = (-1)^a f(u)\partial_{u'}^{-a}\delta(u-u')$, which is true for $a \le 0$ but not for $a > 0$.

which gives

$$\{q_0^2(z), C(u', z')\} = N(u', z')\delta(z - z'), \tag{6.36a}$$

$$\{q_1^2(z), C(u', z')\} = -(u'\partial_{u'} + 3)C(u', z')\eth_z\delta(z - z') + 2\eth_{z'}C(u', z')\delta(z - z'), \tag{6.36b}$$

$$\{q_2^2(z), C(u', z')\} = \frac{u'^2}{2}N(u', z')\eth_z^2\delta(z - z') \tag{6.36c}$$

$$+ u'\Big(3C(u', z')\eth_z^2\delta(z - z') - 2\eth_{z'}C(u', z')\eth_z\delta(z - z')\Big)$$

$$+ 3\partial_{u'}^{-1}\Big(\eth_{z'}^2 C(u', z')\delta(z - z') - 2\eth_{z'}C(u', z')\eth_z\delta(z - z') + C(u', z')\eth_z^2\delta(z - z')\Big),$$

$$\{q_3^2(z), C(u', z')\} = -\frac{u'^3}{6}N(u', z')\eth_z^3\delta(z - z') \tag{6.36d}$$

$$+ \frac{u'^2}{2}\Big(2\eth_{z'}C(u', z')\eth_z^2\delta(z - z') - 3C(u', z')\eth_z^3\delta(z - z')\Big)$$

$$+ u'\partial_{u'}^{-1}\Big(6\eth_{z'}C(u', z')\eth_z^2\delta(z - z') - 3\eth_{z'}^2 C(u', z')\eth_z\delta(z - z') - 3C(u', z')\eth_z^3\delta(z - z')\Big)$$

$$+ \partial_{u'}^{-2}\Big(6\eth_{z'}C(u', z')\eth_z^2\delta(z - z') - 9\eth_{z'}^2 C(u', z')\eth_z\delta(z - z')$$

$$+ 4\eth_{z'}^3 C(u', z')\delta(z - z') - C(u', z')\eth_z^3\delta(z - z')\Big).$$

Using formula

$$f(z)\eth_z^s\delta(z - z') = \sum_{n=0}^{s}(-1)^n\binom{s}{n}\big(\eth_{z'}^n f(z')\big)\eth_z^{s-n}\delta(z - z'), \tag{6.37}$$

we can then check that these brackets agree with the linear part of the brackets (6.11), as they should.

A final identity is now required in order to compute the charge algebra. This is the action of the quadratic charges on the soft graviton operator $\bar{N}_s$. Using (6.35b) and the fact that $\bar{N}(u, z) = \partial_u\bar{C}(u, z)$ in (6.26), one can show that

$$\{q_{s_1}^2(z), \bar{N}_{s_2}(z')\} = \sum_{n=0}^{s_1}(n + 1)\binom{s_1 + s_2 - n}{s_2}\eth_{z'}^n\bar{N}_{s_1+s_2-1}(z')\eth_z^{s_1-n}\delta(z - z'). \tag{6.38}$$

The proof of this relation is given in appendix B of [89] (see also [90] for a proof in the so-called discrete basis) and relies on identities involving the operator $\Delta_u$.

### 6.4.3 Charge algebra

We now have all the necessary ingredients to compute the two terms in the local bracket (6.22), and then derive from this the smeared bracket (1.1). Starting from (6.38) and using $q_s^1(z) = \eth_z^{s+2}\bar{N}_s(z)$ we obtain

$$\{q_{s_1}^2(z), q_{s_2}^1(z')\} = \sum_{n=0}^{s_1}(n + 1)\binom{s_1 + s_2 - n}{s_2}\eth_{z'}^{s_2+2}\Big(\eth_{z'}^n\bar{N}_{s_1+s_2-1}(z')\eth_z^{s_1-n}\delta(z - z')\Big). \tag{6.39}$$

Let us now consider the smeared linear and quadratic charges

$$Q_s^1(Z) := \oint Z(z)q_s^1(z), \qquad Q_s^2(Z) := \oint Z(z)q_s^2(z), \tag{6.40}$$

where $Z$ has helicity $-s$. With this smearing, the bracket (6.39) can be integrated by parts and integrated over $\delta(z - z')$. As shown in appendix I, one can then derive the relation

$$\{Q_{s_1}^2(Z_1), Q_{s_2}^1(Z_2)\} = -(s_1 + 1)Q_{s_1+s_2-1}^1(Z_1\eth Z_2) + \oint \eth\!\!\!/ B_1, \tag{6.41}$$

where the second term on the right-hand side is such that $\bar{\eth}\slashed{\partial}B_1 - \bar{\eth}\slashed{\partial}B_2 = \eth B$, where $\bar{\eth}\slashed{\partial}B_2$ is obtained from $\bar{\eth}\slashed{\partial}B_1$ by swapping $(s_1 \leftrightarrow s_2, Z_1 \leftrightarrow Z_2)$. Note that $Z_1$ and $Z_2$ have respective spin weights $-s_1$ and $-s_2$. Using this, it is then immediate to obtain

$$
\begin{aligned}
\left\{ Q_{s_1}(Z_1), Q_{s_2}(Z_2) \right\}^{(1)} &= \left\{ Q_{s_1}^2(Z_1), Q_{s_2}^1(Z_2) \right\} + \left\{ Q_{s_1}^1(Z_1), Q_{s_2}^2(Z_2) \right\} \\
&= \left\{ Q_{s_1}^2(Z_1), Q_{s_2}^1(Z_2) \right\} - (s_1 \leftrightarrow s_2, Z_1 \leftrightarrow Z_2) \\
&= -Q_{s_1+s_2-1}^1 \big( (s_1+1) Z_1 \eth Z_2 - (s_2+1) Z_2 \eth Z_1 \big).
\end{aligned}
\tag{6.42}
$$

This is indeed the announced result (1.1), which we have derived starting from the quasi-conserved charges (6.18). The present proof is considerably shorter than that presented in [89] because we have chosen to work with the smeared charges. This allows to integrate the delta functions (which enables to bypass the use of many delta function identities) and to integrate by parts freely.

An important question which remains open is that of the structure of the charge bracket beyond the linear truncation. For the spin 0 and spin 1 charges, one can check that the bracket closes as expected beyond linear order. This can be seen from the heuristic calculations presented in appendix H (see (H.4)), or by checking explicitly the $\left\{ quadratic, quadratic \right\} = quadratic$ brackets $\left\{ Q_1^2(\bar{\mathcal{Y}}), Q_0^2(T) \right\} = -Q_0^2 \big( 2\bar{\mathcal{Y}}\eth T - T\eth \bar{\mathcal{Y}} \big)$ and $\left\{ Q_1^2(\bar{\mathcal{Y}}_1), Q_1^2(\bar{\mathcal{Y}}_2) \right\} = -Q_1^2 \big( 2\bar{\mathcal{Y}}_1 \eth \bar{\mathcal{Y}}_2 - 2\bar{\mathcal{Y}}_2 \eth \bar{\mathcal{Y}}_1 \big)$ using the ingredients given above. Starting with spin $s=2$ however the bare (6.3) and renormalized (6.10) charges start to involve cubic terms. This implies for example that the bracket between $Q_2$ and $Q_0$ cannot a priori close to $Q_1$ (unless a non-trivial cancelation happens) as in the linear truncation (1.1), since there will be a contribution of the type $\left\{ cubic, quadratic \right\} = cubic$ while $Q_1$ contains only linear and quadratic terms. This cubic contribution beyond linear order can actually already be seen from the transformation law (4.22), which implies that the bracket between $\mathcal{Q}_0$ and $\mathcal{Q}_2$ produces a term $\sigma_2 \mathcal{Q}_1$, which does indeed contain a cubic contribution. The study of such higher order contributions is deferred to future work. We note that the quadratic and cubic brackets were studied in [90, 91], however using the non-conserved charges (6.16), so it will be useful to generalize this work to the charges (6.18) and their cubic contributions.

Finally, let us end with a brief discussion on the bracket of real charges[20]. As established and discussed in detail in [95], the real part $\mathrm{Re}(Q_s)$ of the higher spin charges is related to the non-radiative multipole moments of the spacetime. The algebra of these multipole moments can therefore be obtained from the bracket of the real charges $\left\{ \mathrm{Re}(Q_{s_1}), \mathrm{Re}(Q_{s_2}) \right\}^{(1)}$, which in turn can be rewritten partly in terms of the mixed bracket between $Q_s$ and its complex conjugate $\bar{Q}_s$. This bracket is however missing from our analysis and that of [89]. In appendix J we partly fill this gap by gathering the ingredients necessary for the computation. This preliminary analysis shows that the mixed bracket $\left\{ Q_{s_1}(Z_1), \bar{Q}_{s_2}(Z_2) \right\}^{(1)}$ does not close for spins $s = 0, 1$ unless we restrict the smearing functions to satisfy $\bar{\eth} Z_1 = 0 = \eth Z_2$. For arbitrary higher spin charges, the bracket can only close if the structure of the algebra described by the smearing functions on the right-hand side contains inverse operators $\eth^{-1}$. This can be seen for example on the bracket (J.8). We will come back to a detailed study of this mixed bracket and of the bracket of the real charges in future work.

# 7 Conclusion and perspectives

In this work we have studied the subleading structure of asymptotically-flat spacetimes, and explained how the higher Bondi aspects appearing in the radial expansion of the transverse metric can be traded for higher spin charges forming the $w_{1+\infty}$ algebra (1.1). In the spirit of [27, 89, 91], this is a direct realization in the gravitational phase space of the symmetry structure unraveled in celestial holography [79, 80] and twistor

---

[20]I thank Geoffrey Compère for raising this question.

theory [69, 83, 84, 91, 92]. For this construction, we have mapped the metric formalism in Bondi–Sachs gauge to the Newman–Penrose formalism, and then studied the expansion of $\Psi_0$ along with its evolution equation. In particular, we have shown that the recursive Einstein evolution equations (3.21) can be obtained if we introduce non-local terms in the map (3.20) between $\Psi_0^{s-2}$ and the higher spin charges $\mathcal{Q}_s$. We have then proved formula (4.22) for the transformation under BMSW transformations of an arbitrary charge $\mathcal{Q}_s$ of spin $s$. This formula reproduces immediately the higher spin bracket (1.1) for $s_1 = 0, 1$ and $s_2 = s$.

After having studied the map between the metric and the Newman–Penrose formalism, and the definition of the higher spin charges, we have studied in section 6 the algebra of the higher spin charges. The main new result to come out of this analysis is formula (6.18) for the charges which are conserved in the radiative vacuum up to quadratic order. We have then verified that this formula for the renormalized charges leads to the correct action (6.35a) on the shear, and then used this result to compute the bracket (6.42). One of our goals, which was achieved, was to shorten and streamline the proof of this bracket given in [89]. This was made possible by the use of smeared generators (6.40) and the identity (6.41).

The upshot of this work and of the previous analysis [89] is rather surprising, since it strongly suggests that one can realize the $w_{1+\infty}$ algebra (1.1) on the phase space of asymptotically-flat spacetimes without the need to impose any self-dual condition or truncation (other than the linearization of the bracket at this stage). In order to clarify the status of this observation, several important open questions should be investigated.

- **Relation with self-dual gravity.** An important open question is whether a self-dual condition should play a role in the construction of the higher spin charges and in the analysis of their bracket. Indeed, no such condition has been required for our construction and that of [89], which is in sharp contrast with the constructions given e.g. in [80, 84]. However, as mentioned in the beginning of section 3.3, from the point of view of our construction this question is most likely related to the need of introducing the non-local quantities $\Psi_0^{s-2}$ in (3.20) in order to identify the higher spin charges $\mathcal{Q}_s$ and obtain the recursion relation (3.21). An interesting observation comes from comparing the non-local term (3.31) required at spin 4 with its equivalent formula in [104, 108]. This comparison suggests that it should be possible to find a choice of tetrad for which the self-dual condition $\bar{\sigma}_2 = 0$ sets $\Phi_0^2 = 0$. This would imply that the evolution equation (3.21) is exact at spin 4 in the self-dual theory. We keep this important investigation for future work.

- **Brackets beyond linear order.** An obvious extension of the present construction is to consider the bracket of renormalized higher spin charges beyond the linear truncation defined by (6.22). In short, the question is whether the algebra of higher spin charges closes beyond linear order. Using the discrete basis introduced in [90], it was shown in [91] that the algebra of the global charges closes at quadratic order, while for the local charges this was verified for the bracket of two spin 2 charges. As mentioned below (6.42), beyond quadratic order the bracket of e.g. $Q_2$ with $Q_0$ contains a contribution of the type $\{cubic, quadratic\} = cubic$, while $Q_1$ contains only linear and quadratic terms. This seems to suggest that the bracket cannot close in full generality in the same form as (1.1). There is still a chance that the full bracket will close, but the resulting structure is so far unknown. This point therefore deserves to be investigated.

- **Mixed helicity and real brackets.** Relatedly, the bracket between mixed helicity charges $Q_s$ and $\bar{Q}_s$ should also be investigated in detail. In particular, this bracket is required in order to compute the bracket of the real charges $\mathrm{Re}(Q_s)$ which are related to the canonical multipole moments via the dictionary established in [95]. We have gathered preliminary formulas for the computation of this bracket in appendix J, and shown that already for the brackets (J.6) it is necessary to impose chiral conditions $\eth Z_1 = 0$ and $\bar{\eth} Z_2 = 0$ on the symmetry parameters. Furthermore, for the higher spin

brackets the example (J.8) suggests that operators $\eth^{-1}$ are needed in order for the mixed helicity brackets to close. This is also an important question which will be studied in future work.

- **Relation with hidden symmetries.** An intriguing question is that of the possible relationship between the higher spin symmetries and so-called hidden symmetries generated by Killing–Yano or Killing–Stäckel tensors. Indeed, we have seen that the higher spin charges and symmetries appear naturally as the continuation of the tower of relationships between memories, soft gravitons, and asymptotic symmetries. At the level of the asymptotic charges (5.1), the mass is paired with a spin 0 symmetry parameter, and the angular momentum with a spin 1 vector. It therefore seems natural to try to interpret the higher spin charges as being associated with higher rank tensors (or scalars with higher Newman–Penrose spin weight). However, when studying the subleading BMS charges as we did in section 5.1, there are of course no independent symmetry parameters associated with the higher Bondi aspects $\mathcal{E}_{ab}^n$ or with $\Psi_0^{n-1}$. More work is therefore required if a relationship with hidden symmetries is to be uncovered. We note however that [87, 88] (see also references therein) establishes a relationship between the integrability of the anti-self-dual Einstein equations and hidden symmetries. This could potentially be used to reinterpret the $w_{1+\infty}$ symmetry in terms of hidden symmetries.

- **Relaxation of the boundary conditions.** A rather challenging task is to study whether the higher spin charges and their $w_{1+\infty}$ algebra can be defined with more relaxed boundary conditions, e.g. in the context of GBMS [41] or BMSW [49], for polyhomogeneous expansions [50, 116–118], or in the partial Bondi gauge [50, 51] (which generically contains $\partial_u q_{ab} \neq 0$ and a free boundary metric with $B_0 \neq 0$ and $U_0^a \neq 0$). Here we have started our analysis with BMSW in order to derive the general transformation law (4.22). This is also why the rewriting of the evolution equations for spin 4 and 5 contain $\overline{\eth} R$ (see e.g. (3.31)). However, the entire analysis of section 6 has been done with the implicit assumption that $\delta q_{ab} = 0$. Studying more relaxed boundary conditions or alternative gauge choices would enable to probe the "robustness" of the $w_{1+\infty}$ structure.

- **Higher spin symmetries at finite distance.** Finally, it would be interesting to study if subleading charges and symmetry algebras, possibly of the $w_{1+\infty}$ form, can appear on any null surface, and in particular on black hole or cosmological horizons. This question is in the spirit of attempts at finding the most general boundary charges and symmetries in general relativity [126, 128, 141–159]. One can also wonder if $w_{1+\infty}$ should play a role in topological theories such as three-dimensional gravity [160]. It was shown in [161, 162] that the $\text{BMS}_3$ and double Virasoro algebras can be obtained from the Sugawara constructions based on the quadratic Casimirs of the isometry groups. This begs the question of whether a $w$-algebra structure can be obtained by considering higher order Casimirs [163, 164].

# Acknowledgements

I would like to thank Glenn Barnich, Laurent Freidel, Daniele Pranzetti, Simone Speziale and Ana-Maria Raclariu for fruitful discussions and comments, Geoffrey Compère, Adrien Fiorucci, Roberto Oliveri, Romain Ruzziconi and Ali Seraj for their feedback, Maciej Dunajski for pointing out references, and Céline Zwikel for collaboration on related topics.

# Appendices

## A  Details on the Newman–Penrose formalism

Our choice of signature is $(-,+,+,+)$. We convert $p$-covariant and $q$-contravariant tensors $F^{b_1...b_q}_{a_1...a_p}$ into spin $s = p - q$ scalars through the projection $\mathcal{F}_s = m_1^{a_1} \ldots m_1^{a_p} \bar{m}_{b_1}^1 \ldots \bar{m}_{b_q}^1 F^{b_1...b_q}_{a_1...a_p}$ onto the frames (3.11) and their complex conjugates of respective spin $+1$ and $-1$. The spin-weighted derivatives, or GHP "edth" operators, are defined by [165]

$$\eth \mathcal{F}_s := \left( m_1^a \partial_a + 2s\beta_1 \right) \mathcal{F}_s = m_1^{a_1} \ldots m_1^{a_p} \bar{m}_{b_1}^1 \ldots \bar{m}_{b_q}^1 m_1^c D_c F^{b_1...b_q}_{a_1...a_p}, \tag{A.1a}$$

$$\bar{\eth} \mathcal{F}_s := \left( \bar{m}_1^a \partial_a + 2s\alpha_1 \right) \mathcal{F}_s = m_1^{a_1} \ldots m_1^{a_p} \bar{m}_{b_1}^1 \ldots \bar{m}_{b_q}^1 \bar{m}_1^c D_c F^{b_1...b_q}_{a_1...a_p}, \tag{A.1b}$$

where $2\beta_1 = -D_a m_1^a$ and $2\alpha_1 = D_a \bar{m}_1^a$. Their commutator is given by

$$\left[ \bar{\eth}, \eth \right] \mathcal{F}_s = \frac{s}{2} R \mathcal{F}_s, \tag{A.2}$$

where $R = R[q]$ is the Ricci scalar of the leading sphere metric $q_{ab}$. From this we get the commutation relations

$$\eth \bar{\eth}^2 \mathcal{F}_3 = \bar{\eth}^2 \eth \mathcal{F}_3 - \frac{5}{2} R \bar{\eth} \mathcal{F}_3 - \frac{3}{2} \bar{\eth} R \, \mathcal{F}_3, \tag{A.3a}$$

$$\eth \bar{\eth}^3 \mathcal{F}_4 = \bar{\eth}^3 \eth \mathcal{F}_4 - \frac{9}{2} R \bar{\eth}^2 \mathcal{F}_4 - \frac{3}{2} \bar{\eth} R \bar{\eth} \mathcal{F}_4 - 2 \bar{\eth}^2 R \, \mathcal{F}_4, \tag{A.3b}$$

$$\eth \bar{\eth}^n \mathcal{F}_s = \bar{\eth}^n \eth \mathcal{F}_s + \frac{R}{4} n(n - 2s - 1) \bar{\eth}^{n-1} \mathcal{F}_s + \sum_{k=1}^{n} \frac{(n-k)(n-k-1-s(k+2))}{2(k+2)(k+1)} \binom{n}{k} \bar{\eth}^k R \, \bar{\eth}^{n-k-1} \mathcal{F}_s, \tag{A.3c}$$

$$\bar{\eth} \eth^n \mathcal{F}_s = \eth^n \bar{\eth} \mathcal{F}_s + \frac{R}{4} n(n + 2s - 1) \eth^{n-1} \mathcal{F}_s + \sum_{k=1}^{n} \frac{(n-k)(n-k-1+s(k+2))}{2(k+2)(k+1)} \binom{n}{k} \eth^k R \, \eth^{n-k-1} \mathcal{F}_s. \tag{A.3d}$$

The various NP spin weights of objects encountered in the main text are

| | $\Psi_4$ | $\Psi_3$ | $\Psi_2$ | $\Psi_1$ | $\Psi_0$ | $\partial_u$ | $\eth$ | $\alpha$ | $\beta$ | $\sigma$ | $\lambda$ | $C$ | $\bar{N}$ | $m_1^a$ | $\bar{m}_1^a$ | $\mathcal{Y}$ |
|---|---|---|---|---|---|---|---|---|---|---|---|---|---|---|---|---|
| $s$ | $-2$ | $-1$ | $0$ | $1$ | $2$ | $0$ | $1$ | $-1$ | $1$ | $2$ | $2$ | $2$ | $-2$ | $1$ | $-1$ | $1$ |  (A.4)

and complex conjugation takes $s \to -s$. Starting from a tetrad $e_i^\mu = (\ell, n, m, \bar{m})$, we define the spin coefficients $\gamma_{ijk} := e_j^\mu e_k^\nu \nabla_\nu e_{i\mu}$, and then choose to write the expansion of these coefficients (for example in the case of $\alpha$) as

$$\alpha = \sum_{n=n_0}^{\infty} \frac{\alpha_n}{r^n}, \tag{A.5}$$

where $n_0$ is specific to each coefficient. All the terms in such an expansion have the same NP spin. For the tetrad (3.8) constructed in the main text we find

$$\alpha = \frac{1}{2}(\gamma_{124} - \gamma_{344}) = \frac{1}{2r}D_a\bar{m}_1^a - \frac{\bar{\sigma}_2\bar{\alpha}_1}{r^2} + \frac{\sigma_2\bar{\sigma}_2\alpha_1}{2r^3} + \mathcal{O}(r^{-4}),$$ 

$$\text{(A.6a)}$$

$$\beta = \frac{1}{2}(\gamma_{123} - \gamma_{343}) = -\frac{\bar{\alpha}_1}{r} + \frac{\sigma_2\alpha_1}{r^2} - \frac{1}{2r^3}\left(\sigma_2\bar{\sigma}_2\bar{\alpha}_1 + \Psi_1^0\right) + \frac{1}{6r^4}\left(2\bar{\eth}\Psi_0^0 - \alpha_1\Psi_0^0\right) + \mathcal{O}(r^{-5}),$$

$$\text{(A.6b)}$$

$$\gamma = \frac{1}{2}(\gamma_{122} - \gamma_{342}) = -\frac{\Psi_2^0}{2r^2} + \frac{1}{6r^3}\left(2\bar{\eth}\Psi_1^0 + \bar{\alpha}_1\bar{\Psi}_1^0 - \alpha_1\Psi_1^0\right) + \mathcal{O}(r^{-4}),$$

$$\text{(A.6c)}$$

$$\epsilon = \frac{1}{2}(\gamma_{121} - \gamma_{341}) = 0,$$

$$\text{(A.6d)}$$

$$\pi = \gamma_{421} = 0,$$

$$\text{(A.6e)}$$

$$\mu = \gamma_{423} = \frac{R}{4r} + \frac{1}{r^2}\left(\partial_u\bar{\sigma}_2\sigma_2 - \Psi_2^0\right) + \frac{1}{8r^3}\left(4\bar{\eth}\Psi_1^0 + \sigma_2\bar{\sigma}_2R\right) + \mathcal{O}(r^{-4}),$$

$$\text{(A.6f)}$$

$$\nu = \gamma_{422} = -\frac{\Psi_3^0}{r} + \mathcal{O}(r^{-2}),$$

$$\text{(A.6g)}$$

$$\lambda = \gamma_{424} = \frac{\partial_u\bar{\sigma}_2}{r} + \frac{\bar{\sigma}_2R}{4r^2} + \mathcal{O}(r^{-3}),$$

$$\text{(A.6h)}$$

$$\kappa = \gamma_{131} = 0,$$

$$\text{(A.6i)}$$

$$\tau = \gamma_{132} = -\frac{\Psi_1^0}{2r^3} + \frac{1}{6r^4}\left(2\bar{\eth}\Psi_0^0 - \sigma_2\bar{\Psi}_1^0\right) + \mathcal{O}(r^{-5}),$$

$$\text{(A.6j)}$$

$$\sigma = \gamma_{133} = \frac{1}{2}e^{-2B}\partial_r\gamma_{ab}m^am^b$$
$$= -\frac{C_{ab}m_1^am_1^b}{2r^2} - \frac{\Psi_0^0}{2r^4} - \frac{\Psi_0^1}{3r^5} - \frac{1}{4r^6}\left(\Psi_0^2 - \frac{\sigma_2\bar{\sigma}_2}{2}\Psi_0^0\right) - \frac{1}{5r^7}\left(\Psi_0^3 - \frac{\sigma_2\bar{\sigma}_2}{2}\Psi_0^1\right) + \mathcal{O}(r^{-8}),$$

$$\text{(A.6k)}$$

$$\rho = \gamma_{134} = \frac{1}{2}e^{-2B}\partial_r\ln\sqrt{\gamma}$$
$$= \frac{1}{r} + \frac{\sigma_2\bar{\sigma}_2}{2r^3} - \frac{1}{8r^5}\left(\sigma_2\bar{\Psi}_0^0 + \bar{\sigma}_2\Psi_0^0 + (\sigma_2\bar{\sigma}_2)^2\right) - \frac{1}{15r^6}\left(\sigma_2\bar{\Psi}_0^1 + \bar{\sigma}_2\Psi_0^1\right) + \mathcal{O}(r^{-7}).$$

$$\text{(A.6l)}$$

Similarly, we can also expand the tetrad vectors themselves. The vector $\ell = e^{-2B}\partial_r$ can be expanded using (2.11). The expansion of the null frames $m^\mu = (0, m^r, m^a)$ and $n^\mu = (1, n^r, n^a)$ is

$$m^r = \frac{m_1^r}{r} + \frac{m_2^r}{r^2} + \frac{m_3^r}{r^3} + \mathcal{O}(r^{-4}),$$

$$\text{(A.7a)}$$

$$m^a = \frac{m_1^a}{r} + \frac{m_2^a}{r^2} + \frac{m_3^a}{r^3} + \frac{m_4^a}{r^4} + \mathcal{O}(r^{-5}),$$

$$\text{(A.7b)}$$

$$n^r = n_0^r + \frac{n_1^r}{r} + \frac{n_2^r}{r^2} + \mathcal{O}(r^{-3}),$$

$$\text{(A.7c)}$$

$$n^a = \frac{n_3^a}{r^3} + \mathcal{O}(r^{-4}),$$

$$\text{(A.7d)}$$

with

$$m_1^r = \frac{1}{2}D^b C_{ab} m_1^a = -\bar{\eth}\bar{\eth}\sigma_2, \tag{A.8a}$$

$$m_2^r = \frac{1}{2}\left(\mathcal{P}_a - \frac{1}{2}C_{ab}D_c C^{bc} - \frac{1}{8}\partial_a[CC]\right)m_1^a = \frac{1}{2}\left(\Psi_1^0 - 2\sigma_2\eth\bar{\sigma}_2 - \eth(\sigma_2\bar{\sigma}_2)\right), \tag{A.8b}$$

$$m_3^r = \frac{1}{6}\left(\frac{6}{16}[CC]D^b C_{ab} + \frac{3}{16}C_{ab}\partial^b[CC] - C_{ab}\mathcal{P}^b - D^b\mathcal{E}_{ab}\right)m_1^a$$
$$= \sigma_2\left(\frac{1}{3}\bar{\Psi}_1^0 - \bar{\sigma}_2\bar{\eth}\sigma_2 - \frac{1}{2}\bar{\eth}(\sigma_2\bar{\sigma}_2)\right) - \frac{1}{6}\bar{\eth}\Psi_0^0, \tag{A.8c}$$

$$m_1^a = \sqrt{\frac{q_{\theta\theta}}{2q}}\left(\frac{\sqrt{q} + iq_{\theta\phi}}{q_{\theta\theta}}\delta_\theta^a - i\delta_\phi^a\right), \tag{A.8d}$$

$$m_2^a = -\frac{1}{2}C^{ab}m_b^1 = \sigma_2\bar{m}_1^a, \tag{A.8e}$$

$$m_3^a = \frac{1}{16}[CC]m_1^a = \frac{1}{2}\sigma_2\bar{\sigma}_2 m_1^a, \tag{A.8f}$$

$$m_4^a = -\frac{1}{6}\mathcal{E}_1^{ab}m_b^1 = -\frac{1}{6}\Psi_0^0\bar{m}_1^a, \tag{A.8g}$$

$$n_0^r = -\frac{R}{4}, \tag{A.8h}$$

$$n_1^r = M, \tag{A.8i}$$

$$n_2^r = \frac{1}{2}\left(V_1 + U_a^2 U_2^a\right) = -\frac{1}{8}\left(\frac{4}{3}\left(\bar{\eth}\Psi_1^0 + \eth\bar{\Psi}_1^0\right) + \sigma_2\bar{\sigma}_2 R\right), \tag{A.8j}$$

$$n_3^a = -\frac{1}{6}\mathcal{P}^a = -\frac{1}{6}\left(\Psi_1^0\bar{m}_1^a + \bar{\Psi}_1^0 m_1^a\right). \tag{A.8k}$$

From (A.6) and (A.8) we can then deduce

$$\left(\bar{m}_2^a\partial_a + 2s\alpha_2\right)\mathcal{F}_s = \bar{\sigma}_2\eth\mathcal{F}_s, \tag{A.9a}$$

$$\left(m_2^a\partial_a + 2s\beta_2\right)\mathcal{F}_s = \sigma_2\bar{\eth}\mathcal{F}_s, \tag{A.9b}$$

$$\left(\bar{m}_3^a\partial_a + 2s\alpha_3\right)\mathcal{F}_s = \frac{1}{2}\sigma_2\bar{\sigma}_2\bar{\eth}\mathcal{F}_s, \tag{A.9c}$$

$$\left(m_3^a\partial_a + 2s\beta_3\right)\mathcal{F}_s = \frac{1}{2}\sigma_2\bar{\sigma}_2\eth\mathcal{F}_s - s\Psi_1^0\mathcal{F}_s, \tag{A.9d}$$

$$\left(m_4^a\partial_a + 2s\beta_4\right)\mathcal{F}_s = \frac{2s}{3}\bar{\eth}\Psi_0^0\mathcal{F}_s - \frac{1}{6}\Psi_0^0\bar{\eth}\mathcal{F}_s, \tag{A.9e}$$

$$\left(n_3^a\partial_a + 2s\gamma_3\right)\mathcal{F}_s = \frac{2s}{3}\bar{\eth}\Psi_1^0\mathcal{F}_s - \frac{1}{6}\left(\Psi_1^0\bar{\eth} + \bar{\Psi}_1^0\eth\right)\mathcal{F}_s. \tag{A.9f}$$

The leading angular frames satisfy

$$m_a^1\bar{m}_b^1 + m_b^1\bar{m}_a^1 = q_{ab}, \tag{A.10a}$$

$$m_a^1\bar{m}_b^1 - m_b^1\bar{m}_a^1 = i\varepsilon_{ab}, \tag{A.10b}$$

$$D_a m_b^1 - D_b m_a^1 = 2i\beta_1\varepsilon_{ab}, \tag{A.10c}$$

$$m_1^b D_b m_1^a - m_1^a D_b m_1^b = 0, \tag{A.10d}$$

$$m_1^b D_b\bar{m}_1^a + \bar{m}_1^a D_b m_1^b = 0, \tag{A.10e}$$

$$im_1^a = -\varepsilon_{ab}m_b^1, \tag{A.10f}$$

$$\eth\bar{m}_1^a = \bar{\eth}m_1^a. \tag{A.10g}$$

# B   Various identities

Here we gather various useful identities and relations, involving in particular the contraction of tensors with the angular frames. Given a symmetric and trace-free tensor $F_{ab}$ and a vector $V_a$ we denote $\mathcal{F} := F_{ab} m_1^a m_1^b$ and $\mathcal{V} := V_a m_1^a$, and we have the following identities:

$$D^2 F_{ab} + R F_{ab} = 2 D^c D_{\langle a} F_{b \rangle c} \tag{B.1}$$

$$D^2 F_{ab} - R F_{ab} = 2 D_{\langle a} D^c F_{b \rangle c} \tag{B.2}$$

$$[CC] = 8 \sigma_2 \bar{\sigma}_2 \tag{B.3}$$

$$\widetilde{C}_{ab} m_1^a m_1^b = -2 i \sigma_2 \tag{B.4}$$

$$F^{ab} S_{ab} = \mathcal{F} \bar{\mathcal{S}} + \bar{\mathcal{F}} \mathcal{S} \tag{B.5}$$

$$\widetilde{F}^{ab} S_{ab} = i (\mathcal{F} \bar{\mathcal{S}} - \bar{\mathcal{F}} \mathcal{S}) \tag{B.6}$$

$$F^{ab} \bar{m}_a^1 \partial_b f = \bar{\mathcal{F}} \eth f \tag{B.7}$$

$$D_a V^a = \bar{\eth} \mathcal{V} + \eth \bar{\mathcal{V}} \tag{B.8}$$

$$D_a \widetilde{V}^a = i (\bar{\eth} \mathcal{V} - \eth \bar{\mathcal{V}}) \tag{B.9}$$

$$(D_a F_{bc}) m_1^a m_1^b m_1^c = \eth \mathcal{F} \tag{B.10}$$

$$(D^b F_{ab}) m_1^a = \bar{\eth} \mathcal{F} \tag{B.11}$$

$$(V^b F_{ab}) m_1^a = \bar{\mathcal{V}} \mathcal{F} \tag{B.12}$$

$$(S^{bc} D_b F_{ca}) m_1^a = \bar{\mathcal{S}} \eth \mathcal{F} \tag{B.13}$$

$$(S_{ca} D_b F^{bc}) m_1^a = \mathcal{S} \eth \bar{\mathcal{F}} \tag{B.14}$$

$$(S_{bc} D_a F^{bc}) m_1^a = \mathcal{S} \eth \bar{\mathcal{F}} + \bar{\mathcal{S}} \eth \mathcal{F} \tag{B.15}$$

$$(D_a V_b) m_1^a m_1^b = \eth \mathcal{V} \tag{B.16}$$

$$(D_a V_b) \bar{m}_1^a m_1^b = \bar{\eth} \mathcal{V} \tag{B.17}$$

$$(V^a D^b F_{ab}) m_1^a m_1^b = \mathcal{Y} \eth \bar{\mathcal{F}} + \bar{\mathcal{Y}} \bar{\eth} \mathcal{F} \tag{B.18}$$

$$(V^c D_c F_{ab}) m_1^a m_1^b = \bar{\mathcal{V}} \eth \mathcal{F} + \mathcal{V} \bar{\eth} \mathcal{F} \tag{B.19}$$

$$(V^c D_a F_{bc}) m_1^a m_1^b = \bar{\mathcal{V}} \eth \mathcal{F} \tag{B.20}$$

$$(F_{ac} D^c V_b) m_1^a m_1^b = \mathcal{F} \bar{\eth} \mathcal{V} \tag{B.21}$$

$$(F_{ac} D_b V^c) m_1^a m_1^b = \mathcal{F} \eth \bar{\mathcal{V}} \tag{B.22}$$

$$D^c (V_a F_{bc}) m_1^a m_1^b = \bar{\eth} (\mathcal{V} \mathcal{F}) \tag{B.23}$$

$$V^a D^b F_{ab} = \mathcal{V} \eth \bar{\mathcal{F}} + \bar{\mathcal{V}} \bar{\eth} \mathcal{F} \tag{B.24}$$

$$D^a D^b F_{ab} = \eth^2 \bar{\mathcal{F}} + \bar{\eth}^2 \mathcal{F} \tag{B.25}$$

$$i D^a D^b \widetilde{F}_{ab} = \eth^2 \bar{\mathcal{F}} - \bar{\eth}^2 \mathcal{F} \tag{B.26}$$

$$(D^2 F_{ab}) m_1^a m_1^b = 2 \eth \bar{\eth} \mathcal{F} + R \mathcal{F} = 2 \bar{\eth} \eth \mathcal{F} - R \mathcal{F} \tag{B.27}$$

$$(S_{ac} D^c D^d F_{bd}) m_1^a m_1^b = \mathcal{S} \bar{\eth}^2 \mathcal{F} \tag{B.28}$$

$$(S^{cd} D_c D_a F_{bd}) m_1^a m_1^b = \bar{\mathcal{S}} \eth^2 \mathcal{F} \tag{B.29}$$

$$(S^{cd} D_a D_b F_{cd}) m_1^a m_1^b = \bar{\mathcal{S}} \eth^2 \mathcal{F} + \mathcal{S} \eth^2 \bar{\mathcal{F}} \tag{B.30}$$

$$(S^{cd} D_c D_d F_{ab}) m_1^a m_1^b = \bar{\mathcal{S}} \eth^2 \mathcal{F} + \mathcal{S} \bar{\eth}^2 \mathcal{F} \tag{B.31}$$

$$(D_c S^{cd} D_a F_{bd}) m_1^a m_1^b = \eth \bar{\mathcal{S}} \eth \mathcal{F} \tag{B.32}$$

$$(D_c S^{cd} D_d F_{ab}) m_1^a m_1^b = \eth \bar{\mathcal{S}} \eth \mathcal{F} + \bar{\eth} \mathcal{S} \bar{\eth} \mathcal{F} \tag{B.33}$$

The mass and the dual mass are related to $\Psi_2^0$ by

$$2M = \Psi_2^0 + \bar{\Psi}_2^0 - \partial_u(\sigma_2\bar{\sigma}_2) = 2\Psi_2^0 + \eth^2\bar{\sigma}_2 - \bar{\eth}^2\sigma_2 - 2\sigma_2\partial_u\bar{\sigma}_2, \tag{B.34a}$$

$$2\mathcal{M} = \Psi_2^0 + \bar{\Psi}_2^0 = 2\Psi_2^0 + \eth^2\bar{\sigma}_2 - \bar{\eth}^2\sigma_2 + \bar{\sigma}_2\partial_u\sigma_2 - \sigma_2\partial_u\bar{\sigma}_2, \tag{B.34b}$$

$$2i\widetilde{\mathcal{M}} = \Psi_2^0 - \bar{\Psi}_2^0 = \bar{\eth}^2\sigma_2 - \eth^2\bar{\sigma}_2 - \bar{\sigma}_2\partial_u\sigma_2 + \sigma_2\partial_u\bar{\sigma}_2. \tag{B.34c}$$

We also have the following expressions for the contractions of the Lie derivative:

$$(\pounds_Y F_{ab})m_1^a m_1^b = \mathcal{Y}\bar{\eth}\mathcal{F} + \bar{\mathcal{Y}}\eth\mathcal{F} + 2\eth\bar{\mathcal{Y}}\mathcal{F}, \tag{B.35a}$$

$$(\pounds_Y F^{ab})m_a^1 m_b^1 = \mathcal{Y}\bar{\eth}\mathcal{F} + \bar{\mathcal{Y}}\eth\mathcal{F} - 2\bar{\eth}\mathcal{Y}\mathcal{F}, \tag{B.35b}$$

$$(\pounds_Y V_a)m_1^a = \mathcal{Y}\bar{\eth}\mathcal{V} + \bar{\mathcal{Y}}\eth\mathcal{V} + \eth\bar{\mathcal{Y}}\mathcal{V} + \eth\mathcal{Y}\bar{\mathcal{V}}, \tag{B.35c}$$

$$(\pounds_Y V^a)m_a^1 = \mathcal{Y}\bar{\eth}\mathcal{V} + \bar{\mathcal{Y}}\eth\mathcal{V} - \bar{\eth}\mathcal{Y}\mathcal{V} - \eth\mathcal{Y}\bar{\mathcal{V}}, \tag{B.35d}$$

together with their complex conjugate. This can then be converted into the spin-weighted Lie derivative $\mathcal{L}_{\mathcal{Y}}$ using (4.19).

## C  Spin 3 evolution with Bondi tetrad

In this appendix we study the evolution equation for $\Psi_0^1$ and the identification of the spin 3 functional $\mathcal{Q}_3$ using the Bondi tetrad (3.1). In order to lighten the notations, we drop the tilde used for the spin coefficients and Weyl scalars built from (3.1). With this tetrad, the expansions of the Weyl scalars and of the spin coefficients change drastically from the expressions we have used throughout the text and obtained with the tetrad (3.8).

The leading frame $m_1^a$ is again given by (3.11). For the spin coefficients however, we find that the quantities of interest for what follows have the new expansion

$$\alpha = \frac{1}{2r}D_a\bar{m}_1^a + \mathcal{O}(r^{-2}), \tag{C.1a}$$

$$\beta = -\frac{\bar{\alpha}_1}{r} + \frac{1}{2r^2}\left(2\alpha_1\sigma_2 - \bar{\alpha}_1\bar{\sigma}_2 - 3\bar{\alpha}_1\sigma_2 - 2\eth\epsilon_2\right) + \mathcal{O}(r^{-3}), \tag{C.1b}$$

$$\gamma = -\frac{\partial_u\epsilon_2}{r} + \mathcal{O}(r^{-2}), \tag{C.1c}$$

$$\epsilon = \frac{1}{4r^2}(\sigma_2 - \bar{\sigma}_2) + \mathcal{O}(r^{-3}), \tag{C.1d}$$

$$\pi = \frac{\eth\bar{\sigma}_2}{r^2} + \mathcal{O}(r^{-3}), \tag{C.1e}$$

$$\mu = \frac{R}{4r} + \mathcal{O}(r^{-2}), \tag{C.1f}$$

$$\tau = \frac{\bar{\eth}\sigma_2}{r^2} + \mathcal{O}(r^{-3}), \tag{C.1g}$$

$$\sigma = -\frac{C_{ab}m_1^a m_1^b}{2r^2} + \frac{4\epsilon_2\sigma_2}{r^3} + \mathcal{O}(r^{-4}). \tag{C.1h}$$

We should note in particular the appearance of $\epsilon_2$ since now $\epsilon \neq 0$. The effect of this term is seen on the expansion of the frame and of the Weyl scalars, which gives

$$m_2^a = 2\epsilon_2 m_1^a + \sigma_2 \bar{m}_1^a, \qquad \Psi_1^1 = -\bar{\eth}\Psi_0^0 + 2\epsilon_2\Psi_1^0, \qquad \Psi_2^1 = -\bar{\eth}\Psi_1^0. \tag{C.2}$$

With this, expanding (3.22d) leads to the evolution equation

$$\partial_u\Psi_0^1 = -\bar{\eth}\left(\eth\Psi_0^0 - 4\sigma_2\Psi_1^0\right) + 4\epsilon_2\left(\eth\Psi_1^0 - 3\sigma_2\Psi_2^0\right) - 4\gamma_1\Psi_0^0. \tag{C.3}$$

Since $\gamma_1 = -\partial_u \epsilon_2$ we can then use the spin 2 evolution equation (3.23d) to finally arrive at

$$\partial_u\big(\Psi_0^1 - 4\epsilon_2\Psi_0^0\big) = -\overline{\eth}\big(\eth\Psi_0^0 - 4\sigma_2\Psi_1^0\big). \tag{C.4}$$

This shows that, when using the tetrad (3.1), the identification of the spin 3 functional $\mathcal{Q}_3$ from the subleading term in $\Psi_0$ is through the non-linear shift $-\overline{\eth}\mathcal{Q}_3 = \Psi_0^1 - 4\epsilon_2\Psi_0^0$. The computation of $\Psi_0^1$ using (3.14) also reveals that $\Psi_0^1 = 6E_{ab}^2 m_1^a m_1^b + 4\epsilon_2\Psi_0^0$, so that at the end of the day, in terms of the data of the Bondi solution space, we still have $-\overline{\eth}\mathcal{Q}_3 = 6E_{ab}^2 m_1^a m_1^b$. This illustrates the subtleties which arise when mapping the metric formalism in BS gauge to the NP one.

# D  Commutator between $\partial_u$ and $\delta_\xi$

In this appendix we evaluate the commutator of the action of $\partial_u$ and $\delta_\xi$ on $\mathcal{Q}_s$. For this, we first compute the time evolution of the transformation law to obtain

$$\begin{aligned}
\partial_u\delta_\xi\mathcal{Q}_s = {}& \big(f\partial_u + \mathcal{L}_\mathcal{Y} + 4W\big)\eth\mathcal{Q}_{s-1} - (s+1)\big(f\partial_u + \mathcal{L}_\mathcal{Y} + 4W\big)\big(\sigma_2\mathcal{Q}_{s-2}\big) \\
& + (s+2)\eth W\,\mathcal{Q}_{s-1} + (s+2)\eth f\big(\eth\mathcal{Q}_{s-2} - s\sigma_2\mathcal{Q}_{s-3}\big).
\end{aligned} \tag{D.1}$$

Then, we use the commutation relation (E.7) to compute the transformation of the time-evolved higher spin charges, which gives

$$\begin{aligned}
\delta_\xi\partial_u\mathcal{Q}_s = {}& \delta_\xi\eth\mathcal{Q}_{s-1} - (s+1)\delta_\xi\big(\sigma_2\mathcal{Q}_{s-2}\big) \\
= {}& \eth\delta_\xi\mathcal{Q}_{s-1} - \omega\eth\mathcal{Q}_{s-1} - (s-1)\eth\omega\mathcal{Q}_{s-1} - (s+1)\delta_\xi\big(\sigma_2\mathcal{Q}_{s-2}\big) \\
= {}& \eth\Big[\big(f\partial_u + \mathcal{L}_\mathcal{Y} + 3W\big)\mathcal{Q}_{s-1} + (s+1)\eth f\mathcal{Q}_{s-2}\Big] \\
& - \omega\eth\mathcal{Q}_{s-1} - (s-1)\eth\omega\mathcal{Q}_{s-1} \\
& - (s+1)\big(f\partial_u + \mathcal{L}_\mathcal{Y} + 4W\big)\big(\sigma_2\mathcal{Q}_{s-2}\big) - (s+1)\eth^2 f\mathcal{Q}_{s-2} - s(s+1)\eth f\sigma_2\mathcal{Q}_{s-3} \\
= {}& \eth f\partial_u\mathcal{Q}_{s-1} + f\partial_u\eth\mathcal{Q}_{s-1} + \eth(\mathcal{L}_\mathcal{Y}\mathcal{Q}_{s-1}) + 3\eth W\,\mathcal{Q}_{s-1} + 3W\eth\mathcal{Q}_{s-1} + \cancel{(s+1)\eth^2 f\mathcal{Q}_{s-2}} + (s+1)\eth f\eth\mathcal{Q}_{s-2} \\
& - \omega\eth\mathcal{Q}_{s-1} - (s-1)\eth\omega\mathcal{Q}_{s-1} \\
& - (s+1)\big(f\partial_u + \mathcal{L}_\mathcal{Y} + 4W\big)\big(\sigma_2\mathcal{Q}_{s-2}\big) - \cancel{(s+1)\eth^2 f\mathcal{Q}_{s-2}} - s(s+1)\eth f\sigma_2\mathcal{Q}_{s-3} \\
= {}& \eth f\eth\mathcal{Q}_{s-2} - s\eth f\sigma_2\mathcal{Q}_{s-3} + f\partial_u\eth\mathcal{Q}_{s-1} + \eth(\mathcal{L}_\mathcal{Y}\mathcal{Q}_{s-1}) + 3\eth W\,\mathcal{Q}_{s-1} + 3W\eth\mathcal{Q}_{s-1} + (s+1)\eth f\eth\mathcal{Q}_{s-2} \\
& - \omega\eth\mathcal{Q}_{s-1} - (s-1)\eth\omega\mathcal{Q}_{s-1} \\
& - (s+1)\big(f\partial_u + \mathcal{L}_\mathcal{Y} + 4W\big)\big(\sigma_2\mathcal{Q}_{s-2}\big) - s(s+1)\eth f\sigma_2\mathcal{Q}_{s-3} \\
= {}& f\partial_u\eth\mathcal{Q}_{s-1} + \eth(\mathcal{L}_\mathcal{Y}\mathcal{Q}_{s-1}) + 3\eth W\,\mathcal{Q}_{s-1} + 3W\eth\mathcal{Q}_{s-1} + (s+2)\eth f\eth\mathcal{Q}_{s-2} \\
& - \omega\eth\mathcal{Q}_{s-1} - (s-1)\eth\omega\mathcal{Q}_{s-1} \\
& - (s+1)\big(f\partial_u + \mathcal{L}_\mathcal{Y} + 4W\big)\big(\sigma_2\mathcal{Q}_{s-2}\big) - s(s+2)\eth f\sigma_2\mathcal{Q}_{s-3}.
\end{aligned} \tag{D.2}$$

We have introduced the coloring to keep track of the various contributions. Subtracting these two results, we finally find that the commutator reduces to

$$\begin{aligned}
\big[\partial_u, \delta_\xi\big]\mathcal{Q}_s = {}& [\mathcal{L}_\mathcal{Y}, \eth]\mathcal{Q}_{s-1} + W\eth\mathcal{Q}_{s-1} + (s-1)\eth W\,\mathcal{Q}_{s-1} + \omega\eth\mathcal{Q}_{s-1} + (s-1)\eth\omega\mathcal{Q}_{s-1} \\
= {}& [\mathcal{L}_\mathcal{Y}, \eth]\mathcal{Q}_{s-1} - \psi\eth\mathcal{Q}_{s-1} - (s-1)\eth\psi\mathcal{Q}_{s-1},
\end{aligned} \tag{D.3}$$

where we have used the fact that $\omega = -W - \psi$.

# E  Identities in conformal gauge

In this appendix we prove two useful relations which hold in conformal gauge $q_{ab} = e^\phi q_{ab}^\circ$, and which we have used in order to prove the result $[\partial_u, \delta_\xi]\mathcal{Q}_s = 0$ of section 4.2. First, let us recall the definitions

$$\mathcal{Y} := Y_a m_1^a, \qquad \omega := -W - \psi, \qquad \psi := -\frac{1}{2} D_a Y^a = -\frac{1}{2}(\eth\bar{\mathcal{Y}} + \bar{\eth}\mathcal{Y}). \tag{E.1}$$

From the transformations laws (4.13a) and (4.11a), we deduce that when $q_{ab} = e^\phi q_{ab}^\circ$ we have

$$\delta_\xi \ln\sqrt{q} = \delta_\xi \phi = 2\omega, \qquad D_a Y_b + D_b Y_a = (D_c Y^c) q_{ab}, \tag{E.2}$$

meaning in particular that $Y^a$ satisfies the conformal Killing equation. Projecting the latter onto $m_1^a$ and $\bar{m}_1^a$ then leads to the conditions $\eth\mathcal{Y} = 0 = \bar{\eth}\bar{\mathcal{Y}}$, which in turn implies from (4.17) that

$$\delta_\xi m_1^a = -\omega m_1^a, \qquad \delta_\xi m_a^1 = \omega m_a^1. \tag{E.3}$$

Let us now consider the commutation relations

$$[\delta, \eth]\mathcal{Q}_s = (\delta m_1^a \partial_a + 2s\delta\beta_1)\mathcal{Q}_s, \qquad [\delta, \bar{\eth}]\mathcal{Q}_s = (\delta\bar{m}_1^a \partial_a + 2s\delta\alpha_1)\mathcal{Q}_s. \tag{E.4}$$

Using the variations

$$\delta\beta_1 = -\frac{1}{2}(D_a \delta m_1^a + \delta\Gamma_{ab}^b[q] m_1^a), \qquad \delta\alpha_1 = +\frac{1}{2}(D_a \delta\bar{m}_1^a + \delta\Gamma_{ab}^b[q]\bar{m}_1^a), \tag{E.5}$$

and the fact that $\Gamma_{ab}^b[q] = \partial_a \ln\sqrt{q}$, we then find

$$\delta_\xi\beta_1 = -\omega\beta_1 - \frac{1}{2}\eth\omega, \qquad \delta_\xi\alpha_1 = -\omega\alpha_1 + \frac{1}{2}\bar{\eth}\omega, \tag{E.6}$$

which therefore leads to the commutation relations

$$[\delta_\xi, \eth]\mathcal{Q}_s = -\omega\eth\mathcal{Q}_s - s\eth\omega\mathcal{Q}_s, \qquad [\delta_\xi, \bar{\eth}]\mathcal{Q}_s = -\omega\bar{\eth}\mathcal{Q}_s + s\bar{\eth}\omega\mathcal{Q}_s. \tag{E.7}$$

Using the definition (4.19) and the conditions $\eth\mathcal{Y} = 0 = \bar{\eth}\bar{\mathcal{Y}}$, we can also compute

$$\mathcal{L}_\mathcal{Y}(\eth\mathcal{Q}_s) = \mathcal{Y}\bar{\eth}\eth\mathcal{Q}_s + \bar{\mathcal{Y}}\eth^2\mathcal{Q}_s - \frac{s+1}{2}(\bar{\eth}\mathcal{Y} - \eth\bar{\mathcal{Y}})\eth\mathcal{Q}_s, \tag{E.8a}$$

$$\eth(\mathcal{L}_\mathcal{Y}\mathcal{Q}_s) = \eth\mathcal{Y}\bar{\eth}\mathcal{Q}_s + \mathcal{Y}\eth\bar{\eth}\mathcal{Q}_s + \eth\bar{\mathcal{Y}}\eth\mathcal{Q}_s + \bar{\mathcal{Y}}\eth^2\mathcal{Q}_s - \frac{s}{2}(\eth\bar{\eth}\mathcal{Y} - \eth^2\bar{\mathcal{Y}})\mathcal{Q}_s - \frac{s}{2}(\bar{\eth}\mathcal{Y} - \eth\bar{\mathcal{Y}})\eth\mathcal{Q}_s, \tag{E.8b}$$

which upon using (A.2) leads to the commutation relations

$$[\mathcal{L}_\mathcal{Y}, \eth]\mathcal{Q}_s = \psi\eth\mathcal{Q}_s + s\eth\psi\mathcal{Q}_s, \qquad [\mathcal{L}_\mathcal{Y}, \bar{\eth}]\mathcal{Q}_s = \psi\bar{\eth}\mathcal{Q}_s - s\bar{\eth}\psi\mathcal{Q}_s. \tag{E.9}$$

One should recall that these expressions have been obtained in the conformal gauge for simplicity. They can however easily be extended to the case of an arbitrary $q_{ab}$.

# F  Pseudo-differential, delta, and theta function identities

We make use of the identities

$$\delta(-u - u') = \delta(u + u'), \tag{F.1a}$$

$$\partial_u^n \delta(u - u') = (-1)^n \partial_{u'}^n \delta(u - u'), \qquad \text{for } n \geq 0, \tag{F.1b}$$

$$f(u)\delta(u - u') = f(u')\delta(u - u'), \tag{F.1c}$$

$$\partial_u^{-1}\delta(u - u') = \int_{+\infty}^{u} du'' \, \delta(u'' - u') = -\int_{-\infty}^{-u} du'' \, \delta(u'' + u') = -\theta(u' - u), \tag{F.1d}$$

$$\partial_u^{-1}\theta(u' - u) = \int_{+\infty}^{u} du'' \, \theta(u' - u'') = -(u' - u)\theta(u' - u), \tag{F.1e}$$

$$\partial_u^{-1}\big(f(u)\delta(u - u')\big) = \int_{+\infty}^{u} du'' \, f(u'')\delta(u'' - u') = -f(u')\theta(u' - u), \tag{F.1f}$$

$$\partial_u^{-n}\big(f(u)\delta(u - u')\big) = -\frac{(u - u')^{n-1}}{(n-1)!} f(u')\theta(u' - u) \qquad \text{for } n \geq 1, \tag{F.1g}$$

$$\partial_u^{-1}\big(f(u)\theta(u' - u)\big) = \int_{+\infty}^{u} du'' \, f(u'')\theta(u' - u'') = -\big(F(u') - F(u)\big)\theta(u' - u), \tag{F.1h}$$

where $F$ is the antiderivative of $f$. The identity (6.12) comes from the general integral Leibniz rule

$$\partial_u^{-1}(fg) = \sum_{n=0}^{\infty} (-1)^n \big(\partial_u^n f\big)\big(\partial_u^{-(n+1)}g\big), \tag{F.2}$$

which can also be used to obtain

$$\partial_u^{-1}\left(\frac{(-u)^s}{s!} f(u)\right) = \sum_{n=0}^{s} \frac{(-u)^n}{n!} \partial_u^{s-n-1} f(u). \tag{F.3}$$

# G  Action of $\mathcal{Q}_{0,1,2,3}$ on $C$

We give here the brackets of the charges $\mathcal{Q}_{0,1,2,3}$ with $C$ in the case where we keep the theta function $\theta(u' - u)$. These brackets are computed using the fundamental Poisson brackets (6.7) and the identities of appendix F. They are given by

$$\big\{\mathcal{Q}_0(u, z), \bar{N}(u', z')\big\} = \partial_{u'}\bar{N}(u', z)\delta(z - z')\theta(u' - u), \tag{G.1a}$$

$$\begin{aligned}
\big\{\mathcal{Q}_0(u, z), C(u', z')\big\} &= \big\{\mathcal{Q}_0^1(u, z), C(u', z')\big\} + \big\{\mathcal{Q}_0^2(u, z), C(u', z')\big\} \\
&= \big\{\partial_u^{-1}\eth_z \mathcal{Q}_{-1}(u, z), C(u', z')\big\} - \big\{\partial_u^{-1}(C\mathcal{Q}_{-2})(u, z), C(u', z')\big\} \\
&= \eth_z^2 \delta(z - z')\theta(u' - u) + \partial_{u'}\big(C(u', z)\theta(u' - u)\big)\delta(z - z'),
\end{aligned} \tag{G.1b}$$

$$\begin{aligned}
\big\{\mathcal{Q}_1(u, z), C(u', z')\big\} &= \big\{\partial_u^{-1}\eth_z \mathcal{Q}_0(u, z), C(u', z')\big\} - 2\big\{\partial_u^{-1}(C\mathcal{Q}_{-1})(u, z), C(u', z')\big\} \\
&= (u - u')\eth_z^3 \delta(z - z')\theta(u' - u) + \partial_{u'}\eth_z\big((u - u')C(u', z)\delta(z - z')\theta(u' - u)\big) \\
&\quad - 2C(u', z)\eth_z\delta(z - z')\theta(u' - u),
\end{aligned} \tag{G.1c}$$

$$\begin{aligned}
\big\{\mathcal{Q}_2(u, z), C(u', z')\big\} &= \big\{\partial_u^{-1}\eth_z \mathcal{Q}_1(u, z), C(u', z')\big\} - 3\big\{\partial_u^{-1}(C\mathcal{Q}_0)(u, z), C(u', z')\big\} \\
&= \frac{1}{2}(u - u')^2 \eth_z^4 \delta(z - z')\theta(u' - u) + \frac{1}{2}\partial_{u'}\eth_z^2\big((u - u')^2 C(u', z)\delta(z - z')\theta(u' - u)\big) \\
&\quad - 2(u - u')\eth_z\big(C(u', z)\eth_z\delta(z - z')\big)\theta(u' - u) \\
&\quad + 3H(u, u', z)\eth_z^2\delta(z - z')\theta(u' - u) + 3\partial_{u'}\big(C(u', z)H(u, u', z)\theta(u' - u)\big)\delta(z - z'),
\end{aligned} \tag{G.1d}$$

$$\{\mathcal{Q}_3(u,z), C(u',z')\} = \{\partial_u^{-1}\eth_z\mathcal{Q}_2(u,z), C(u',z')\} - 4\{\partial_u^{-1}(C\mathcal{Q}_1)(u,z), C(u',z')\} \tag{G.2a}$$

$$= \frac{1}{6}(u-u')^3\eth_z^5\delta(z-z')\theta(u'-u) + \frac{1}{6}\partial_{u'}\eth_z^3\Big((u-u')^3 C(u',z)\delta(z-z')\theta(u'-u)\Big)$$

$$- (u-u')^2\eth_z^2\Big(C(u',z)\eth_z\delta(z-z')\Big)\theta(u'-u)$$

$$+ 3(u-u')\eth_z\Big(C(u',z)^2\delta(z-z')\Big)\theta(u'-u)$$

$$+ 3\eth_z\Big(\big(\partial_{u'}^{-2}C(u',z) - \partial_u^{-2}C(u,z)\big)\big(\eth_z^2\delta(z-z') + N(u',z)\delta(z-z')\big)\Big)\theta(u'-u)$$

$$+ 3(u-u')\eth_z\Big(\partial_{u'}^{-1}C(u',z)\big(\eth_z^2\delta(z-z') + N(u',z)\delta(z-z')\big)\Big)\theta(u'-u)$$

$$- 4H(u,u',z)\Big(\eth_z\big(C(u',z)\delta(z-z')\big) + 2C(u',z)\eth_z\delta(z-z')\Big)\theta(u'-u)$$

$$- 4u'H(u,u',z)\eth_z\Big(\eth_z^2\delta(z-z') + N(u',z)\delta(z-z')\Big)\theta(u'-u)$$

$$+ 4\int_u^{u'} \mathrm{d}u''\, u''C(u'',z)\eth_z\Big(\eth_z^2\delta(z-z') + N(u',z)\delta(z-z')\Big)\theta(u'-u),$$

where we have used the colors to distinguish the origin of the various terms.

## H   Alternative proof of some brackets

In this appendix we give a rather straightforward proof of the bracket (1.1) for $Q_{0,1,2}$. We will deliberately keep this proof heuristic by dropping the various coordinate labels and delta functions, but the readers familiar with Poisson brackets in the Hamiltonian analysis of constrained systems will recognize standard (and innocent) notational shortcuts. In particular we will denote the fundamental Poisson bracket simply by $\{\bar{N}, C\} = 1$. Let us then consider the notation

$$\int := \int_{\mathcal{I}^+} \mathrm{d}^3x = \oint \int_{-\infty}^{+\infty} \mathrm{d}u, \tag{H.1}$$

and the boundary conditions $\bar{N}\big|_{\mathcal{I}^+_\pm} = 0 = N\big|_{\mathcal{I}^+_\pm}$. We then define the smeared fluxes by computing the integral over $\mathcal{I}^+$ of the time evolution of the charges (6.10), with time-independent test functions. Integrating by parts over $\eth$ and $\partial_u$ subject to the above boundary conditions, this gives

$$Q_0(T) = -\int T\partial_u\mathcal{Q}_0$$

$$= \int (TN + \eth^2 T)\bar{N}, \tag{H.2a}$$

$$Q_1(\bar{\mathcal{Y}}) = -\int \bar{\mathcal{Y}}\partial_u\big(\mathcal{Q}_1 - u\eth_u\mathcal{Q}_1\big)$$

$$= \int \Big((3\eth\bar{\mathcal{Y}} + 2\bar{\mathcal{Y}}\eth)C + u(\eth\bar{\mathcal{Y}}N + \eth^3\bar{\mathcal{Y}})\Big)\bar{N}, \tag{H.2b}$$

$$Q_2(Z) = -\int Z\partial_u\left(\mathcal{Q}_2 - u\eth\mathcal{Q}_1 + \frac{u^2}{2}\eth^2\mathcal{Q}_0 + 3\partial_u^{-1}C\mathcal{Q}_0\right)$$

$$= \int -3Z\partial_u^{-1}CC\partial_u\bar{N} - 3ZC\eth^2\bar{C} + \left(2u\eth(\eth ZC) + \frac{u^2}{2}\eth^4 Z + u\eth^2 ZC + \frac{u^2}{2}\eth^2 ZN\right)\bar{N}, \tag{H.2c}$$

where for the spin 2 we have assumed the extra boundary condition $C\big|_{\mathcal{I}^+_\pm} = 0$. Using the bracket $\{\bar{N}, C\} = 1$, we find that the action of the spin 0 and spin 1 fluxes on the shear is

$$\{Q_0(T), C\} = TN + \eth^2 T, \tag{H.3a}$$

$$\begin{aligned}\{Q_1(\bar{\mathcal{Y}}), C\} &= (3\eth\bar{\mathcal{Y}} + 2\bar{\mathcal{Y}}\eth)C + u(\eth\bar{\mathcal{Y}}N + \eth^3\bar{\mathcal{Y}}) \\ &= (3\eth\bar{\mathcal{Y}} + 2\bar{\mathcal{Y}}\eth)C + u\{Q_0(\eth\bar{\mathcal{Y}}), C\},\end{aligned} \tag{H.3b}$$

consistently reproducing (6.14). Similarly, we can compute the Poisson brackets of the spin 0 and spin 1 fluxes directly to find

$$\{Q_1(\bar{\mathcal{Y}}), Q_0(T)\} = -Q_0(2\bar{\mathcal{Y}}\eth T - T\eth\bar{\mathcal{Y}}), \tag{H.4a}$$

$$\{Q_1(\bar{\mathcal{Y}}_1), Q_1(\bar{\mathcal{Y}}_2)\} = -Q_1(2\bar{\mathcal{Y}}_1\eth\bar{\mathcal{Y}}_2 - 2\bar{\mathcal{Y}}_2\eth\bar{\mathcal{Y}}_1), \tag{H.4b}$$

where no linearization is actually required.

In order to compute the brackets involving the flux of the spin 2 charge, we now need to consider the linear truncation. Just like in (6.4), one can note that (H.2) contains soft, quadratic hard, and cubic contributions. The soft contributions involve only $\bar{N}$ and therefore commute among each other. For the linearized bracket between the spin 2 and spin 0 fluxes we then find

$$\begin{aligned}\{Q_2(Z), Q_0(T)\}^{(1)} &= \{Q_2^1(Z), Q_0^2(T)\} + \{Q_2^2(Z), Q_0^1(T)\} \\ &= \int u\eth^3(T\eth Z - 3Z\eth T)\bar{N} \\ &= -Q_1^1(3Z\eth T - T\eth Z),\end{aligned} \tag{H.5}$$

where we have used the boundary condition $C\big|_{\mathcal{I}^+_\pm} = 0$ in order to trade $\bar{C}$ for $-u\bar{N}$ by integration by parts. Similarly, for the spin 2 and spin 1 fluxes we get

$$\begin{aligned}\{Q_2(Z), Q_1(\bar{\mathcal{Y}})\}^{(1)} &= \{Q_2^1(Z), Q_1^2(\bar{\mathcal{Y}})\} + \{Q_2^2(Z), Q_1^1(\bar{\mathcal{Y}})\} \\ &= \int \frac{u^2}{2}\eth^4(2\bar{\mathcal{Y}}\eth Z - 3Z\eth\bar{\mathcal{Y}})\bar{N} \\ &= -Q_2^1(3Z\eth\bar{\mathcal{Y}} - 2\bar{\mathcal{Y}}\eth Z),\end{aligned} \tag{H.6}$$

While for the two spin 2 fluxes we find

$$\begin{aligned}\{Q_2(Z_1), Q_2(Z_2)\}^{(1)} &= \{Q_2^1(Z_1), Q_2^2(Z_2)\} + \{Q_2^2(Z_1), Q_2^1(Z_2)\} \\ &= \int \frac{u^3}{6}\eth^5(3Z_2\eth Z_1 - 3Z_1\eth Z_2)\bar{N} \\ &= -Q_3^1(3Z_1\eth Z_2 - 3Z_2\eth Z_1).\end{aligned} \tag{H.7}$$

Finally, we can also use these short-hand proofs to compute the bracket of the charges with their complex conjugates, and recover the results of appendix J. Imposing the conditions $\bar{\eth}T \overset{!}{=} 0$ and $\eth\mathcal{Y} \overset{!}{=} 0$ which are necessary for the brackets to close, we find

$$\{Q_0^2(T), \bar{Q}_1^1(\mathcal{Y})\} = -\bar{Q}_0^1(T\bar{\eth}\mathcal{Y}), \tag{H.8a}$$

$$\{Q_0^1(T), \bar{Q}_1^2(\mathcal{Y})\} = -Q_0^1(T\bar{\eth}\mathcal{Y}), \tag{H.8b}$$

consistently with the brackets (J.6a) and (J.7) found from the more detailed calculations of appendix J.

# I Proof of the bracket between soft and hard charges

In this appendix we give a proof of the bracket (6.41) between the quadratic hard and the soft charges. Starting from (6.39), we first derive the bracket of the smeared charges (6.40). This is done by integrating by parts over $\eth_{z'}$ and $\eth_z$, before then integrating over $z'$ using the delta function $\delta(z - z')$. This leads to the bracket of smeared charges

$$\mathcal{B} := \left\{ Q_{s_1}^2(Z_1), Q_{s_2}^1(Z_2) \right\} = \oint \sum_{n=0}^{s_1} (-1)^{s_1+s_2-n}(n+1)\binom{s_1+s_2-n}{s_2}\eth^{s_1-n}Z_1\eth^{s_2+2}Z_2\eth^n\bar{N}_{s_1+s_2-1}, \quad (I.1)$$

where the right-hand side should be understood as evaluated at $z$. Using the generalized integration by parts formula

$$f\eth^{n+1}g = (-1)^{n+1}\eth^{n+1}fg + \eth\left(\sum_{k=0}^{n}(-1)^k\eth^k f\eth^{n-k}g\right), \quad (I.2)$$

we can then free $Z_1$ and write the bracket as

$$\mathcal{B} = \oint \sum_{n=0}^{s_1} (-1)^{s_2}(n+1)\binom{s_1+s_2-n}{s_2}Z_1\eth^{s_1-n}\left(\eth^{s_2+2}Z_2\eth^n\bar{N}_{s_1+s_2-1}\right) + \oint \eth B_1', \quad (I.3)$$

where the boundary terms is

$$B_1' = \sum_{n=0}^{s_1}\sum_{k=0}^{s_1-n-1} (-1)^{s_1+s_2-n+k}(n+1)\binom{s_1+s_2-n}{s_2}\left(\eth^{s_1-n-1-k}Z_1\right)\eth^k\left(\eth^{s_2+2}Z_2\eth^n\bar{N}_{s_1+s_2-1}\right). \quad (I.4)$$

Let us now drop this boundary term since it does not contribute to the bracket. Using the general Leibniz rule, we can then write

$$\mathcal{B} = \oint \sum_{n=0}^{s_1}\sum_{k=0}^{s_1-n} (-1)^{s_2}(n+1)\binom{s_1+s_2-n}{s_2}\binom{s_1-n}{k}Z_1\eth^{s_2+2+k}Z_2\eth^{s_1-k}\bar{N}_{s_1+s_2-1}. \quad (I.5)$$

Now, the sum over $k$ can be extended to run from $k = 0$ to $k = s_1$ because of the vanishing binomial coefficients, and the sum over $n$ can be performed using

$$\sum_{n=0}^{s_1}(n+1)\binom{s_1+s_2-n}{s_2}\binom{s_1-n}{k} = \frac{(s_1+s_2+2)_2}{(k+s_2+2)_2}\binom{s_1+s_2}{s_2}\binom{s_1}{k}, \quad (I.6)$$

where $(x)_n$ is the falling factorial. This leads to

$$\mathcal{B} = \oint \sum_{k=0}^{s_1}(-1)^{s_2}\frac{(s_1+s_2+2)_2}{(k+s_2+2)_2}\binom{s_1+s_2}{s_2}\binom{s_1}{k}Z_1\eth^{s_2+2+k}Z_2\eth^{s_1-k}\bar{N}_{s_1+s_2-1}. \quad (I.7)$$

Using once again (I.2), we can now remove all but one derivative of $Z_2$ to obtain

$$\mathcal{B} = \oint \sum_{k=0}^{s_1}(-1)^{k+1}\frac{(s_1+s_2+2)_2}{(k+s_2+2)_2}\binom{s_1+s_2}{s_2}\binom{s_1}{k}\eth Z_2\eth^{s_2+k+1}\left(Z_1\eth^{s_1-k}\bar{N}_{s_1+s_2-1}\right) + \oint \eth B_1'', \quad (I.8)$$

where the new boundary terms is

$$B_1'' = \sum_{k=0}^{s_1}\sum_{\ell=0}^{s_2+k} (-1)^{s_2+\ell}\frac{(s_1+s_2+2)_2}{(k+s_2+2)_2}\binom{s_1+s_2}{s_2}\binom{s_1}{k}\left(\eth^{s_2+k-\ell}\eth Z_2\right)\eth^\ell\left(Z_1\eth^{s_1-k}\bar{N}_{s_1+s_2-1}\right). \quad (I.9)$$

Let us now drop this boundary term as well. The general Leibniz rule then leads to

$$\mathcal{B} = \oint \sum_{k=0}^{s_1} \sum_{p=0}^{s_2+k+1} (-1)^{k+1} \frac{(s_1+s_2+2)_2}{(k+s_2+2)_2} \binom{s_1+s_2}{s_2} \binom{s_1}{k} \binom{s_2+k+1}{p} \eth Z_2 \eth^p Z_1 \eth^{s_1+s_2+1-p} \bar{N}_{s_1+s_2-1}. \quad \text{(I.10)}$$

Now, one can use the formula

$$\sum_{k=0}^{s_1} (-1)^{k+1} \frac{(s_1+s_2+2)_2}{(k+s_2+2)_2} \binom{s_1+s_2}{s_2} \binom{s_1}{k} = -(s_1+1) \quad \text{(I.11)}$$

to isolate the term $p=0$ in the sum and obtain

$$\mathcal{B} = -(s_1+1) \oint Z_1 \eth Z_2 \eth^{s_1+s_2+1} \bar{N}_{s_1+s_2-1} + \oint \eth\!\!\!/ B_1, \quad \text{(I.12)}$$

where

$$\eth\!\!\!/ B_1 = \sum_{k=0}^{s_1} \sum_{p=1}^{s_2+k+1} (-1)^{k+1} \frac{(s_1+s_2+2)_2}{(k+s_2+2)_2} \binom{s_1+s_2}{s_2} \binom{s_1}{k} \binom{s_2+k+1}{p} \eth Z_2 \eth^p Z_1 \eth^{s_1+s_2+1-p} \bar{N}_{s_1+s_2-1}. \quad \text{(I.13)}$$

This is the bracket announced in (6.41), namely

$$\{Q_{s_1}^2(Z_1), Q_{s_2}^1(Z_2)\} = -(s_1+1)Q_{s_1+s_2-1}^1(Z_1 \eth Z_2) + \oint \eth\!\!\!/ B_1. \quad \text{(I.14)}$$

The notation is used to indicate that the extra term on the right-hand side is not a total derivative.

The rest of the proof relies on showing that when anti-symmetrizing this term in $(1,2)$ we obtain a total derivative. For this, let us integrate by parts over $(p-1)$ derivatives of $Z_1$ in (I.13) in order to obtain

$$\eth\!\!\!/ B_1 = \eth B + \sum_{k=0}^{s_1} \sum_{p=1}^{s_2+k+1} (-1)^{k+p} \frac{(s_1+s_2+2)_2}{(k+s_2+2)_2} \binom{s_1+s_2}{s_2} \binom{s_1}{k} \binom{s_2+k+1}{p} \eth Z_1 \eth^{p-1}\left( \eth Z_2 \eth^{s_1+s_2+1-p} \bar{N}_{s_1+s_2-1}\right), \quad \text{(I.15)}$$

where

$$B = \sum_{k=0}^{s_1} \sum_{p=1}^{s_2+k+1} \sum_{i=0}^{p-2} (-1)^{k+i+1} \frac{(s_1+s_2+2)_2}{(k+s_2+2)_2} \binom{s_1+s_2}{s_2} \binom{s_1}{k} \binom{s_2+k+1}{p} \eth^{p-1-i} Z_1 \eth^i\left( \eth Z_2 \eth^{s_1+s_2+1-p} \bar{N}_{s_1+s_2-1}\right). \quad \text{(I.16)}$$

Let us now consider $\eth\!\!\!/ B_2$, which is obtained from $\eth\!\!\!/ B_1$ by swapping $(s_1 \leftrightarrow s_2, Z_1 \leftrightarrow Z_2)$. Writing $\eth\!\!\!/ B_1$ in the form (I.15) and $\eth\!\!\!/ B_2$ in the form (I.13), we obtain

$$\eth\!\!\!/ B_1 - \eth\!\!\!/ B_2 = \eth B$$

$$+ \sum_{k=0}^{s_1} \sum_{p=1}^{s_2+k+1} (-1)^{k+p} \frac{(s_1+s_2+2)_2}{(k+s_2+2)_2} \binom{s_1+s_2}{s_2} \binom{s_1}{k} \binom{s_2+k+1}{p} \eth Z_1 \eth^{p-1}\left( \eth Z_2 \eth^{s_1+s_2+1-p} \bar{N}_{s_1+s_2-1}\right)$$

$$- \sum_{k=0}^{s_2} \sum_{p=1}^{s_1+k+1} (-1)^{k+1} \frac{(s_1+s_2+2)_2}{(k+s_1+2)_2} \binom{s_1+s_2}{s_1} \binom{s_2}{k} \binom{s_1+k+1}{p} \eth Z_1 \eth^p Z_2 \eth^{s_1+s_2+1-p} \bar{N}_{s_1+s_2-1}. \quad \text{(I.17)}$$

Now, we can check that the second and third lines cancel exactly. For this, we rewrite the second line as

$$\sum_{k=0}^{s_1}\sum_{p=1}^{s_2+k+1}(-1)^{k+p}\frac{(s_1+s_2+2)_2}{(k+s_2+2)_2}\binom{s_1+s_2}{s_2}\binom{s_1}{k}\binom{s_2+k+1}{p}\eth Z_1\eth^{p-1}\Big(\eth Z_2\eth^{s_1+s_2+1-p}\bar{N}_{s_1+s_2-1}\Big)$$

$$=\sum_{k=0}^{s_1}\sum_{p=1}^{s_2+k+1}\sum_{i=0}^{p-1}(-1)^{k+p}\frac{(s_1+s_2+2)_2}{(k+s_2+2)_2}\binom{s_1+s_2}{s_2}\binom{s_1}{k}\binom{s_2+k+1}{p}\binom{p-1}{i}\eth Z_1\eth^{p-i}Z_2\eth^{s_1+s_2+1-p+i}\bar{N}_{s_1+s_2-1}$$

$$=\sum_{k=0}^{s_1}\sum_{p=1}^{s_2+k+1}\sum_{j=1}^{p}(-1)^{k+p}\frac{(s_1+s_2+2)_2}{(k+s_2+2)_2}\binom{s_1+s_2}{s_2}\binom{s_1}{k}\binom{s_2+k+1}{p}\binom{p-1}{p-j}\eth Z_1\eth^j Z_2\eth^{s_1+s_2+1-j}\bar{N}_{s_1+s_2-1}$$

$$=\sum_{k=0}^{s_1}\sum_{j=1}^{s_2+k+1}\sum_{p=1}^{j}(-1)^{k+j}\frac{(s_1+s_2+2)_2}{(k+s_2+2)_2}\binom{s_1+s_2}{s_2}\binom{s_1}{k}\binom{s_2+k+1}{j}\binom{j-1}{j-p}\eth Z_1\eth^p Z_2\eth^{s_1+s_2+1-p}\bar{N}_{s_1+s_2-1},$$

(I.18)

and then notice that for a fixed $p$ we have

$$\sum_{k=0}^{s_1}\sum_{j=1}^{s_2+k+1}(-1)^{k+j}\frac{(s_1+s_2+2)_2}{(k+s_2+2)_2}\binom{s_1+s_2}{s_2}\binom{s_1}{k}\binom{s_2+k+1}{j}\binom{j-1}{j-p}$$

$$=\sum_{k=0}^{s_2}(-1)^{k+1}\frac{(s_1+s_2+2)_2}{(k+s_1+2)_2}\binom{s_1+s_2}{s_1}\binom{s_2}{k}\binom{s_1+k+1}{p}.$$

(I.19)

At the end of the day we finally arrive at

$$\slashed{\eth}B_1-\slashed{\eth}B_2=\eth B,$$

(I.20)

as announced.

## J    Bracket between $Q$ and $\bar{Q}$

In this appendix we study the bracket between the smeared renormalized charges $Q$ and their complex conjugate $\bar{Q}$. For this computation, we need the action of the quadratic renormalized charges on the positive helicity soft graviton operator, which is given by [89]

$$\{q_{s_1}^2(z),N_{s_2}(z')\}=\sum_{n=0}^{s_1}(n+1)\binom{s_1+s_2-n-4}{s_2-4}\eth_{z'}^n N_{s_1+s_2-1}(z')\eth_z^{s_1-n}\delta(z-z'),$$

(J.1)

and where we should recall that $\bar{N}_{\text{here}}=N_{[89]}$. Using the fact that $\bar{q}_s^1(z)=\bar{\eth}_z^{s+2}N_s(z)$, we then obtain

$$\{q_{s_1}^2(z),\bar{q}_{s_2}^1(z')\}=\sum_{n=0}^{s_1}(n+1)\binom{s_1+s_2-n-4}{s_2-4}\bar{\eth}_{z'}^{s_2+2}\Big(\eth_{z'}^n N_{s_1+s_2-1}(z')\eth_z^{s_1-n}\delta(z-z')\Big),$$

(J.2a)

$$\{\bar{q}_{s_2}^2(z'),q_{s_1}^1(z)\}=\sum_{n=0}^{s_2}(n+1)\binom{s_1+s_2-n-4}{s_1-4}\eth_z^{s_1+2}\Big(\bar{\eth}_z^n \bar{N}_{s_1+s_2-1}(z)\bar{\eth}_{z'}^{s_2-n}\delta(z-z')\Big),$$

(J.2b)

which are the two terms entering the bracket

$$\{q_{s_1}(z),\bar{q}_{s_2}(z')\}^{(1)}:=\{q_{s_1}^2(z),\bar{q}_{s_2}^1(z')\}+\{q_{s_1}^1(z),\bar{q}_{s_2}^2(z')\}.$$

(J.3)

Using smearing functions $Z_1$ and $Z_2$ of respective helicity $-s_1$ and $+s_2$, we can introduce smeared charges as in (6.40). After integrating by parts over $\eth_{z'}$ and $\eth_z$, and then integrating over $z'$ using the delta function $\delta(z - z')$, we obtain

$$\left\{Q^2_{s_1}(Z_1), \bar{Q}^1_{s_2}(Z_2)\right\} = \oint \sum_{n=0}^{s_1} (-1)^{s_1+s_2}(n+1)\binom{s_1+s_2-n-4}{s_2-4}\eth^n\left(\eth^{s_1-n}Z_1\bar{\eth}^{s_2+2}Z_2\right)N_{s_1+s_2-1}, \quad \text{(J.4a)}$$

$$\left\{\bar{Q}^2_{s_2}(Z_2), Q^1_{s_1}(Z_1)\right\} = \oint \sum_{n=0}^{s_2} (-1)^{s_1+s_2}(n+1)\binom{s_1+s_2-n-4}{s_1-4}\bar{\eth}^n\left(\bar{\eth}^{s_2-n}Z_2\eth^{s_1+2}Z_1\right)\bar{N}_{s_1+s_2-1}, \quad \text{(J.4b)}$$

where the binomial coefficients should be written in terms of $\Gamma$ functions in order to be well-defined for all spins.

This calculation shows that the bracket (J.3) cannot close without an additional input or manipulation. Indeed, since (J.4a) and (J.4b) feature respectively $N_{s_1+s_2-1}$ and $\bar{N}_{s_1+s_2-1}$, these brackets can only close to $\bar{Q}^1_{s_1+s_2-1}$ and $Q^1_{s_1+s_2-1}$. For this to be the case however, (J.4a) must be written as a smearing of $\bar{\eth}^{s_1+s_2+1}N_{s_1+s_2-1}$, while (J.4b) must be written as a smearing of $\eth^{s_1+s_2+1}\bar{N}_{s_1+s_2-1}$. For $s_1 = 0$ and $s_2 = s$ in (J.4a) this can be achieved by writing

$$\left\{Q^2_0(Z_1), \bar{Q}^1_s(Z_2)\right\} = -\oint \bar{\eth}Z_2\bar{\eth}^{s+1}\left(Z_1N_{s-1}\right)$$

$$= -\sum_{k=0}^{s+1}(-1)^k\binom{s+1}{k}\oint \bar{\eth}^{-k}\left(\bar{\eth}Z_2\bar{\eth}^kZ_1\right)\bar{\eth}^{s+1}N_{s-1}, \quad \text{(J.5)}$$

which therefore requires to invert $\bar{\eth}$ in the smearing function. Since we do not expect such terms to enter the bracket of e.g. the spin 0 and spin 1 charges, we can consider the restriction $\bar{\eth}Z_1 \stackrel{!}{=} 0$ on the smearing functions. With this extra condition we find the brackets

$$\left\{Q^2_0(Z_1), \bar{Q}^1_s(Z_2)\right\} = -\bar{Q}^1_{s-1}\left(Z_1\bar{\eth}Z_2\right), \quad \text{(J.6a)}$$

$$\left\{Q^2_1(Z_1), \bar{Q}^1_s(Z_2)\right\} = -\bar{Q}^1_s\left(2Z_1\eth Z_2 + (s-1)Z_2\eth Z_1\right). \quad \text{(J.6b)}$$

Similarly, in order for the bracket (J.4b) to close we must impose the condition $\eth Z_2 \stackrel{!}{=} 0$. We then find for example

$$\left\{Q^1_s(Z_1), \bar{Q}^2_1(Z_2)\right\} = (s-1)Q^1_s\left(Z_1\bar{\eth}Z_2\right). \quad \text{(J.7)}$$

For the bracket of two spin 2 contributions however, we now find

$$\left\{Q^2_2(Z_1), \bar{Q}^1_2(Z_2)\right\} = \oint \eth^2Z_1\bar{\eth}^4Z_2N_3 + 3Z_1\eth^2\bar{\eth}^4Z_2N_3 + 4\eth Z_1\eth\bar{\eth}^4Z_2N_3. \quad \text{(J.8)}$$

The above conditions on $Z_1$ and $Z_2$ are now not sufficient in order for this bracket to close because the presence of $N_3$ requires to have $\bar{\eth}^5$ in order to arrive at $\bar{Q}^1_3$. It therefore seems that for the higher spin charges it is necessary to invert $\bar{\eth}$ in the smearing functions in order to obtain a closed bracket (J.3).

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
