# Peer review of "Celestial $w_{1+\infty}$ charges and the subleading structure of asymptotically-flat spacetimes"

_SciPost Physics_

## Round 1 · Referee Report · Anonymous (Referee 1) · 2024-5-21

Strengths

  1. Very clearly written
  2. Presents a novel idea, namely that a preferred choice of tetrad simplifies the asymptotic expansion of the Einstein equations in GR. This idea may have far-reaching implications for holography in asymptotically flat spacetimes (where the simpler structures arising in this way can be traced back to the symmetries of a conformal field theory)

Report

In this paper the author demonstrates that a particular choice of frame near null infinity allows for the asymptotic Einstein equations to a certain subleading order in a $r^{-1}$ expansion to be written as system of recursive differential equations for a collection of higher spin-$s$ charges. The analysis is done explicitly up to and including $s = 5$. The $s = 5$ equation appears here for the first time. The analysis is greatly simplified by a convenient choice on frame (where certain spin connection coefficients vanish) when constructing the NP variables. The author derives the action of the BMSW on all spin-$s$ charges, generalizing the results of \cite{Freidel:2021qpz} and further identifies the charge aspects in an expansion of the Iyer-Wald charges at large-$r$. Finally, it is shown that a new set of renormalized charges constructed such that they are conserved in the vacuum satisfy a (linearized) $w_{1+\infty}$ algebra on the gravitational phase space. This generalizes the analysis of \cite{Freidel:2021ytz} in the presence of non-linear contributions to the higher-spin charges which render the non-linear charges conserved in the vacuum. The paper is very clearly written and contains a collection of new results on gravity in asymptotically flat spacetimes. I believe it will be useful in future studies of holography in this context. I therefore recommend it highly for publication in SciPost after the following (optional) comments are addressed.

Requested changes

  1. In the first paragraph of page 10, the spin coefficients $\kappa, \epsilon, \pi$ seem to be referred to without having been defined. Their tilde version is defined shortly below, for the Bondi frame, but it might be useful for the reader to have these defined when they are introduced (or at least have a reference to the definition).

  2. I didn't understand the sentence regarding the absence of anomalies in the transformation laws of various metric components (first sentence in section 4.1). I thought the anomalies were defined as inhomogeneous shifts in the metric components under supertranslations and superrotations which seem to be present (in eg. 4.14, 4.15). Please clarify?

  3. The author refers to the linear term in the charges as soft, and the quadratic and higher order terms as hard. However, the higher order terms may contain soft components (as was for example illustrated in section 5 of \cite{Freidel:2021dfs} by evaluating a Fourier transform of the relevant term). It seems plausible that the terms by which the spin-$s$ charges the author defines and those in \cite{Freidel:2021ytz} differ are all non-linear soft components. I was wondering if the author computed the Fourier transform of at least the corrections to the first few $s$ charges to clarify the nature of these terms? If so, perhaps this would deserve a comment.

Recommendation

Publish (easily meets expectations and criteria for this Journal; among top 50%)

  • validity: top
  • significance: high
  • originality: high
  • clarity: top
  • formatting: perfect
  • grammar: excellent

Author:  Marc Geiller  on 2024-11-22  [id 4983]

(in reply to Report 1 on 2024-05-21)

1) I have added a reference to the defining equations for the spin coefficients. 2) I have rewritten the paragraph in the beginning of section 4.1 in order to clarify the previously confusing comment about anomalies. 3) I did not compute the Fourier transform of the corrections to the higher spin charges. It would indeed be interesting to study the action of the cubic and higher order terms (for example (6.18) and (6.19) give the renormalized and conserved spin 4 charge) on the shear and their interpretation as collinear corrections to the soft theorems. In the present work however, I am following the terminology of \cite{Freidel:2021dfs} and \cite{Freidel:2021ytz}, and referring to ‘soft’ and ‘quadratic hard’ as the $k=1$ and $k=2$ terms in (6.4). The reason for this is that $k$ is the correct quantity which enables to decompose the bracket as in (6.22). I have added footnote 15 in order to mention this point.

Attachment:

NPcharges_rxOjNPH.pdf

---

## Round 1 · Referee Report · Geoffrey Compère (Referee 2) · 2024-5-23

Strengths

1- The phase space of Einstein gravity at null infinity is worked out up to fourth subleading order. 2- The Newman-Penrose formalism is developed in Bondi-Sachs gauge and an explicit dictionary is worked out with the metric formalism. 3- The article makes progress in the understanding of spin-weight 4 and 5 flux-balance laws. 4- There is a proposed redefinition of the spin 3 charge, which transforms under Weyl-BMS symmetries in accordance with a recursive evolution equation. 5- The article provides a correction to the definition of conserved higher spin charges up to quadratic order defined earlier in the literature such that it obeys the $w_{1+\infty}$ algebra at linear order.

Weaknesses

1- The construction of the curly higher Bondi aspects is plagued with ambiguities, which are left unresolved. (Indeed, they are defined from the Weyl scalars for a given tetrad choice). No discussion appears on how to alleviate such an ambiguity. 2- The definition of the spin 4 charge requires to introduce a non-local intermediate expression (a $u$ integral), contrary to the lower order charges. It is unclear why such a definition arise, what is its physical meaning and if could be avoided. 3- The conjectures at the end of Section 5.1. that symmetries associated with higher spin weight charges could arise from Killing-Yano tensors is not based on clear evidence.

Report

This paper is a technical tour-de-force that allows to strengthen the understanding of the subleading structure of asymptotically flat spacetimes on several fronts. Given the significant technical advancements that this article has made, which opens new pathways with clear potential, the criteria of acceptance 2 and 3 in SciPost Physics are met.

Requested changes

1- Regarding a comment after Eq. (1.1). The bracket is the linear bracket but the action of one charge on the other involves all terms present in the charge (the hard'' and ''soft'' pieces). So calling itsoft order'' could be misleading. 2- Page 6: The acronym BMSW is not explained. Instead, Weyl-BMS was used in the introduction. 3- Page 11: "it will not be able'' is grammatically incorrect in English. 4- Page 20. It is unclear how to derive the conformal weights $\Delta$ in table (4.16). Please provide a reference. 5- Page 21. Equation (4.21)-(4.22). It should be clearly stated what has been proven, and what is a conjecture. Are these equations for $s=-2,\dots, 2,3$ proven and conjectured for $s \geq 4$? 6- Page 22. The statement that the commutator (4.23) vanishes away from conformal gauge is not proven even if the authors claim that it can be proven. Could the author cite an unpublished note? Alternatively, the author could argue that such a tensorial equality holds even in the presence of a diffeomorphism on all quantities involved, if such an argument can be made. 7- Page 23. Paragraph starting Section 5. It should be stated that the conformal Killing equation holds globally and has 6 independent solutions. The local solutions that admit poles also have $\delta q_{ab} \neq 0$ at the poles, and integrations over the sphere will collect contributions for these poles. The global solutions are much more restrictive than given by the conditions $\eth \mathcal Y =0$. The subsequent analysis is ignoring the poles. If local solutions are admitted, the analysis of the charges needs to be revised to include a discussion about the poles within the integrations over the sphere, which are currently incorrectly neglected. 8- Page 35 after the discussion following Eq. (6.31). The authors of [89] use restrictive boundary conditions on the shear of the form that the shear follows off more rapidly than any power law of $u$ in order to prove the algebra for arbitrary spin weight $s$. Please clarify which boundary conditions on the shear are being used for this derivation. 9- "in details" should read "in detail" in English.

Recommendation

Ask for minor revision

  • validity: high
  • significance: high
  • originality: high
  • clarity: good
  • formatting: perfect
  • grammar: good

Author:  Marc Geiller  on 2024-11-22  [id 4982]

(in reply to Report 2 by Geoffrey Compère on 2024-05-23)

1) I have removed the reference to ‘soft order’ and added a more detailed footnote. 2) I have now introduced BMSW after the first occurence of BMS-Weyl in the last paragraph at the bottom of page 2. 3) I have rephrased this sentence. 4) I have added two references. 5) I have added a few sentences clarifying the status of these equations below (4.22). 6) Indeed, I believe that one can argue along these lines, that the identity is tensorial and can be proven to holds even with arbitrary sphere diffeomorphisms. I initially thought of including the extended proof in another appendix, but it is just calculation and does not contain anything too insightful. I have therefore cited instead an unpublished calculation. 7) I have added footnote 13 to clarify this point. Here I am indeed discarding local solutions and therefore discarding the poles as well. 8) Here we are using the same fall-offs as in [89], which are stated in (6.2). I have recalled these fall-offs at the very end of the introduction, in the second to last paragraph above section 2. Note that I have also removed a previously misleading comment below (6.31), which was suggesting that the fall-off conditions used in the present work are different from those in [89]. They are indeed identical. 9) I have corrected this everywhere it appears.

Attachment:

NPcharges.pdf

---

## Editorial Decision

resubmitted